# On the Optimization and Generalization of Multi-head Attention

**Puneesh Deora**[*]                                                      *puneeshdeora@ece.ubc.ca*
*University of British Columbia*

**Rouzbeh Ghaderi**[*]                                                      *rghaderi@ece.ubc.ca*
*University of British Columbia*

**Hossein Taheri**[*]                                                      *hossein@ucsb.edu*
*University of California, Santa Barbara*

**Christos Thrampoulidis**                                                  *cthrampo@ece.ubc.ca*
*University of British Columbia*

**Reviewed on OpenReview:** *https://openreview.net/forum?id=wTGjn7JvYK*

## Abstract

The training and generalization dynamics of the Transformer's core mechanism, namely the Attention mechanism, remain under-explored. Besides, existing analyses primarily focus on single-head attention. Inspired by the demonstrated benefits of overparameterization when training fully-connected networks, we investigate the potential optimization and generalization advantages of using multiple attention heads. Towards this goal, we derive convergence and generalization guarantees for gradient-descent training of a single-layer multi-head self-attention model, under a suitable realizability condition on the data. We then establish primitive conditions on the initialization that ensure realizability holds. Finally, we demonstrate that these conditions are satisfied for a simple tokenized-mixture model. We expect the analysis can be extended to various data-model and architecture variations.

## 1 Introduction

Transformers have emerged as a promising paradigm in deep learning, primarily attributable to their distinctive self-attention mechanism. Motivated by the model's state-of-the-art performance in natural language processing (Devlin et al., 2019; Brown et al., 2020; Raffel et al., 2020) and computer vision (Dosovitskiy et al., 2021; Radford et al., 2021; Touvron et al., 2021), the theoretical study of the attention mechanism has seen a notable surge in interest recently. Numerous studies have already explored the expressivity of Attention, e.g. (Baldi & Vershynin, 2022; Dong et al., 2021; Yun et al., 2020a;b; Sanford et al., 2023; Bietti et al., 2023), and initial findings regarding memory capacity have been very recently studied in (Baldi & Vershynin, 2022; Dong et al., 2021; Yun et al., 2020a;b; Mahdavi et al., 2023). In an attempt to comprehend optimization aspects of training attention models, Sahiner et al. (2022); Ergen et al. (2022) have investigated convex-relaxations, while Tarzanagh et al. (2023a) investigates the model's implicit bias. Additionally, Edelman et al. (2021) have presented capacity and Rademacher complexity-based generalization bounds for Self-Attention. However, the exploration of the *finite-time* optimization and generalization dynamics of gradient-descent (GD) for training attention models largely remains an open question.

Recent contributions in this direction, which serve as motivation for our work, include the studies by Jelassi et al. (2022); Li et al. (2023a); Oymak et al. (2023). These works concentrate on single-layer attention models with a *single attention head*. Furthermore, despite necessary simplifying assumptions made for the data, the

---

[*]These authors contributed equally. Alphabetical ordering used.

analyses are rather intricate and appear highly specialized on the individual attention and data model. These direct and highly specialized analyses present certain challenges. First, it remains uncertain whether they can be encompassed within a broader framework that can potentially be extended to more complex attention architectures and diverse data models. Second, they appear disconnected from existing frameworks that have been flourishing in recent years for conventional fully-connected and convolutional neural networks e.g., (Jacot et al., 2018; Ji & Telgarsky, 2020; Richards & Rabbat, 2021; Liu et al., 2020; Taheri & Thrampoulidis, 2023). Consequently, it is also unclear how the introduction of attention alters the analysis landscape.

In this work, we study the optimization and generalization properties of multi-head attention mechanism trained by gradient methods. Our approach specifically leverages the use of *multiple attention heads*. Despite the operational differences between attention heads in an attention model and hidden nodes in an MLP, we demonstrate, from an analysis perspective, that this parallelism enables the exploitation of frameworks developed for the latter to study the former. Particularly for the generalization analysis, we leverage recent advancements in the application of the algorithmic-stability framework to overparameterized MLPs (Richards & Kuzborskij, 2021; Taheri & Thrampoulidis, 2023).

**Contributions.** We study training and generalization of gradient descent optimization for a multi-head attention (MHA) layer with $H$ heads in a binary classification setting. For this setting, detailed in Section 2, we analyze training with logistic loss both the attention weights (parameterizing the softmax logits), as well as, the linear decoder that turns output tokens to label prediction.

In Section 3, we characterize key properties of the empirical loss $\widehat{L}$, specifically establishing that it is self-bounded and satisfies a key self-bounded weak-convexity property, i.e. $\lambda_{\min}(\nabla^2 \widehat{L}(\boldsymbol{\theta})) \gtrsim -\frac{\kappa}{\sqrt{H}} \widehat{L}(\boldsymbol{\theta})$ for a parameter $\kappa$ that depends only mildly on the parameter vector $\boldsymbol{\theta}$. Establishing these properties (and also quantifying $\kappa$) involves carefully computing and bounding the gradient and Hessian of the MHA layer, calculations that can be useful beyond the context of our paper.

In Sections 4.1-4.2, we present our training and generalization bounds in their most general form. The bounds are given in terms of the empirical loss $\widehat{L}(\boldsymbol{\theta})$ and the distance $\|\boldsymbol{\theta} - \boldsymbol{\theta}_0\|$ to initialization $\boldsymbol{\theta}_0$ of an appropriately chosen target vector $\boldsymbol{\theta}$. The distance to initialization also controls the minimum number of heads $H \gtrsim \|\boldsymbol{\theta} - \boldsymbol{\theta}_0\|^6$ required for the bounds to hold. The choice of an appropriate parameter $\boldsymbol{\theta}$ that makes the bounds tight is generically specific to the data setting and the chosen initialization. To guide such a choice, in Section 4.3, we formalize primitive and straightforward-to-check conditions on the initialization $\boldsymbol{\theta}_0$ that ensure it is possible to find an appropriate $\boldsymbol{\theta}$. In short, provided the model output at initialization is logarithmic on the train-set size $n$ and the data are separable with respect to the neural-tangent kernel (NTK) features of the MHA model with constant margin $\gamma$, then Corollary 2 shows that with step-size $\eta = \widetilde{O}(1)$ and $\Theta(n)$ gradient descent steps, the train loss and generalization gap is bounded by $\widetilde{\mathcal{O}}(1/n)$ provided only a polylogarithmic number of heads $H = \Omega(\log^6(n))$. We remark that the aforementioned NTK separability assumption, although related to, differs from the standard NTK analysis. Besides, while this assumption is sufficient to apply our general bounds, it is not a necessary condition.

In Section 5, we investigate a tokenized mixture data model with label-(ir)relevant tokens. We show that after one randomized gradient step from zero initialization, the NTK features of the MHA model separate the data with margin $\gamma_\star$. Thus, applying our general analysis from Section 4.1, we establish training and generalization bounds as described above, for a logarithmic number of heads. Towards assessing the optimality of these bounds, we demonstrate that MHA is expressive enough to achieve margin $\gamma_{\text{attn}}$ that is superior to $\gamma_\star$. The mechanism to reach $\gamma_{\text{attn}}$ involves selecting key-query weights of sufficiently large norm, which saturates the softmax nonlinearity by suppressing label-irrelevant tokens. We identify the large-norm requirement as a potential bottleneck in selecting those weights as target parameters in our theory framework and discuss open questions regarding extending the analytical framework into this specific regime.

The remaining parts are organised as follows. Proof sketches of our main training/generalization bounds are given in Section 6. The paper concludes in Section 7 with remarks on our findings' implications and open questions. Detailed proofs are in the appendix, where we also present synthetic numerical experiments.

**Related work.** We give a brief overview of the most relevant works on understanding optimization/generalization of self-Attention or its variants. Please see Section H for more detailed exposition.

Oymak et al. (2023) diverges from traditional self-Attention by focusing on a variant called prompt-Attention, aiming to gain understanding of prompt-tuning. Jelassi et al. (2022) shed light on how ViTs learn spatially localized patterns using gradient-based methods. Li et al. (2023a) provides sample complexity bounds for achieving zero generalization error on training three-layer ViTs for classification tasks for a similar tokenized mixture data model as ours. Contemporaneous work Tian et al. (2023) presents SGD-dynamics of single-layer attention for next-token prediction by re-parameterizing the original problem in terms of the softmax and classification logit matrices, while Tarzanagh et al. (2023b;a) study the implicit bias of training the softmax weights $\boldsymbol{W}$ with a fixed decoder $\boldsymbol{U}$. All these works focus on a single attention head; instead, we leverage the use of multiple heads to establish connections to the literature on GD training of overparameterized MLPs. Conceptually, Hron et al. (2020) drew similar connections, linking multi-head attention to a Gaussian process in the limit as the number of heads approaches infinity. In contrast, we study the more practical regime of finite heads and obtain *finite-time* optimization and generalization bounds.

Among the extensive studies on training/generalization of overparameterized MLPs, our work closely aligns with Nitanda et al. (2019); Ji & Telgarsky (2020); Cao & Gu (2019); Chen et al. (2020); Telgarsky (2022); Taheri & Thrampoulidis (2023) focusing on classification with logistic loss. Conceptually, our findings extend this research to attention models. The use of algorithmic-stability tools towards order-optimal generalization bounds for overparameterized MLPs has been exploited recently by Richards & Kuzborskij (2021); Richards & Rabbat (2021); Taheri & Thrampoulidis (2023); Lei et al. (2022). To adapt these tools to the MHA layer, we critically utilize the smoothness of the softmax function and derive bounds on the growth of the model's gradient/Hessian, which establish a self-bounded weak convexity property for the empirical risk (see Corollary 1). Our approach also involves training both the classifier and attention weights, necessitating several technical adjustments detailed in Section 6 and Appendix B.1.

## 2 Preliminaries

**Notation.** $\boldsymbol{\varphi}(\cdot) : \mathbb{R}^T \to \mathbb{R}^T$ denotes the softmax map and $\boldsymbol{\varphi}'(\boldsymbol{v}) \coloneqq \nabla \boldsymbol{\varphi}(\boldsymbol{v}) = \text{diag}(\boldsymbol{\varphi}(\boldsymbol{v})) - \boldsymbol{\varphi}(\boldsymbol{v})\boldsymbol{\varphi}(\boldsymbol{v})^\top$ its gradient at $\boldsymbol{v} \in \mathbb{R}^T$. For $t \in [T]$, $\boldsymbol{\varphi}_t(\boldsymbol{v})$ is the $t$-th entry of $\boldsymbol{\varphi}(\boldsymbol{v}) \in \mathbb{R}^T$. For $\boldsymbol{A} \in \mathbb{R}^{n \times m}$, $\boldsymbol{A}_{i,:}$ is its $i$-th row and $\boldsymbol{A}_{:,j}$ is its $j$-th column. Recall the induced matrix norm $\|\boldsymbol{A}\|_{p,q} = \max_{\|\boldsymbol{v}\|_p=1} \|\boldsymbol{A}\boldsymbol{v}\|_q$ and particularly the following: $\|\boldsymbol{A}\|_{2,\infty} = \max_{i\in[n]} \|\boldsymbol{A}_{i,:}\|$, $\|\boldsymbol{A}\|_{1,2} = \max_{j\in[m]} \|\boldsymbol{A}_{:,j}\|$, and $\|\boldsymbol{A}\|_{1,\infty} = \max_{j\in[m]} \|\boldsymbol{A}_{:,j}\|_\infty$. For simplicity, $\|\boldsymbol{A}\|, \|\boldsymbol{v}\|$ denote Euclidean norms and $\lambda_{\min}(\boldsymbol{A})$ the minimum eigenvalue. We let $a \wedge b = \min\{a, b\}$ and $a \vee b = \max\{a, b\}$. concat denotes vector concatenation. All logarithms are natural logarithms (base $e$). We represent the line segment between $\boldsymbol{w}_1, \boldsymbol{w}_2 \in \mathbb{R}^{d'}$ as $[\boldsymbol{w}_1, \boldsymbol{w}_2] = \{\boldsymbol{w} : \boldsymbol{w} = \alpha\boldsymbol{w}_1 + (1-\alpha)\boldsymbol{w}_2, \alpha \in [0, 1]\}$. Finally, to simplify the exposition we use "$\gtrsim$" or "$\lesssim$" notation to hide absolute constants. We also occasionally use standard notations $\mathcal{O}, \Omega$ and $\widetilde{\mathcal{O}}, \widetilde{\Omega}$ to hide poly-log factors. Unless otherwise stated these order-wise notations are with respect to the training-set size $n$. Whenever used, exact constants are specified in the appendix.

**Single-head Self-attention.** A single-layer self-attention head $\text{ATTN} : \mathbb{R}^{T \times d} \to \mathbb{R}^{T \times d}$ with context size $T$ and dimension $d$ parameterized by key, query and value matrices $\boldsymbol{W}_Q, \boldsymbol{W}_K \in \mathbb{R}^{d \times d_h}, \boldsymbol{W}_V \in \mathbb{R}^{d \times d_v}$ is given by:

$$\text{ATTN}(\boldsymbol{X}; \boldsymbol{W}_Q, \boldsymbol{W}_K, \boldsymbol{W}_V) \coloneqq \boldsymbol{\varphi}(\boldsymbol{X}\boldsymbol{W}_Q\boldsymbol{W}_K^\top\boldsymbol{X}^\top)\boldsymbol{X}\boldsymbol{W}_V \,.$$

Here, $\boldsymbol{X} = [\boldsymbol{x}_1, \boldsymbol{x}_2, \ldots, \boldsymbol{x}_T]^\top \in \mathbb{R}^{T \times d}$ is the input token matrix and $\boldsymbol{\varphi}(\boldsymbol{X}\boldsymbol{W}_Q\boldsymbol{W}_K^\top\boldsymbol{X}^\top) \in \mathbb{R}^{T \times T}$ is the attention matrix. (Softmax applied to a matrix acts row-wise.) To turn the Attention output in a prediction label, we compose ATTN with a linear projection head (aka decoder). Thus, the model's output is[*]

$$\Phi(\boldsymbol{X}; \boldsymbol{W}, \boldsymbol{U}) \coloneqq \langle \boldsymbol{U}, \boldsymbol{\varphi}(\boldsymbol{X}\boldsymbol{W}\boldsymbol{X}^\top)\boldsymbol{X} \rangle \,. \tag{1}$$

Note that we absorb the value weight matrix $\boldsymbol{W}_V$ into the projector $\boldsymbol{U} = [\boldsymbol{u}_1, \ldots, \boldsymbol{u}_T]^\top \in \mathbb{R}^{T \times d}$. Also, we parameterize throughout the key-query product matrix as $\boldsymbol{W} \coloneqq \boldsymbol{W}_Q\boldsymbol{W}_K^\top$.

**Multi-head Self-attention.** Our focus is on the multi-head attention (MHA) model with $H$ heads:

$$\sum_{h\in[H]} \text{ATTN}(\boldsymbol{X}; \boldsymbol{W}_{Qh}, \boldsymbol{W}_{Kh}, \boldsymbol{W}_{Vh})\boldsymbol{W}_{Oh},$$

---

[*]While we focus on (i) Full-projection: trainable matrix $\boldsymbol{U} \in \mathbb{R}^{T \times d}$, our results also apply to (ii) Pooling: $\boldsymbol{U} = \boldsymbol{u}\mathbb{1}_T^\top$ with trainable $\boldsymbol{u} \in \mathbb{R}^d$, and (iii) Last-token output: $\boldsymbol{U} = \begin{bmatrix} 0_{d\times(T-1)} & \boldsymbol{u} \end{bmatrix}^\top$ with trainable $\boldsymbol{u} \in \mathbb{R}^d$.

for output matrices $\boldsymbol{W}_{Oh} \in \mathbb{R}^{d_v \times d}$. Absorbing $\boldsymbol{W}_{Vh}\boldsymbol{W}_{Oh}$ into a projection layer (similar to the single-head attention) and parameterizing $\boldsymbol{W}_h := \boldsymbol{W}_{Qh}\boldsymbol{W}_{Kh}^\top$ we arrive at the following MHA model:

$$\widetilde{\Phi}(\boldsymbol{X}; \widetilde{\boldsymbol{W}}, \widetilde{\boldsymbol{U}}) := \frac{1}{\sqrt{H}} \sum_{h \in [H]} \Phi(\boldsymbol{X}; \boldsymbol{W}_h, \boldsymbol{U}_h) = \frac{1}{\sqrt{H}} \sum_{h \in [H]} \langle \boldsymbol{U}_h, \boldsymbol{\varphi}(\boldsymbol{X}\boldsymbol{W}_h\boldsymbol{X}^\top)\boldsymbol{X} \rangle, \tag{2}$$

parameterized by $\widetilde{\boldsymbol{W}} := \mathrm{concat}\left(\{\boldsymbol{W}_h\}_{h \in [H]}\right)$ and $\widetilde{\boldsymbol{U}} := \mathrm{concat}\left(\{\boldsymbol{U}_h\}_{h \in [H]}\right)$. The $1/\sqrt{H}$ scaling is analogous to the normalization in MLP literature e.g. (Du et al., 2019; Ji & Telgarsky, 2021; Richards & Kuzborskij, 2021), ensuring the model variance is of constant order when $\boldsymbol{U}_h$ is initialized $\mathcal{O}_H(1)$. Note that these relaxations sacrifice some generality since it is common practice to set $d_h$ and $d_v$ such that $d_v = d/H < d$, thus imposing low-rank restrictions on matrices $\boldsymbol{W}_{Qh}\boldsymbol{W}_{Kh}^\top$, $\boldsymbol{W}_{Vh}\boldsymbol{W}_{Oh}$. We defer a treatment of these to future work.

Throughout, we will use $\boldsymbol{\theta}_h := \mathrm{concat}(\boldsymbol{U}_h, \boldsymbol{W}_h) \in \mathbb{R}^{dT+d^2}$, to denote the trainable parameters of the $h$-attention head and $\widetilde{\boldsymbol{\theta}} := \mathrm{concat}(\{\boldsymbol{\theta}_h\}_{h \in [H]}) \in \mathbb{R}^{H(dT+d^2)}$ for the trainable parameters of the overall model. More generally, we use the convention of applying " $\widetilde{\cdot}$ " notation for quantities relating to the multi-head model. Finally, with some slight abuse of notation, we define: $\|\widetilde{\boldsymbol{\theta}}\|_{2,\infty} := \max_{h \in [H]} \|\boldsymbol{\theta}_h\|$.

**Training.** Given training set $(\boldsymbol{X}_i, y_i)_{i \in [n]}$, with $n$ IID samples, we minimize logistic-loss based empirical risk

$$\widehat{L}(\widetilde{\boldsymbol{\theta}}) := \frac{1}{n} \sum_{i \in [n]} \ell(y_i \widetilde{\Phi}(\boldsymbol{X}_i; \widetilde{\boldsymbol{\theta}})) := \frac{1}{n} \sum_{i \in [n]} \log(1 + e^{-y_i \widetilde{\Phi}(\boldsymbol{X}_i; \widetilde{\boldsymbol{\theta}})}).$$

Our analysis extends to any convex, smooth, Lipschitz and self-bounded loss.[†] The empirical risk is minimized as an approximation of the *test loss* defined as $L(\widetilde{\boldsymbol{\theta}}) := \mathbb{E}_{(\boldsymbol{X},y)}[\ell(y\widetilde{\Phi}(\boldsymbol{X}; \widetilde{\boldsymbol{\theta}}))]$. We consider standard gradient-descent (GD) applied to empirical risk $\widehat{L}$. Formally, initialized at $\widetilde{\boldsymbol{\theta}}^{(0)}$ and equipped with step-size $\eta > 0$, at each iteration $k \geq 0$, GD performs the following update:

$$\widetilde{\boldsymbol{\theta}}^{(k+1)} = \widetilde{\boldsymbol{\theta}}^{(k)} - \eta \nabla \widehat{L}(\widetilde{\boldsymbol{\theta}}^{(k)}).$$

## 3  Gradient and Hessian bounds of soft-max attention

This section establishes bounds on the gradient and Hessian of the logistic empirical risk $\widehat{L}(.)$ evaluated on the multi-head attention model. To do this, we first derive bounds on the Euclidean norm and spectral-norm for the gradient and Hessian of the self-attention model. In order to simplify notations, we state here the bounds for the single-head model (see App. A.1 for multi-head model): $\Phi(\boldsymbol{X}; \boldsymbol{\theta}) := \Phi(\boldsymbol{X}; \boldsymbol{W}, \boldsymbol{U}) = \langle \boldsymbol{U}, \boldsymbol{\varphi}(\boldsymbol{X}\boldsymbol{W}\boldsymbol{X}^\top)\boldsymbol{X} \rangle$.

**Lemma 1** (Gradient/Hessian formulas). *For all $\boldsymbol{a} \in \mathbb{R}^T$, $\boldsymbol{b}, \boldsymbol{c} \in \mathbb{R}^d$ the model's gradients satisfy:*

- $\nabla_{\boldsymbol{U}}\Phi(\boldsymbol{X}; \boldsymbol{\theta}) = \boldsymbol{\varphi}(\boldsymbol{X}\boldsymbol{W}\boldsymbol{X}^\top)\boldsymbol{X}$, *and* $\qquad \nabla_{\boldsymbol{W}}\Phi(\boldsymbol{X}; \boldsymbol{\theta}) = \sum_{t=1}^{T} \boldsymbol{x}_t \boldsymbol{u}_t^\top \boldsymbol{X}^\top \boldsymbol{\varphi}'(\boldsymbol{X}\boldsymbol{W}^\top \boldsymbol{x}_t)\boldsymbol{X}$.

- $\nabla_{\boldsymbol{W}}\langle \boldsymbol{a}, \nabla_{\boldsymbol{U}}\Phi(\boldsymbol{X}; \boldsymbol{\theta})\boldsymbol{b} \rangle = \sum_{t=1}^{T} \boldsymbol{x}_t a_t \boldsymbol{b}^\top \boldsymbol{X}^\top \boldsymbol{\varphi}'(\boldsymbol{X}\boldsymbol{W}^\top \boldsymbol{x}_t)\boldsymbol{X}$, *and*

$\nabla_{\boldsymbol{W}}\langle \boldsymbol{c}, \nabla_{\boldsymbol{W}}\Phi(\boldsymbol{X}; \boldsymbol{\theta})\boldsymbol{b} \rangle = \sum_{t=1}^{T} (\boldsymbol{c}^\top \boldsymbol{x}_t)\,\boldsymbol{x}_t \mathrm{d}^\top \boldsymbol{\varphi}'(\boldsymbol{X}\boldsymbol{W}^\top \boldsymbol{x}_t)\boldsymbol{X}$

$\qquad\qquad where\ \mathrm{d} := \mathrm{diag}(\boldsymbol{X}\boldsymbol{b})\boldsymbol{X}\boldsymbol{u}_t - \boldsymbol{X}\boldsymbol{u}_t\boldsymbol{b}^\top \boldsymbol{X}^\top \boldsymbol{\varphi}(\boldsymbol{X}\boldsymbol{W}^\top \boldsymbol{x}_t) - \boldsymbol{X}\boldsymbol{b}\boldsymbol{u}_t^\top \boldsymbol{X}^\top \boldsymbol{\varphi}(\boldsymbol{X}\boldsymbol{W}^\top \boldsymbol{x}_t)$.

These calculations imply the following useful bounds.

**Proposition 1** (Model Gradient/Hessian bounds). *The Euclidean norm of the gradient and the spectral norm of the Hessian of the single-head Attention model* (1) *are bounded as follows:*

- $\|\nabla_{\boldsymbol{\theta}}\Phi(\boldsymbol{X}; \boldsymbol{\theta})\| \leq 2\|\boldsymbol{X}\|_{2,\infty}^2 \sum_{t=1}^{T} \|\boldsymbol{X}\boldsymbol{u}_t\|_\infty + \sqrt{T}\|\boldsymbol{X}\|_{2,\infty}$.

- $\|\nabla_{\boldsymbol{\theta}}^2 \Phi(\boldsymbol{X}; \boldsymbol{\theta})\| \leq 6\,d\,\|\boldsymbol{X}\|_{2,\infty}^2 \|\boldsymbol{X}\|_{1,\infty}^2 \sum_{t=1}^{T} \|\boldsymbol{X}\boldsymbol{u}_t\|_\infty + 2\sqrt{T\,d}\,\|\boldsymbol{X}\|_{2,\infty}^2 \|\boldsymbol{X}\|_{1,\infty}$.

---

[†] A function $\ell : \mathbb{R} \to \mathbb{R}$ is self-bounded if $\exists\, C > 0$ such that $|\ell'(t)| \leq C\ell(t)$.

Next, we focus on the empirical loss $\widehat{L}$. To derive bounds on its gradient and Hessian, we leverage the model's bounds from Proposition 1 and the fact that logistic loss is self-bounded, i.e., $|\ell'(t)| \le \ell(t)$. To provide concrete statements, we introduce first a mild boundedness assumption.

**Assumption 1** (Bounded data). *Data* $(\boldsymbol{X}, y) \in \mathbb{R}^{T \times d} \times \mathbb{R}$ *satisfy the following conditions almost surely:* $y \in \{\pm 1\}$, *and for some* $R \ge 1$, *it holds for all* $t \in [T]$ *that* $\|\boldsymbol{x}_t\| \le R$.

**Corollary 1** (Loss properties). *Under Assumption 1, the objective's gradient and Hessian satisfy the bounds:*[‡]

*(1)*  $\|\nabla \widehat{L}(\widetilde{\boldsymbol{\theta}})\| \le \beta_1(\widetilde{\boldsymbol{\theta}})\, \widehat{L}(\widetilde{\boldsymbol{\theta}}),$    $\beta_1(\widetilde{\boldsymbol{\theta}}) := \sqrt{T}\, R\, \left(2\, R^2\, \|\widetilde{\boldsymbol{\theta}}\|_{2,\infty} + 1\right).$

*(2)*  $\|\nabla^2 \widehat{L}(\widetilde{\boldsymbol{\theta}})\| \le \beta_2(\widetilde{\boldsymbol{\theta}}),$    $\beta_2(\widetilde{\boldsymbol{\theta}}) := \frac{1}{\sqrt{H}}\, \beta_3(\widetilde{\boldsymbol{\theta}}) + \frac{1}{4}\, \beta_1(\widetilde{\boldsymbol{\theta}})^2.$

*(3)*  $\lambda_{\min}(\nabla^2 \widehat{L}(\widetilde{\boldsymbol{\theta}})) \ge -\frac{\beta_3(\widetilde{\boldsymbol{\theta}})}{\sqrt{H}} \widehat{L}(\widetilde{\boldsymbol{\theta}})$    $\beta_3(\widetilde{\boldsymbol{\theta}}) := 2\sqrt{T\,d}\, R^3\, \left(3\sqrt{d}\, R^2\, \|\widetilde{\boldsymbol{\theta}}\|_{2,\infty} + 1\right).$

The loss properties above are crucial for the training and generalization analysis. Property (1) establishes self-boundedness of the empirical loss, which is used to analyze stability of GD updates for generalization. Property (2) is used to establish descent of gradient updates for appropriate choice of step-size $\eta$. Note that the smoothness upper bound is $\widetilde{\boldsymbol{\theta}}$-dependent, hence to show descent we need to also guarantee boundedness of the updates. Finally, property (3) establishes a self-bounded weak-convexity property of the loss, which is crucial to both the training and generalization analysis. Specifically, as the number of heads $H$ increases, the minimum eigenvalue becomes less negative, indicating an approach towards convex-like behavior.

## 4 Main results

In this section, we present our training and generalization bounds for multi-head attention.

### 4.1 Training bounds

We state our main result on train loss convergence in the following theorem. See App. B for exact constants and the detailed proofs.

**Theorem 1** (Training loss). *Fix iteration horizon* $K \ge 1$ *and any* $\widetilde{\boldsymbol{\theta}} \in \mathbb{R}^{H(dT + d^2)}$ *and* $H$ *satisfying*

$$\sqrt{H} \gtrsim d\, T^{1/2} R^5 \|\widetilde{\boldsymbol{\theta}}\|_{2,\infty} \|\widetilde{\boldsymbol{\theta}} - \widetilde{\boldsymbol{\theta}}^{(0)}\|^3. \tag{3}$$

*Fix step-size* $\eta \le 1 \wedge 1/\rho(\widetilde{\boldsymbol{\theta}}) \wedge \frac{\|\widetilde{\boldsymbol{\theta}} - \widetilde{\boldsymbol{\theta}}^{(0)}\|^2}{K \widehat{L}(\widetilde{\boldsymbol{\theta}})} \wedge \frac{\|\widetilde{\boldsymbol{\theta}} - \widetilde{\boldsymbol{\theta}}^{(0)}\|^2}{\widehat{L}(\widetilde{\boldsymbol{\theta}}^{(0)})}$, *with* $\rho(\widetilde{\boldsymbol{\theta}}) \lesssim d^{3/2}\, T^{3/2} R^{13} \|\widetilde{\boldsymbol{\theta}}\|_{2,\infty}^2 \|\widetilde{\boldsymbol{\theta}} - \widetilde{\boldsymbol{\theta}}^{(0)}\|^2$. *Then, the following bounds hold for the training loss and the weights' norm at iteration* $K$ *of GD:*

$$\widehat{L}(\widetilde{\boldsymbol{\theta}}^{(K)}) \le \frac{1}{K} \sum_{k=1}^{K} \widehat{L}(\widetilde{\boldsymbol{\theta}}_k) \le 2\widehat{L}(\widetilde{\boldsymbol{\theta}}) + \frac{5\|\widetilde{\boldsymbol{\theta}} - \widetilde{\boldsymbol{\theta}}^{(0)}\|^2}{4\eta K}, \tag{4}$$

$$\|\widetilde{\boldsymbol{\theta}}^{(K)} - \widetilde{\boldsymbol{\theta}}^{(0)}\| \le 4\|\widetilde{\boldsymbol{\theta}} - \widetilde{\boldsymbol{\theta}}^{(0)}\|.$$

Yielding a concrete train loss bound requires an appropriate set of target parameters $\widetilde{\boldsymbol{\theta}}$ in the sense of minimizing the bound in (4). Hence, $\widetilde{\boldsymbol{\theta}}$ should simultaneously attain small loss ($\widehat{L}(\widetilde{\boldsymbol{\theta}})$) and distance to initialization ($\|\widetilde{\boldsymbol{\theta}} - \widetilde{\boldsymbol{\theta}}^{(0)}\|$). This desiderata is formalized in Assumption 2 below. The distance to initialization, as well as $\|\widetilde{\boldsymbol{\theta}}\|_{2,\infty}$, determine how many heads are required for our bounds to hold. Also, in view of the bound in (4), it is reasonable that an appropriate choice for $\widetilde{\boldsymbol{\theta}}$ attains $\widehat{L}(\widetilde{\boldsymbol{\theta}})$ of same order as $\|\widetilde{\boldsymbol{\theta}} - \widetilde{\boldsymbol{\theta}}^{(0)}\|^2/K$. Hence, the theorem's restriction on the step-size is governed by the inverse local-smoothness of the loss: $\eta \lesssim 1/\rho(\widetilde{\boldsymbol{\theta}})$.

### 4.2 Generalization bounds

Next we bound the expected generalization gap. Expectations are with respect to (w.r.t) randomness of the train set. See App. C for the detailed proof, which is based on algorithmic-stability.

---

[‡]In all the bounds in this paper involving $\|\widetilde{\boldsymbol{\theta}}\|_{2,\infty}$, it is possible to substitute this term with $\max_{h \in [H]} \|\boldsymbol{U}_h\|$. However, for the sake of notation simplicity, we opt for a slightly looser bound $\max_{h \in [H]} \|\boldsymbol{U}_h\| \le \max_{h \in [H]} \|\boldsymbol{\theta}_h\| =: \|\widetilde{\boldsymbol{\theta}}\|_{2,\infty}$.

**Theorem 2** (Generalization loss). *Fix any $K \geq 1$, any $\widetilde{\boldsymbol{\theta}}$ and $H$ satisfying (3), and any step-size $\eta$ satisfying the conditions of Thm. 1. Then the expected generalization gap at iteration $K$ satisfies,*

$$\mathbb{E}\big[L(\widetilde{\boldsymbol{\theta}}^{(K)}) - \widehat{L}(\widetilde{\boldsymbol{\theta}}^{(K)})\big] \leq \frac{4}{n}\, \mathbb{E}\Big[2\,K\,\widehat{L}(\widetilde{\boldsymbol{\theta}}) + \frac{9\|\widetilde{\boldsymbol{\theta}} - \widetilde{\boldsymbol{\theta}}^{(0)}\|^2}{4\eta}\Big]. \tag{5}$$

The condition on the number of heads is same up to constants to the corresponding condition in Theorem 1. Also, the generalization-gap bound translates to test-loss bound by combining with Thm. 1. Finally, similar to Thm. 1, we can get concrete bounds under the realizability assumption; see Cor. 4 in App. C.2. For the generalization analysis, we require that the realizability assumption holds almost surely over all training sets sampled from the data distribution.

The bounds on optimization and generalization are up to constants same as analogous bounds for logistic regression (Soudry et al., 2018; Ji & Telgarsky, 2018; Shamir, 2021). Yet, for these bounds to be valid, we require sufficiently large number of heads as well as the existence of an appropriate set of target parameters $\widetilde{\boldsymbol{\theta}}$, as stated in the conditions of theorem. Namely, these conditions are related to the realizability condition, which guarantees small training error near initialization. The next assumption formalizes these conditions.

**Assumption 2** (Realizability). *There exist non-increasing functions $g : \mathbb{R}_+ \to \mathbb{R}_+$ and $g_0 : \mathbb{R}_+ \to \mathbb{R}_+$ such that $\forall \epsilon > 0$, there exists model parameters $\widetilde{\boldsymbol{\theta}}_{(\epsilon)} \in \mathbb{R}^{H(dT+d^2)}$ for which: (i) the empirical loss over $n$ data samples satisfies $\widehat{L}(\widetilde{\boldsymbol{\theta}}_{(\varepsilon)}) \leq \varepsilon$, (ii) $\|\widetilde{\boldsymbol{\theta}}_{(\varepsilon)} - \widetilde{\boldsymbol{\theta}}^{(0)}\| \leq g_0(\varepsilon)$, and, (iii) $\|\widetilde{\boldsymbol{\theta}}_{(\varepsilon)}\|_{2,\infty} \leq g(\varepsilon)$.*

With this assumption, we can specialize the result of Thms. above to specific data settings; see Cor. 3 and 4 in App. B.5 and C.2. In the next section we will further show how the realizability assumption is satisfied.

### 4.3 Primitive conditions for checking realizability

Here, we introduce a set of more primitive and straightforward-to-check conditions on the data and initialization that ensure the realizability Assumption 2 holds.

**Definition 1** (Good initialization). *We say $\widetilde{\boldsymbol{\theta}}^{(0)} = \mathrm{concat}(\boldsymbol{\theta}_1^{(0)}, \ldots, \boldsymbol{\theta}_H^{(0)})$ is a* good *initialization with respect to training data $(\boldsymbol{X}_i, y_i)_{i \in [n]}$ provided the following three properties hold.*

*P1. **Parameter $L_{2,\infty}$-bound:** There exists parameter $B_2 \geq 1$ such that $\forall h \in [H]$ it holds $\|\boldsymbol{\theta}_h^{(0)}\|_2 \leq B_2$.*

*P2. **Model bound:** There exists parameter $B_\Phi \geq 1$ such that $\forall i \in [n]$ it holds $\big|\widetilde{\Phi}(\boldsymbol{X}_i; \widetilde{\boldsymbol{\theta}}^{(0)})\big| \leq B_\Phi$.*

*P3. **NTK separability:** There exists $\widetilde{\boldsymbol{\theta}}_\star \in \mathbb{R}^{H(dT+d^2)}$ and $\gamma > 0$ such that $\|\widetilde{\boldsymbol{\theta}}_\star\| = \sqrt{2}$ and $\forall i \in [n]$, it holds $y_i \big\langle \nabla \widetilde{\Phi}\big(\boldsymbol{X}_i; \widetilde{\boldsymbol{\theta}}^{(0)}\big), \widetilde{\boldsymbol{\theta}}_\star \big\rangle \geq \gamma$.*

Prop. 7 in the appendix shows that starting from a good initialization we can always find $\widetilde{\boldsymbol{\theta}}_{(\epsilon)}$ satisfying the realizability Assumption 2 provided large enough number of heads. Thus, given good initialization, we can immediately apply Theorems 1 and 2 to get the following concrete bounds.

**Corollary 2** (General bounds under good initialization). *Suppose* good *initialization $\widetilde{\boldsymbol{\theta}}^{(0)}$ and let*

$$\sqrt{H} \gtrsim d\,T^{1/2}\,R^5\,B_2^2\,\big(g_0(1/K)\big)^3, \qquad \text{where} \quad g_0\Big(\frac{1}{K}\Big) = \frac{2B_\Phi + \log(K)}{\gamma}.$$

*Further fix step-size $\eta \leq 1 \wedge 1/\rho(K) \wedge \frac{4B_\Phi^2}{\gamma^2 \log(1+e^{B_\Phi})}$ with $\rho(K) \gtrsim d^{3/2}\,T^{3/2}\,R^{13}\,g_0\big(\frac{1}{K}\big)^4$. Then, it holds that*

$$\widehat{L}(\widetilde{\boldsymbol{\theta}}^{(K)}) \leq \frac{2}{K} + \frac{5\,(2B_\Phi + \log(K))^2}{4\gamma^2\,\eta\,K}, \quad \text{and} \quad \mathbb{E}\big[L(\widetilde{\boldsymbol{\theta}}^{(K)}) - \widehat{L}(\widetilde{\boldsymbol{\theta}}^{(K)})\big] \leq \frac{17\,(2B_\Phi + \log(K))^2}{\gamma^2\,\eta\,n}.$$

Consider training loss after $K$ GD steps: Assuming $B_\Phi = \widetilde{\mathcal{O}}_K(1)$ and $\gamma = \mathcal{O}_K(1)$, then choosing $\eta = \widetilde{\mathcal{O}}_K(1)$, the corollary guarantees train loss is $\widetilde{\mathcal{O}}_K(\frac{1}{K})$ provided polylogarithmic number of heads $H = \Omega(\log^6(K))$. Moreover, after $K \approx n$ GD steps the expected test loss is $\mathcal{O}(1/n)$.

**Remark 1.** *The last two conditions (P2 and P3) for* `good` *initialization are similar to the conditions needed in (Taheri & Thrampoulidis, 2023; Ji & Telgarsky, 2020; Nitanda et al., 2019) for analysis of two-layer MLPs. Compared to (Ji & Telgarsky, 2020; Nitanda et al., 2019) which assume random Gaussian initialization $\widetilde{\boldsymbol{\theta}}^{(0)}$, and similar to (Taheri & Thrampoulidis, 2023) the NTK separability assumption (P3) can potentially accommodate deterministic $\widetilde{\boldsymbol{\theta}}^{(0)}$. Condition (P1) appears because we allow training both layers of the model. Specifically the $L_{2,\infty}$ norm originates from the Hessian bounds in Corollary 1.*

## 5 Application to tokenized-mixture model

We now demonstrate through an example how our results apply to specific data models.

**Data model: An example.** Consider $M+2$ distinct patterns $\{\boldsymbol{\mu}_+, \boldsymbol{\mu}_-, \boldsymbol{\nu}_1, \boldsymbol{\nu}_2, ..., \boldsymbol{\nu}_M\}$, where discriminative patterns $\boldsymbol{\mu}_\pm$ correspond to labels $y = \pm 1$. The tokens are split into (i) a label-relevant set ($\mathcal{R}$) and (ii) a label-irrelevant set ($\mathcal{R}^c \coloneqq [T]\backslash\mathcal{R}$). Conditioned on the label and $\mathcal{R}$, the tokens $\boldsymbol{x}_t, t \in [T]$ are IID as follows

$$\boldsymbol{x}_t|y = \begin{cases} \boldsymbol{\mu}_y & , t \in \mathcal{R} \\ \boldsymbol{\nu}_{j_t} + \boldsymbol{z}_t & , j_t \sim \text{Unif}(1, ..., M) \text{ and } t \in \mathcal{R}^c, \end{cases} \tag{DM1}$$

where $\boldsymbol{z}_t$ are noise vectors. Let $\mathcal{D}$ denote the joint distribution induced by the described $(\mathbf{X}, y)$ pairs.

**Assumption 3.** *The labels are equi-probable and we further assume the following:*

• ***Orthogonal, equal-energy means:*** *All patterns are orthogonal to each other, i.e. $\boldsymbol{\mu}_+ \perp \boldsymbol{\mu}_- \perp \boldsymbol{\nu}_\ell \perp \boldsymbol{\nu}_{\ell'}, \forall \ell, \ell' \in [M]$. Also, for all $y \in \{\pm 1\}, \ell \in [M]$ that $\|\boldsymbol{\mu}_y\| = \|\boldsymbol{\nu}_\ell\| = S$, where $S$ denotes the signal strength.*

• ***Sparsity level:*** *The number of label-relevant tokens is $|\mathcal{R}| = \zeta T$; for sparsity level $\zeta \in (0, 1)$.*

• ***Noise distribution:*** *The noise tokens $\boldsymbol{z}_t$ are sampled from a distribution $\mathcal{D}_z$, such that it holds almost surely for $\boldsymbol{z}_t \sim \mathcal{D}_z$ that $|\langle \boldsymbol{z}_t, \boldsymbol{\mu}_y \rangle| \le Z_\mu, y \in \{\pm 1\}$ and $|\langle \boldsymbol{z}_t, \boldsymbol{\nu}_\ell \rangle| \le Z_\nu/M, \forall \ell \in [M]$. Moreover, $\|\boldsymbol{z}_t\| \le Z$. Overall, Assumption 1 is satisfied with $R = \sqrt{S^2 + Z^2 + 2Z_\nu/M}$.*

The above assumptions can be relaxed, but without contributing new insights. We have chosen to present a model that is representative and transparent in its analysis.

### 5.1 Finding a `good` initialization

To apply the general Corollary 2 to the specific data model DM1, it suffices to find `good` initialization. While we cannot directly show that $\widetilde{\boldsymbol{\theta}}^{(0)} = 0$ is `good`, we can show this is the case for first step of gradient descent $\widetilde{\boldsymbol{\theta}}^{(1)}$. Thus, we consider training in two phases as follows.

**First phase: One step of GD as initialization.** We use $n_1$ training samples to update the model parameters by running one-step of gradient descent starting from zero initialization. Specifically,

$$(\boldsymbol{U}_h^{(1)}, \boldsymbol{W}_h^{(1)}) = \boldsymbol{\theta}_h^{(1)} = \boldsymbol{\theta}_h^{(0)} - \alpha_h \sqrt{H} \cdot \nabla_{\boldsymbol{\theta}_h} \widehat{L}_{n_1}(\boldsymbol{\theta}_h^{(0)}), \text{ where } \quad \boldsymbol{\theta}_h^{(0)} = \mathbf{0} \ \forall \ h \in [H].$$

Here, $\alpha_h$ denotes the step-size for head $h \in [H]$ and the scaling by $\sqrt{H}$ guarantees the update of each head is $\mathcal{O}(1)$. The lemma below shows that at the end of this phase, we have $\|\boldsymbol{U}_h^{(1)} - \frac{\zeta \alpha_h}{2} \mathbb{1}_T \boldsymbol{u}_\star^\top\|_F = O(1/\sqrt{n_1})$, where $\boldsymbol{u}_\star$ is the oracle classifier $\boldsymbol{u}_\star = \boldsymbol{\mu}_+ - \boldsymbol{\mu}_-$. On the other hand, the attention weight-matrix does *not* get updated; the interesting aspect of the training lies in the second phase, which involves updating $\boldsymbol{W}$.

**Lemma 2** (First phase). *After the first-gradient step as described above, we have $\boldsymbol{U}_h^{(1)} = \alpha_h \mathbb{1}_T \left( \frac{\zeta}{2} \boldsymbol{u}_\star^\top + \boldsymbol{p}^\top \right)$ and $\boldsymbol{W}_h^{(1)} = \mathbf{0}$. where with probability at least $1 - \delta \in (0, 1)$ over the randomness of labels there exists positive universal constant $C > 0$ such that*

$$\|\boldsymbol{p}\| \le C \left( 2S + Z \right) \left( \sqrt{\frac{d}{n_1}} + \sqrt{\frac{\log(1/\delta)}{n_1}} \right) =: P. \tag{6}$$

**Second phase: GD with constant step size.** During the second phase, $K$ gradient steps are performed on $n$ new samples (distinct from those used in the first phase). Concretely, $\widetilde{\boldsymbol{\theta}}^{(k+1)} = \widetilde{\boldsymbol{\theta}}^{(k)} - \eta \cdot \nabla_{\widetilde{\boldsymbol{\theta}}} \widehat{L}_n(\widetilde{\boldsymbol{\theta}}^{(k)}), \quad k = 1, \ldots, K$, with $\widetilde{\boldsymbol{\theta}}^{(1)} = \text{concat}\left(\{\boldsymbol{\theta}_h^{(1)}\}_{h \in [H]}\right)$ the step obtained by the first-phase update and $\eta$ the step-size of the second phase. In order to analyze the second phase, during which both $\widetilde{\boldsymbol{W}}$ and $\widetilde{\boldsymbol{U}}$ get updated, we employ the general results of Section 4. To do so, we show that $\widetilde{\boldsymbol{\theta}}^{(1)}$ serves as good initialization as per Definition 1.

**Proposition 2.** *Consider the first-phase iterate $\{\boldsymbol{\theta}_h^{(1)}\}_{h \in [H]}$ and condition on the event $\|\boldsymbol{p}\| \leq P$ (depending only on the data randomness in the first phase) of Lemma 2. Suppose the step-size of the first phase is chosen IID $\alpha_h \sim \text{Unif}(\pm 1), h \in [H]$. Then, the initialization $\widetilde{\boldsymbol{\theta}}^{(1)} = \text{concat}\left(\boldsymbol{\theta}_1^{(1)}, \ldots, \boldsymbol{\theta}_H^{(1)}\right)$ is good with respect to data sampled from DM1 and satisfying Assumption 3. Specifically, the three desired properties hold as follows.*
* *Almost surely, **P1** holds with $B_2 = \sqrt{T}(S + P)$.*
* *With probability $1 - \delta \in (0, 1)$, **P2** holds with $B_\Phi = TR(S + P)\sqrt{2\log(n/\delta)}$.*
* *Suppose $\sqrt{H} \gtrsim \frac{R^4 T(S+P)}{\gamma_\star} \cdot \sqrt{2\log(n/\delta)}$. Then, with probability $1 - \delta \in (0, 1)$, **P3** holds with $\gamma = \gamma_\star/2$ where*

$$\gamma_\star := \frac{T(1-\zeta)\zeta\left(\zeta S^4 - 7\bar{Z}S^2 - 12\bar{Z}^2 - 16\frac{\bar{Z}^3}{S^2}\right)}{4\sqrt{2(M+1)}} - PT^{5/2}(S+Z)^3 + \frac{S\sqrt{T}\left(\zeta - 2(1-\zeta)\frac{Z_\mu}{S^2}\right)}{\sqrt{2}}, \tag{7}$$

*and $\bar{Z} := Z_\mu \vee Z_\nu$. The randomness is with respect to the sampling of $\alpha_h, h \in [H]$.*

The parameter $\gamma_\star$ in (7) represents the NTK margin of the model at initialization $\widetilde{\boldsymbol{\theta}}^{(1)}$. By Corollary 2, larger margin translates to better train/generalization bounds and smaller requirements on the number of heads. For a concrete example, suppose $T \vee M = \mathcal{O}(1)$ and $Z \vee \bar{Z} = \mathcal{O}(S)$. Then, provided first-phase sample size $n_1 \gtrsim S^2 d$ so that $P = \mathcal{O}(1)$, it holds $\gamma_\star = \gamma_{\text{lin}} + \Omega(\zeta^2(1-\zeta)S^4)$, where $\gamma_{\text{lin}} = \Omega(\zeta S)$ is the margin of a linear model for the same dataset (see App. F). Overall, applying Cor. 2 for $K = n$ and a polylogarithmic $\text{polylog}(n)$ number of heads leads to $\tilde{\mathcal{O}}\left(\frac{1}{\eta\gamma_\star^2 n}\right)$ train loss and expected generalization gap.

## 5.2 Proof sketch of P3: NTK separability

It is instructive to see how **P3** follows as it sheds light on the choice of an appropriate target parameter $\widetilde{\boldsymbol{\theta}}$ as per Thms. 1 and 2. We choose

$$\boldsymbol{W}_\star = \boldsymbol{\mu}_+\boldsymbol{\mu}_+^\top + \boldsymbol{\mu}_-\boldsymbol{\mu}_-^\top + \sum_{\ell \in [M]} \boldsymbol{\nu}_\ell(\boldsymbol{\mu}_+ + \boldsymbol{\mu}_-)^\top \quad \text{and} \quad \boldsymbol{U}_\star = \mathbb{1}_T\boldsymbol{u}_\star^\top = \mathbb{1}_T(\boldsymbol{\mu}_+ - \boldsymbol{\mu}_-)^\top,$$

and normalize parameters such that $\boldsymbol{\theta}_\star := (\overline{\boldsymbol{U}}_\star = \frac{1}{\|\boldsymbol{U}_\star\|_F}\boldsymbol{U}_\star, \text{sign}(\alpha)\overline{\boldsymbol{W}}_\star = \text{sign}(\alpha)\frac{1}{\|\boldsymbol{W}_\star\|_F}\boldsymbol{W}_\star)$. It is easy to see that $\boldsymbol{U}_\star$ is the optimal classifier for the label-relevant tokens. To gain intuition on the choice of $\boldsymbol{W}_\star$, note that $\boldsymbol{W}_\star = \boldsymbol{W}_{K,\star}\boldsymbol{W}_{Q,\star}^\top$, with key-query matrices chosen as $\boldsymbol{W}_{K,\star} = [\boldsymbol{\mu}_+ \quad \boldsymbol{\mu}_- \quad \boldsymbol{\nu}_1 \quad \cdots \quad \boldsymbol{\nu}_M] \in \mathbb{R}^{d \times (M+2)}$ and $\boldsymbol{W}_{Q,\star} = [\boldsymbol{\mu}_+ \quad \boldsymbol{\mu}_- \quad \boldsymbol{\mu}_+ + \boldsymbol{\mu}_- \quad \cdots \quad \boldsymbol{\mu}_+ + \boldsymbol{\mu}_-] \in \mathbb{R}^{d \times (M+2)}$. With these choices, the relevance scores (aka softmax logits) of relevant tokens turn out to be strictly larger compared to the irrelevant tokens. Concretely, we show in App. D.2.3 that the $t$-th row $\boldsymbol{r}_t(\boldsymbol{X}; \boldsymbol{W}_\star) = \boldsymbol{X}\boldsymbol{W}_\star^\top\boldsymbol{x}_t$ of the softmax-logit matrix satisfies the following:

$$\forall t : [\boldsymbol{r}_t]_{t'} = \begin{cases} \mathcal{O}(S^4) & , t' \in \mathcal{R}, \\ \mathcal{O}(S^2) & , t' \in \mathcal{R}^c. \end{cases} \tag{8}$$

Thus, under this parameter choice, softmax can attend to label-relevant tokens and supresses noisy irrelevant tokens. In turn, this increases the signal-to-noise ratio for classification using $\boldsymbol{U}_\star$.

We now show how to compute $\mathbb{E}_{\boldsymbol{\theta}^{(1)}} y\langle\nabla_{\boldsymbol{\theta}}\Phi(\boldsymbol{X}; \boldsymbol{\theta}^{(1)}), \boldsymbol{\theta}_\star\rangle$ for a single head. Recall $\boldsymbol{\theta}_\star$ consists of $\overline{\boldsymbol{U}}_\star, \overline{\boldsymbol{W}}_\star$. First, since $\boldsymbol{W}^{(1)} = \boldsymbol{0}$, using Assumption 3, a simple calculation shows $y\langle\nabla_{\boldsymbol{U}}\Phi(\boldsymbol{X}; \boldsymbol{\theta}^{(1)}), \overline{\boldsymbol{U}}_\star\rangle \geq \frac{S\sqrt{T}}{\sqrt{2}}\left(\zeta - 2(1-\zeta)\frac{Z_\mu}{S^2}\right)$. Second, to compute $\mathbb{E}_{\alpha \sim \text{Unif}(\pm 1)} y\langle\nabla_{\boldsymbol{W}}\Phi(\boldsymbol{X}; \boldsymbol{\theta}^{(1)}), \text{sign}(\alpha)\overline{\boldsymbol{W}}_\star\rangle$ it follows from Lemma 1 that

$$\nabla_{\boldsymbol{W}}\Phi(\boldsymbol{X}; \boldsymbol{\theta}^{(1)}) = \frac{\alpha\zeta}{2}\sum_{t \in [T]} \boldsymbol{x}_t\boldsymbol{u}_\star^\top\boldsymbol{X}^\top\boldsymbol{\varphi}'(\boldsymbol{0})\boldsymbol{X} + \alpha\sum_{t \in [T]} \boldsymbol{x}_t\boldsymbol{p}^\top\boldsymbol{X}^\top\boldsymbol{\varphi}'(\boldsymbol{0})\boldsymbol{X}.$$

Note the first term is dominant here since the second term can be controlled by making $\|\boldsymbol{p}\|_2$ small as per Lemma 2. Thus, ignoring here the second term (see Appendix D.2.3 for full calculation) $y\langle\nabla_{\boldsymbol{W}}\Phi(\boldsymbol{X};\boldsymbol{\theta}^{(1)}),\boldsymbol{W}_\star\rangle$ is governed by the following term: $\frac{\alpha\zeta}{2}\sum_{t\in[T]}y\,\boldsymbol{u}_\star^\top\boldsymbol{X}^\top\boldsymbol{\varphi}'(\boldsymbol{0})\boldsymbol{X}\boldsymbol{W}_\star^\top\boldsymbol{x}_t = \frac{\alpha\zeta}{2}\sum_{t\in[T]}y\,\boldsymbol{u}_\star^\top\boldsymbol{X}^\top\boldsymbol{\varphi}'(\boldsymbol{0})\,\boldsymbol{r}_t$. Note that $\boldsymbol{\varphi}'(\boldsymbol{0}) = \boldsymbol{I} - \frac{1}{T}\mathbb{1}_T\mathbb{1}_T^\top$. To simplify the exposition here, let us focus on the identity component and leave treatment of the the rank-one term to the detailed proof. The corresponding term then becomes

$$\frac{\alpha\zeta}{2}\sum_{t\in[T]}\sum_{t'\in[T]}\underbrace{\left(y\,\boldsymbol{u}_\star^\top\boldsymbol{x}_{t'}\right)}_{\text{class. logits}}\cdot\underbrace{\left([\boldsymbol{r}_t]_{t'}\right)}_{\text{softmax logits}},$$

which involves for each output token $t$, the sum of products over all tokens $t' \in [T]$ of softmax logits (i.e. relevant scores $[\boldsymbol{r}_t]_{t'}$) and corresponding classification logits (i.e. $y\,\boldsymbol{u}_\star^\top\boldsymbol{x}_{t'}$). Note that by choice of $\boldsymbol{u}_\star$ and $\boldsymbol{W}_\star$, both the classification and softmax logits are large from label-relevant tokens, while being small for noise tokens. Intuitively, this allows for a positive margin $\gamma_\star$ as stated in Proposition 2. We defer the detailed calculations to Appendix D.2.3.

In the appendix, we also detail how to yield the computation for the MHA, which builds on the calculations for the single-head attention model above. In short, we simply choose multi-head parameter $\widetilde{\boldsymbol{\theta}}_\star$ as $\widetilde{\boldsymbol{\theta}}_\star = \frac{1}{\sqrt{H}}\operatorname{concat}\left(\boldsymbol{\theta}_\star(\boldsymbol{\theta}_1^{(1)}),\ldots,\boldsymbol{\theta}_\star(\boldsymbol{\theta}_H^{(1)})\right)$. This guarantees that $\|\widetilde{\boldsymbol{\theta}}_\star\| = \sqrt{2}$ and maintains the multi-head NTK margin be at least $\gamma_\star$ in expectation. To complete the proof, it remains to get a high-probability version of this bound. To do this, notice that $\boldsymbol{\theta}_h^{(1)}$ are IID, hence we can apply Hoeffding's inequality, which finally gives the desired bound on the NTK margin provided sufficient number of heads $H$, which controls the degree of concentration when applying Hoeffding's inequality. See Lemmas 14 and 15 for details.

## 5.3  Is the NTK margin optimal?

Below, we discuss the optimality of the NTK margin $\gamma_\star$. First, define set of parameters $\boldsymbol{\theta}_{\text{opt}} := (\boldsymbol{U}_{\text{opt}},\boldsymbol{W}_{\text{opt}})$:

$$\boldsymbol{U}_{\text{opt}} := \frac{1}{S\sqrt{2T}}\boldsymbol{U}_\star \qquad\text{and}\qquad \boldsymbol{W}_{\text{opt}} := \frac{1}{S^2\sqrt{2(M+1)}}\boldsymbol{W}_\star\,, \tag{9}$$

normalized so that $\|\boldsymbol{\theta}_{\text{opt}}\|_F = \sqrt{2}$. Recall here the definitions of $\boldsymbol{U}_\star,\boldsymbol{W}_\star$ in the section above. As we already explained above, this choice of parameters guarantees that relevant tokens are assigned larger relevance and classification scores compared to irrelevant ones. Specifically about $\boldsymbol{W}_\star$, we saw in Eq. (8) that it ensures a gap of $\mathcal{O}(S^2)$ between relevance scores of label-relevant and label-irrelevant tokens. Thanks to this gap, it is possible for softmax to fully attend to the label-relevant tokens by saturating the softmax. To do this, it suffices to scale-up $\boldsymbol{W}_\star$ by an amount $\propto 1/S^2$. This is formalized in the proposition below.

**Proposition 3** (Attention expressivity for tokenized mixture model)**.** *Consider single-head attention model. Suppose the noise level is such that $Z_\mu = Z_\nu \leq S^2/8$. For any $\epsilon > 0$, consider $\Gamma_\epsilon$ satisfying $\Gamma_\epsilon \geq \frac{8\sqrt{2(M+1)}}{3S^2}\log\left(\frac{\zeta^{-1}-1}{\epsilon}\right)$. Then, the attention scores corresponding to weights $\Gamma_\epsilon\cdot\boldsymbol{W}_{opt}$ satisfy*

$$\forall t \in [T]: 0 \leq 1 - \sum_{t'\in\mathcal{R}}\boldsymbol{\varphi}_{t'}(\boldsymbol{x}_t^\top\Gamma_\epsilon\boldsymbol{W}_{opt}\boldsymbol{X}^T) = \sum_{t'\in\mathcal{R}^c}\boldsymbol{\varphi}_{t'}(\boldsymbol{x}_t^\top\Gamma_\epsilon\boldsymbol{W}_{opt}\boldsymbol{X}^T) \leq \epsilon. \tag{10}$$

*Thus, almost surely over data $(\boldsymbol{X},y)$ generated from data model DM1 the margin of single-head attention with parameters $(\boldsymbol{U}_{opt},\Gamma_\epsilon\cdot\boldsymbol{W}_{opt})$ satisfies*

$$y\Phi(\boldsymbol{X};\boldsymbol{U}_{opt},\Gamma_\epsilon\cdot\boldsymbol{W}_{opt}) \geq \gamma_{attn} := \gamma_{attn}(\epsilon) := \frac{\sqrt{T}}{\sqrt{2}S}\left(S^2(1-\epsilon) - 2\epsilon Z_\mu\right). \tag{11}$$

From Eq. (10), note that as $\epsilon \to 0$ and $\Gamma_\epsilon \to \infty$, the softmax map saturates, i.e. it approaches a hard-max map that attends only to the label-relevant tokens ($\mathcal{R}$) and suppress the rest ($\mathcal{R}^c$). As a consequence of this, Eq. (11) shows that the achieved margin approaches $\gamma_{\text{attn}} := S\sqrt{T}/\sqrt{2}$. Note this is independent of the sparsity level $\zeta$. In particular, $\gamma_{\text{attn}} \geq \gamma_\star \geq \gamma_{\text{lin}}$ and the gap increases with decreasing sparsity. See appendix for experiments and discussion regarding the margin achieved by GD for data model DM1.

Following Proposition 3, a natural question arises: Is it possible to choose "good" parameters $\widetilde{\boldsymbol{\theta}} = (\widetilde{\boldsymbol{U}}, \widetilde{\boldsymbol{W}})$ based on the set of optimal parameters $\boldsymbol{\theta}_{\text{opt}}$? This would then yield train-loss and expected generalization-gap bounds $\tilde{\mathcal{O}}\left(1/(\eta \gamma_{\text{attn}}^2 n)\right)$ after $\Theta(n)$ steps of GD starting at $\widetilde{\boldsymbol{\theta}}^{(0)} = \boldsymbol{0}$. To investigate this question, define the following parameters for each head, aligning with the aforementioned "good" directions of Proposition 3:

$$\boldsymbol{U}_h := \frac{\log(n)}{\gamma_{\text{attn}}} \frac{1}{H^{1/2}} \boldsymbol{U}_{\text{opt}}, \quad \boldsymbol{W}_h := \frac{C}{H^p} \boldsymbol{W}_{\text{opt}},$$

for some $C > 0$, $p > 0$, and $\forall h \in [H]$. To yield the margin $\gamma_{\text{attn}}$ of (10), we need that each $\boldsymbol{W}_h$ has norm at least $\Gamma_\epsilon \propto 1/S^2$. Thus, we need $\|\boldsymbol{W}_h\| \gtrsim \frac{1}{S^2} \implies S^2 \gtrsim \frac{1}{C} \cdot H^p$. Now, in order to apply Thms. 1 and 2, the requirement on the number of heads $H$ in terms of distance of $\widetilde{\boldsymbol{\theta}}$ to $\widetilde{\boldsymbol{\theta}}^{(0)} = \boldsymbol{0}$ yields the following condition:

$$H^{1/2} \gtrsim S^5 \|\widetilde{\boldsymbol{\theta}}\|^3. \tag{12}$$

Note that $\|\widetilde{\boldsymbol{U}}\| = \frac{\log(n)}{\gamma_{\text{attn}}} \approx \frac{\log(n)}{S}$, $\|\widetilde{\boldsymbol{W}}\| = C \cdot H^{1/2-p}$. Hence, in computing $\|\widetilde{\boldsymbol{\theta}}\|$, we distinguish two cases. First, assume that $\|\widetilde{\boldsymbol{W}}\| \geq \|\widetilde{\boldsymbol{U}}\|$ which implies that $S \gtrsim \frac{\log(n)}{C} \cdot H^{p-1/2}$ and $\|\widetilde{\boldsymbol{\theta}}\| \gtrsim \|\widetilde{\boldsymbol{W}}\| \vee \|\widetilde{\boldsymbol{U}}\| = C \cdot H^{1/2-p}$. Since

$$S \gtrsim \frac{1}{C^{1/2}} \cdot H^{p/2} \vee \frac{\log(n)}{C} \cdot H^{p-1/2},$$

by using Eq. (12), we get the following conditions on $H$:

$$H^{1/2} \gtrsim S^5 \cdot C^3 \cdot H^{3/2-3p} \gtrsim C^{1/2} \cdot H^{3/2-p/2} \vee \frac{\log^5(n)}{C^2} \cdot H^{2p-1} \implies H^{p-2} \gtrsim C \quad \text{and} \quad H^{p-1/4} \lesssim \frac{C}{\log^{5/2}(n)}.$$

Combining these two gives $C \lesssim \frac{C}{\log^{5/2}(n)} \implies \log(n) \lesssim 1$, a contradiction since $n > 1$. Thus, there are no possible choices for $p$ and $C$ that satisfy both conditions. The case $\|\widetilde{\boldsymbol{W}}\| \leq \|\widetilde{\boldsymbol{U}}\|$ can be treated similarly leading to the same conclusion; thus, is omitted for brevity.

Intuitively, this contradiction arises because of the large $\|\boldsymbol{W}_h\|$ requirement to achieve margin $\gamma_{\text{attn}}$. Finally, one can ask if it is possible to resolve the contradiction by changing the scaling of normalization with respect to $H$ in the MHA model Eq. (2), from $1/H^{1/2}$ to $1/H^c$ for $c > 0$. It can be shown via the same argument that no such value of $c$ exists for which $\widetilde{\boldsymbol{\theta}}$ constructed above satisfies the overparameterization requirement $H^c \gtrsim S^5 \|\widetilde{\boldsymbol{\theta}}\|^3$. We thus conclude that the construction of weights in Proposition 3 does not yield a target parameter that simultaneously achieves low empirical loss and allows choosing $H$ large enough as per (3). This triggers interesting questions for future research: Does GD converge to weights attaining margin $\gamma_{\text{attn}}$ as in Proposition 3? If so, under what conditions on initialization? See also the remarks in Section 7.

# 6 Proof Sketch of Section 4

Throughout this section we drop the $\sim$ in $\widetilde{\boldsymbol{\theta}}$ and $\widetilde{\Phi}(\boldsymbol{X}_i; \widetilde{\boldsymbol{\theta}})$ as everything refers to the full model. Moreover, $\widetilde{\boldsymbol{\theta}}^{(K)}$ and $\widetilde{\boldsymbol{\theta}}^{(0)}$ are denoted by $\boldsymbol{\theta}_K$ and $\boldsymbol{\theta}_0$. Refer to Figure 1 in the App. for a summary of the sketch.

## 6.1 Training analysis

The proof begins by showing step-wise descent for any iteration $k \geq 0$ of GD (see Lemma 7), where step-size at each iteration $\eta_k \leq \frac{1}{\rho_k}$ depends on the objective's local smoothness parameters $\rho_k = \beta_2(\boldsymbol{\theta}_k) \vee \beta_2(\boldsymbol{\theta}_{k+1})$:

$$\widehat{L}(\boldsymbol{\theta}_{k+1}) \leq \widehat{L}(\boldsymbol{\theta}_k) - \frac{\eta_k}{2} \left\|\nabla \widehat{L}(\boldsymbol{\theta}_k)\right\|^2. \tag{13}$$

Now, using Taylor's theorem we can link $\widehat{L}(\boldsymbol{\theta}_k)$ to $\widehat{L}(\boldsymbol{\theta})$ for any $\boldsymbol{\theta}$ as follows:

$$\widehat{L}(\boldsymbol{\theta}) \geq \widehat{L}(\boldsymbol{\theta}_k) + \langle \nabla \widehat{L}(\boldsymbol{\theta}_k), \boldsymbol{\theta} - \boldsymbol{\theta}_k \rangle + \frac{1}{2} \min_{\boldsymbol{\theta}_{k_\alpha}} \lambda_{\min}\left(\nabla^2 \widehat{L}(\boldsymbol{\theta}_{k_\alpha})\right) \|\boldsymbol{\theta} - \boldsymbol{\theta}_k\|^2, \tag{14}$$

where $\boldsymbol{\theta}_{k_\alpha} := \alpha\boldsymbol{\theta}_k + (1-\alpha)\boldsymbol{\theta}$, $\alpha \in [0,1]$. We can plug this into (13) to relate the loss at iterates $\boldsymbol{\theta}_k$ and $\boldsymbol{\theta}_{k+1}$. To continue, we need to lower bound $\min_{\boldsymbol{\theta}_{k_\alpha}} \lambda_{\min}(\nabla^2\widehat{L}(\boldsymbol{\theta}_{k_\alpha}))$. For this, we use the following property of the loss objective from Corollary 1: $\forall\boldsymbol{\theta}: \ \lambda_{\min}(\nabla^2\widehat{L}(\boldsymbol{\theta})) \geq -\kappa(\boldsymbol{\theta}) \cdot \widehat{L}(\boldsymbol{\theta})$, where $\kappa(\boldsymbol{\theta}) := \frac{\beta_3(\boldsymbol{\theta})}{\sqrt{H}}$. Note from the definition of $\beta_3(\cdot)$ that $\forall\boldsymbol{\theta}_1,\boldsymbol{\theta}_2: \ \max_{\boldsymbol{\theta}\in[\boldsymbol{\theta}_1,\boldsymbol{\theta}_2]} \beta_3(\boldsymbol{\theta}) = \beta_3(\boldsymbol{\theta}_1) \vee \beta_3(\boldsymbol{\theta}_2)$. Thus, the above property of the loss implies the following *local self-bounded weak convexity* property on the line $[\boldsymbol{\theta}_1,\boldsymbol{\theta}_2]$ for arbitrary points $\boldsymbol{\theta}_1,\boldsymbol{\theta}_2$:

$$\forall\boldsymbol{\theta}_1,\boldsymbol{\theta}_2: \ \min_{\boldsymbol{\theta}\in[\boldsymbol{\theta}_1,\boldsymbol{\theta}_2]} \lambda_{\min}(\nabla^2\widehat{L}(\boldsymbol{\theta})) \geq -\frac{\beta_3(\boldsymbol{\theta}_1) \vee \beta_3(\boldsymbol{\theta}_2)}{\sqrt{H}} \cdot \max_{\boldsymbol{\theta}\in[\boldsymbol{\theta}_1,\boldsymbol{\theta}_2]} \widehat{L}(\boldsymbol{\theta}). \tag{15}$$

Therefore, using Eq. (15) in Eq. (14), we can get:

$$\widehat{L}(\boldsymbol{\theta}) \geq \widehat{L}(\boldsymbol{\theta}_k) + \langle\nabla\widehat{L}(\boldsymbol{\theta}_k),\boldsymbol{\theta}-\boldsymbol{\theta}_k\rangle - \frac{1}{2}\frac{\beta_3(\boldsymbol{\theta}_1) \vee \beta_3(\boldsymbol{\theta}_2)}{\sqrt{H}} \cdot \max_{\alpha\in[0,1]} \widehat{L}(\boldsymbol{\theta}_{k_\alpha}) \|\boldsymbol{\theta}-\boldsymbol{\theta}_k\|^2. \tag{16}$$

To apply the Descent Lemma in (13), we need to fix a step-size such that satisfies the condition of the Lemma at each iteration $\eta \leq \eta_k$ for all $k < K$. Then, combining with Eq. (16) and applying standard telescope summation, we arrive at the following:

$$\frac{1}{K}\sum_{k=1}^{K}\widehat{L}(\boldsymbol{\theta}_k) \leq \widehat{L}(\boldsymbol{\theta}) + \frac{\|\boldsymbol{\theta}-\boldsymbol{\theta}_0\|^2}{2\eta K} + \frac{1}{2K}\sum_{k=0}^{K-1}\frac{\beta_3(\boldsymbol{\theta}) \vee \beta_3(\boldsymbol{\theta}_k)}{\sqrt{H}} \cdot \max_{\alpha\in[0,1]} \widehat{L}(\boldsymbol{\theta}_{k_\alpha}) \|\boldsymbol{\theta}-\boldsymbol{\theta}_k\|^2. \tag{17}$$

Next, we use the following generalized local quasi-convexity (GLQC) of the loss function.

**Proposition 4** (GLQC property: Slight variation of Prop. 8 of Taheri & Thrampoulidis (2023)). *Let $\boldsymbol{\theta}_1$ and $\boldsymbol{\theta}_2$ be two points that are sufficiently close to each other, such that*

$$2\left(\beta_3(\boldsymbol{\theta}_1) \vee \beta_3(\boldsymbol{\theta}_2)\right)\|\boldsymbol{\theta}_1-\boldsymbol{\theta}_2\|^2 \leq \sqrt{H}. \tag{18}$$

*Then,* $\max_{\boldsymbol{\theta}\in[\boldsymbol{\theta}_1,\boldsymbol{\theta}_2]} \widehat{L}(\boldsymbol{\theta}) \leq \frac{4}{3}\left(\widehat{L}(\boldsymbol{\theta}_1) \vee \widehat{L}(\boldsymbol{\theta}_2)\right).$

Using Proposition 4 in Eq. (17) and assuming sufficiently large heads $H$ such that $\sqrt{H} \geq 2\left(\beta_3(\boldsymbol{\theta}) \vee \beta_3(\boldsymbol{\theta}_k)\right)\|\boldsymbol{\theta}-\boldsymbol{\theta}_k\|^2$, we can get the advertised regret bound in (4).

In order to remove the dependence of $H$ on iteration $k$, by an induction argument we can show bounded iterates-norm i.e. $\|\boldsymbol{\theta}_k-\boldsymbol{\theta}\| \leq 3\|\boldsymbol{\theta}-\boldsymbol{\theta}_0\|$ (see Lemma 10). Using this and the definition of $\beta_3(\cdot)$ we can control $\beta_3(\boldsymbol{\theta}) \vee \beta_3(\boldsymbol{\theta}_k)$ as $\left(\beta_3(\boldsymbol{\theta}) \vee \beta_3(\boldsymbol{\theta}_k)\right) \lesssim \|\boldsymbol{\theta}-\boldsymbol{\theta}_0\| + \|\boldsymbol{\theta}\|_{2,\infty}$ to get the desired requirement of heads $\sqrt{H} \gtrsim \|\boldsymbol{\theta}\|_{2,\infty}\|\boldsymbol{\theta}-\boldsymbol{\theta}_0\|^3$ stated in Eq. (3).

The remaining piece to guarantee descent at each step is establishing a $\rho(\boldsymbol{\theta})$ such that $\rho_k \leq \rho(\boldsymbol{\theta})$ for all $k < K$. To do this, we recall that $\rho_k = \beta_2(\boldsymbol{\theta}_k) \vee \beta_2(\boldsymbol{\theta}_{k+1})$. By definition of $\beta_2(\cdot)$ in Corollary 1, we can control $\beta_2(\boldsymbol{\theta}_k) \vee \beta_2(\boldsymbol{\theta}_{k+1})$ with controlling $\|\boldsymbol{\theta}_k\|_{2,\infty} \vee \|\boldsymbol{\theta}_{k+1}\|_{2,\infty}$ as $\left(\|\boldsymbol{\theta}_k\|_{2,\infty} \vee \|\boldsymbol{\theta}_{k+1}\|_{2,\infty}\right) \lesssim \|\boldsymbol{\theta}-\boldsymbol{\theta}_k\| + \|\boldsymbol{\theta}\|_{2,\infty} + 1$. Using iterates-norm bound and setting $\rho(\boldsymbol{\theta}) = \left(\frac{2\sqrt{T d}R^3}{\sqrt{H}} + \frac{T R^2}{4}\right)\alpha(\boldsymbol{\theta})^2$ with $\alpha(\boldsymbol{\theta}) := 3\sqrt{d}R^2\left[3\sqrt{T}R^3\left(3\|\boldsymbol{\theta}-\boldsymbol{\theta}_0\| + \|\boldsymbol{\theta}\|_{2,\infty}\right) + 2\sqrt{T}R\right]$, satisfies the desired condition for the Descent Lemma completing the proof.

## 6.2 Generalization analysis

In order to bound the expected generalization gap, we leverage the algorithmic stability framework. To begin, consider the leave-one-out (loo) training loss $\widehat{L}^{-i}(\boldsymbol{\theta}) := \frac{1}{n}\sum_{j\neq i}\ell_j(\boldsymbol{\theta})$ for $i \in [n]$, where $\ell_j(\boldsymbol{\theta}) := \ell(y_j\Phi(\boldsymbol{X}_j;\boldsymbol{\theta}))$ denotes the $j$-th sample loss. With these, define the loo model updates of GD on the loo loss for $\eta > 0$:

$$\boldsymbol{\theta}_{k+1}^{-i} := \boldsymbol{\theta}_k^{-i} - \eta\nabla\widehat{L}^{-i}(\boldsymbol{\theta}_k^{-i}), \ k \geq 0, \qquad \boldsymbol{\theta}_0^{-i} = \boldsymbol{\theta}_0.$$

The following lemma relates expected generalization loss to average model stability for any $G$-Lipschitz loss.

**Lemma 3** (Lei & Ying (2020), Thm. 2). *For $G$-Lipschitz loss and for all iterates $K$, it holds that* $\mathbb{E}\left[L(\boldsymbol{\theta}_K) - \widehat{L}(\boldsymbol{\theta}_K)\right] \leq 2G \cdot \mathbb{E}\left[\frac{1}{n}\sum_{i=1}^{n}\|\boldsymbol{\theta}_K - \boldsymbol{\theta}_K^{-i}\|\right].$

To bound the average model-stability on the r.h.s of the lemma's inequality, we use GD expansiveness. Specifically applying (Taheri & Thrampoulidis, 2023, Lemma B.1.) to our setting, gives $\forall \boldsymbol{\theta}, \boldsymbol{\theta}'$:

$$\| (\boldsymbol{\theta} - \eta \nabla \widehat{L}(\boldsymbol{\theta})) - (\boldsymbol{\theta}' - \eta \nabla \widehat{L}(\boldsymbol{\theta}')) \| \le \max_{\alpha \in [0,1]} \left\{ \left( 1 + \frac{\eta \beta_3(\boldsymbol{\theta}_\alpha)}{\sqrt{H}} \widehat{L}(\boldsymbol{\theta}_\alpha) \right) \vee \eta \beta_2(\boldsymbol{\theta}_\alpha) \right\} \| \boldsymbol{\theta} - \boldsymbol{\theta}' \|, \tag{19}$$

where, $\boldsymbol{\theta}_\alpha = \alpha \boldsymbol{\theta} + (1-\alpha) \boldsymbol{\theta}'$, $\alpha \in [0,1]$. Using this and gradient self-boundedness from Corollary 1, we get:

$$\left\| \boldsymbol{\theta}_{k+1} - \boldsymbol{\theta}_{k+1}^{\neg i} \right\| \le \max_{\alpha \in [0,1]} \left\{ \left( 1 + \frac{\eta \beta_3(\boldsymbol{\theta}_{k_\alpha}^{\neg i})}{\sqrt{H}} \widehat{L}^{\neg i}(\boldsymbol{\theta}_{k_\alpha}^{\neg i}) \right) \vee \eta \beta_2(\boldsymbol{\theta}_{k_\alpha}^{\neg i}) \right\} \cdot \left\| \boldsymbol{\theta}_k - \boldsymbol{\theta}_k^{\neg i} \right\| + \frac{\eta \beta_1(\boldsymbol{\theta}_k)}{n} \ell_i(\boldsymbol{\theta}_k), \tag{20}$$

where $\boldsymbol{\theta}_{k_\alpha}^{\neg i} \coloneqq \alpha \boldsymbol{\theta}_k + (1-\alpha) \boldsymbol{\theta}_k^{\neg i}$ for $\alpha \in [0,1]$. Further using the bounded iterates-norm property from the training analysis, we can control $\beta_2(\boldsymbol{\theta}_{k_\alpha}^{\neg i}) \le \tilde{\beta}_2(\boldsymbol{\theta})$ and $\beta_3(\boldsymbol{\theta}_{k_\alpha}^{\neg i}) \le \tilde{\beta}_3(\boldsymbol{\theta})$ making them independent of $k$ (See Lemma 11 for the definitions of $\tilde{\beta}_2(\cdot), \tilde{\beta}_3(\cdot)$). In order to invoke the Descent Lemma, we set the step-size same as in the training analysis. Thus, (20) becomes:

$$\left\| \boldsymbol{\theta}_{k+1} - \boldsymbol{\theta}_{k+1}^{\neg i} \right\| \le \left( (1 + \frac{\eta \tilde{\beta}_3(\boldsymbol{\theta})}{\sqrt{H}}) \max_{\alpha \in [0,1]} \widehat{L}^{\neg i}(\boldsymbol{\theta}_{k_\alpha}^{\neg i}) \right) \left\| \boldsymbol{\theta}_k - \boldsymbol{\theta}_k^{\neg i} \right\| + \frac{\eta \beta_1(\boldsymbol{\theta}_k)}{n} \ell_i(\boldsymbol{\theta}_k) . \tag{21}$$

As in the training analysis, we can control the loo empirical loss $\widehat{L}^{\neg i}$ for any point on the line $[\boldsymbol{\theta}_k, \boldsymbol{\theta}_k^{\neg i}]$ of two sufficiently close points satisfying $\sqrt{H} \ge 2 \left( \beta_3(\boldsymbol{\theta}_k) \vee \beta_3(\boldsymbol{\theta}_k^{\neg i}) \right) \| \boldsymbol{\theta}_k - \boldsymbol{\theta}_k^{\neg i} \|^2$. Using Prop. 4, Eq. (21) becomes

$$\left\| \boldsymbol{\theta}_{k+1} - \boldsymbol{\theta}_{k+1}^{\neg i} \right\| \le (1 + \alpha_{k,i}) \left\| \boldsymbol{\theta}_k - \boldsymbol{\theta}_k^{\neg i} \right\| + \frac{\eta \tilde{\beta}_1(\boldsymbol{\theta})}{n} \ell_i(\boldsymbol{\theta}_k), \tag{22}$$

where $\alpha_{k,i} \coloneqq \frac{4\eta \tilde{\beta}_3(\boldsymbol{\theta})}{3\sqrt{H}} \left( \widehat{L}^{\neg i}(\boldsymbol{\theta}_k) + \widehat{L}^{\neg i}(\boldsymbol{\theta}_k^{\neg i}) \right)$ and $\beta_1(\boldsymbol{\theta}_k) \le \tilde{\beta}_1(\boldsymbol{\theta})$ similar to $\beta_2(\cdot), \beta_3(\cdot)$ using bounded iterates-norm. Unrolling the iterates in (22), summing over $i \in [n]$ and using training regret bounds, we have the following average model stability bound for any iterate $K$: $\frac{1}{n} \sum_{i=1}^n \left\| \boldsymbol{\theta}_K - \boldsymbol{\theta}_K^{\neg i} \right\| \le \frac{2\eta \tilde{\beta}_1(\boldsymbol{\theta})}{n} \left( 2 K \widehat{L}(\boldsymbol{\theta}) + \frac{9 \| \boldsymbol{\theta} - \boldsymbol{\theta}_0 \|^2}{4\eta} \right)$, Combining this with an application of Lemma 3 for our objective, which is $G \le \tilde{\beta}_1(\boldsymbol{\theta})$-Lipschitz from Corollary 1, and using $\eta \le \frac{1}{\rho(\boldsymbol{\theta})} \le \frac{1}{(\tilde{\beta}_1(\boldsymbol{\theta}))^2}$, we get the desired generalization gap stated in Thm. 2.

# 7 Concluding remarks

We studied convergence and generalization of GD for training a multi-head attention layer in a classification task. Our training and generalization bounds hold under an appropriate realizability condition asking for the existence of an a target model $\widetilde{\boldsymbol{\theta}}$ achieving good train loss while being sufficiently close to initialization. In particular, from the condition on the number of heads $H$ in (3), we need $\widetilde{\boldsymbol{\theta}}$ is at most $\tilde{\mathcal{O}} \left( d^{-1/3} T^{-1/6} R^{-5/3} H^{1/6} \right)$ far from initialization (provided $\| \widetilde{\boldsymbol{\theta}} \|_{2,\infty} = \mathcal{O}(1)$). In Sec. 4.3 we showed that such a model exists if the initialization is chosen appropriately. Specifically it suffices that $\| \widetilde{\boldsymbol{\theta}}^{(0)} \|_{2,\infty} = \mathcal{O}(1)$, the model output at initialization is $\tilde{\mathcal{O}}(1)$-bounded and that the data are linearly separable with margin $\gamma$ with respect to the NTK features of the model at initialization. Then, $\mathcal{O} \left( d^2 T R^{10} \, \mathrm{polylog}(n)/\gamma^6 \right)$ number of heads guarantee that $\Theta(n)$ GD steps result in train and test loss bounds $\tilde{\mathcal{O}} \left( 1/(\eta \gamma^2 n) \right)$. In Sec. 5 we applied our results to a tokenized-mixture model. We showed that after one randomized gradient step from $\mathbf{0}$, the model satisfies the above conditions for good intialization. For this initialization, we computed the NTK margin $\gamma_\star$ which in turn governs the guaranteed rate of convergence and generalization based on our general bounds. This opens several interesting questions for future work.

First, does random initialization of attention weights satisfy NTK separability, and if so, what is the corresponding margin? Second, are there other initialization strategies that guarantee the realizability conditions are satisfied? Here, note that our conditions for good initialization are only shown to be sufficient for realizability leaving room for improvements. Third, how suboptimal is the best NTK margin (among other potential natural initializations) compared to the model's global margin $\arg\max_{\| \widetilde{\boldsymbol{\theta}} \| = \sqrt{2}} \min_{i \in [n]} y_i \widetilde{\Phi}(\boldsymbol{X}_i; \widetilde{\boldsymbol{\theta}})$? In Proposition 3 we showed for the data model DM1 that there exists single-head attention model $\boldsymbol{\theta}_{\mathrm{opt}} =$

$(\boldsymbol{U}_{\mathrm{opt}}, \boldsymbol{W}_{\mathrm{opt}})$ with $\|\boldsymbol{\theta}_{\mathrm{opt}}\| = \sqrt{2}$ such that $y\Phi(\boldsymbol{X}; \boldsymbol{U}_{\mathrm{opt}}, \Gamma_\epsilon \cdot \boldsymbol{W}_{\mathrm{opt}}) = \frac{S\sqrt{T}}{\sqrt{2}}\left((1-\epsilon) - 2\epsilon Z_\mu/S\right)$ for all $\Gamma_\epsilon \gtrsim \frac{\log((\zeta^{-1}-1)/\epsilon)}{S}$ and any $\epsilon \in (0,1)$ (see App. E). In particular, as $\epsilon \to 0$ and $\Gamma_\epsilon \to \infty$ (for which the softmax map gets saturated and attends to tokens with highest relevance score) the achieved margin approaches $\gamma_{\mathrm{attn}} := S\sqrt{T}/\sqrt{2}$, which is independent of the sparsity level $\zeta$. In particular, $\gamma_{\mathrm{attn}} \geq \gamma_\star \geq \gamma_{\mathrm{lin}}$ and the gap increases with decreasing sparsity. Is it possible to establish finite-time convergence bounds to models with margin $\approx \gamma_{\mathrm{attn}}$ under appropriate initialization? How is the answer affected by the fact that the optimal attention weights in this case are diverging in norm ($\Gamma_\epsilon \to \infty$)? Using our approach, we argued in Sec. 5.3 that the key challenge is the saturation of norm of $\boldsymbol{W}_{\mathrm{opt}}$ ($\Gamma_\epsilon$), which does not allow the appropriate realizability condition to hold (at least for $\boldsymbol{0}$ initialization). Finally, it is interesting to consider other data models for which multiple heads are necessary to interpolate the data.

## 8 Acknowledgements

This work is supported by NSERC Discovery Grant RGPIN-2021-03677, NSF Grant CCF-2009030, and a CIFAR AI Catalyst grant. The authors also acknowledge use of the Sockeye cluster by UBC Advanced Research Computing. PD thanks Bhavya Vasudeva for the helpful discussions.

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

## Contents

## A  Gradients and Hessian Calculations

We define the following for convenience

$$\widehat{L}'(\boldsymbol{\theta}) = \frac{1}{n}\sum_{i\in[n]} |\ell'(y_i\Phi(\boldsymbol{X}_i,\boldsymbol{\theta}))|\,, \qquad \text{and} \qquad \widehat{L}''(\boldsymbol{\theta}) = \frac{1}{n}\sum_{i\in[n]} |\ell''(y_i\Phi(\boldsymbol{X}_i,\boldsymbol{\theta}))|\,.$$

For logistic loss $\ell(z) := \log(1 + e^{-z})$:

$$\widehat{L}'(\boldsymbol{\theta}) \le \widehat{L}(\boldsymbol{\theta}) \le 1, \qquad \text{and} \qquad \widehat{L}''(\boldsymbol{\theta}) \le \frac{1}{4}. \tag{24}$$

We use $\vec{\boldsymbol{A}}$ to denote the the vectorization of a matrix $\boldsymbol{A}$ and $\odot$ to denote the Hadamard product. Finally, we define $\mathbf{e}_i^{(n)}$ as the $i$-th standard basis vector in $\mathbb{R}^n$.

## A.1 Gradient/Hessian calculations for multihead-attention

**Lemma 4.** *Let the softmax attention model $\Phi(\boldsymbol{X}; \boldsymbol{\theta})$ in Eq. (1). Then, for all vectors $\boldsymbol{a} \in \mathbb{R}^T$ and $\boldsymbol{b}, \boldsymbol{c} \in \mathbb{R}^d$ it holds:*

1. $\nabla_{\boldsymbol{U}} \Phi(\boldsymbol{X}; \boldsymbol{\theta}) = \boldsymbol{\varphi}(\boldsymbol{X}\boldsymbol{W}\boldsymbol{X}^\top)\boldsymbol{X}$ .

2. $\nabla_{\boldsymbol{W}} \Phi(\boldsymbol{X}; \boldsymbol{\theta}) = \sum\limits_{t=1}^{T} \boldsymbol{x}_t \boldsymbol{u}_t^\top \boldsymbol{X}^\top \boldsymbol{\varphi}'(\boldsymbol{X}\boldsymbol{W}^\top \boldsymbol{x}_t)\boldsymbol{X}$ .

3. $\nabla_{\boldsymbol{W}} \langle \boldsymbol{a}, \nabla_{\boldsymbol{U}} \Phi(\boldsymbol{X}; \boldsymbol{\theta})\boldsymbol{b} \rangle = \sum\limits_{t=1}^{T} \boldsymbol{x}_t a_t \boldsymbol{b}^\top \boldsymbol{X}^\top \boldsymbol{\varphi}'(\boldsymbol{X}\boldsymbol{W}^\top \boldsymbol{x}_t)\boldsymbol{X}$ .

4. $\nabla_{\boldsymbol{W}} \langle \boldsymbol{c}, \nabla_{\boldsymbol{W}} \Phi(\boldsymbol{X}; \boldsymbol{\theta})\boldsymbol{b} \rangle = \sum\limits_{t=1}^{T} \boldsymbol{c}^\top \boldsymbol{x}_t \boldsymbol{x}_t \Big( \boldsymbol{u}_t^\top \boldsymbol{X}^\top \operatorname{diag}(\boldsymbol{X}\boldsymbol{b}) - \boldsymbol{\varphi}(\boldsymbol{X}\boldsymbol{W}^\top \boldsymbol{x}_t)^\top \boldsymbol{X}\boldsymbol{b} \, (\boldsymbol{X}\boldsymbol{u}_t)^\top$
$$- \boldsymbol{\varphi}(\boldsymbol{X}\boldsymbol{W}^\top \boldsymbol{x}_t)^\top \boldsymbol{X}\boldsymbol{u}_t \, (\boldsymbol{X}\boldsymbol{b})^\top \Big) \boldsymbol{\varphi}'(\boldsymbol{X}\boldsymbol{W}^\top \boldsymbol{x}_t)\boldsymbol{X} .$$

*Proof.* For simplicity, we denote $\boldsymbol{G} := \boldsymbol{X}\boldsymbol{W}\boldsymbol{X}^\top$ and the rows of $\boldsymbol{\varphi}(\boldsymbol{G})$ as $\boldsymbol{\varphi}(\boldsymbol{g}_t) = \boldsymbol{\varphi}(\boldsymbol{X}\boldsymbol{W}^\top \boldsymbol{x}_t)$, $t \in [T]$. Recall, for any $\boldsymbol{v} \in \mathbb{R}^d$,

$$\boldsymbol{\varphi}'(\boldsymbol{d}) = \nabla_{\boldsymbol{v}} \boldsymbol{\varphi}(\boldsymbol{v}) = \operatorname{diag}(\boldsymbol{\varphi}(\boldsymbol{v})) - \boldsymbol{\varphi}(\boldsymbol{v})\boldsymbol{\varphi}(\boldsymbol{v})^\top , \tag{25}$$

i.e.,

$$\boldsymbol{\varphi}(\boldsymbol{v} + \boldsymbol{\delta}) = \boldsymbol{\varphi}(\boldsymbol{v}) + \big( \operatorname{diag}(\boldsymbol{\varphi}(\boldsymbol{v})) - \boldsymbol{\varphi}(\boldsymbol{v})\boldsymbol{\varphi}(\boldsymbol{v})^\top \big) \boldsymbol{\delta} + o(\|\boldsymbol{\delta}\|^2) . \tag{26}$$

We start with the gradient with respect to $\boldsymbol{U}$,

$$\nabla_{\boldsymbol{U}} \Phi(\boldsymbol{X}; \boldsymbol{\theta}) = \boldsymbol{\varphi}(\boldsymbol{G})\boldsymbol{X} . \tag{27}$$

Next step is to compute the gradient with respect to $\boldsymbol{W}$, $\nabla_{\boldsymbol{W}} \Phi(\boldsymbol{X}; \boldsymbol{\theta}) = \sum_{t=1}^{T} \nabla_{\boldsymbol{W}} \left( \boldsymbol{u}_t^\top \boldsymbol{X}^\top \boldsymbol{\varphi}(\boldsymbol{g}_t) \right)$. Using Eq. (26),

$$\boldsymbol{u}_t^\top \boldsymbol{X}^\top \boldsymbol{\varphi}(\boldsymbol{X}(\boldsymbol{W} + \boldsymbol{\Delta})^\top \boldsymbol{x}_t) = \boldsymbol{u}_t^\top \boldsymbol{X}^\top \big( \boldsymbol{\varphi}(\boldsymbol{g}_t) + \big( \operatorname{diag}(\boldsymbol{\varphi}(\boldsymbol{g}_t)) - \boldsymbol{\varphi}(\boldsymbol{g}_t)\boldsymbol{\varphi}(\boldsymbol{g}_t)^\top \big) \boldsymbol{X}\boldsymbol{\Delta}^\top \boldsymbol{x}_t \big) + o(\|\boldsymbol{\Delta}\|^2)$$
$$= \boldsymbol{u}_t^\top \boldsymbol{X}^\top \boldsymbol{\varphi}(\boldsymbol{g}_t) + \operatorname{tr}(\boldsymbol{x}_t \boldsymbol{u}_t^\top \boldsymbol{X}^\top \boldsymbol{\varphi}'(\boldsymbol{X}\boldsymbol{W}^\top \boldsymbol{x}_t)\boldsymbol{X}\boldsymbol{\Delta}^\top) + o(\|\boldsymbol{\Delta}\|^2) .$$

Thus,

$$\nabla_{\boldsymbol{W}} \left( \boldsymbol{u}_t^\top \boldsymbol{X}^\top \boldsymbol{\varphi}(\boldsymbol{g}_t) \right) = \boldsymbol{x}_t \boldsymbol{u}_t^\top \boldsymbol{X}^\top \boldsymbol{\varphi}'(\boldsymbol{X}\boldsymbol{W}^\top \boldsymbol{x}_t)\boldsymbol{X} \tag{28}$$

and

$$\nabla_{\boldsymbol{W}} \Phi(\boldsymbol{X}; \boldsymbol{\theta}) = \sum_{t=1}^{T} \boldsymbol{x}_t \boldsymbol{u}_t^\top \boldsymbol{X}^\top \boldsymbol{\varphi}'(\boldsymbol{X}\boldsymbol{W}^\top \boldsymbol{x}_t)\boldsymbol{X} . \tag{29}$$

For the third statement of the lemma, we have the following sequence of equalities,

$$\nabla_{\boldsymbol{W}}\langle\boldsymbol{a},\nabla_{\boldsymbol{U}}\Phi(\boldsymbol{X};\boldsymbol{\theta})\,\boldsymbol{b}\rangle = \nabla_{\boldsymbol{W}}\langle\boldsymbol{a},\boldsymbol{\varphi}(\boldsymbol{G})\boldsymbol{X}\boldsymbol{b}\rangle = \nabla_{\boldsymbol{W}}\left(\boldsymbol{a}^{\top}\boldsymbol{\varphi}(\boldsymbol{G})\,\boldsymbol{X}\boldsymbol{b}\right)$$

$$= \nabla_{\boldsymbol{W}}\operatorname{tr}\left(\boldsymbol{b}\boldsymbol{a}^{\top}\boldsymbol{\varphi}(\boldsymbol{G})\boldsymbol{X}\right) = \sum_{t=1}^{T}\nabla_{\boldsymbol{W}}\left(a_t\boldsymbol{b}^{\top}\boldsymbol{X}^{\top}\boldsymbol{\varphi}_t(\boldsymbol{g})\right)$$

$$= \sum_{t=1}^{T}\boldsymbol{x}_t a_t\boldsymbol{b}^{\top}\boldsymbol{X}^{\top}\boldsymbol{\varphi}'(\boldsymbol{X}\boldsymbol{W}^{\top}\boldsymbol{x}_t)\boldsymbol{X}\,, \tag{30}$$

where in the last equality we used Eq. (28).
For the Hessian with respect to $\boldsymbol{W}$, we have:

$$\nabla_{\boldsymbol{W}}\langle\boldsymbol{c},\nabla_{\boldsymbol{W}}\Phi(\boldsymbol{X};\boldsymbol{\theta})\,\boldsymbol{b}\rangle = \sum_{t=1}^{T}\nabla_{\boldsymbol{W}}\Big(\underbrace{\boldsymbol{c}^{\top}\boldsymbol{x}_t\boldsymbol{u}_t^{\top}\boldsymbol{X}^{\top}}_{\boldsymbol{p}_t^{\top}}\,\boldsymbol{\varphi}'(\boldsymbol{X}\boldsymbol{W}^{\top}\boldsymbol{x}_t)\,\underbrace{\boldsymbol{X}\boldsymbol{b}}_{\boldsymbol{q}_t}\Big)$$

$$= \sum_{t=1}^{T}\Big(\underbrace{\nabla_{\boldsymbol{W}}\boldsymbol{p}_t^{\top}\operatorname{diag}(\boldsymbol{q}_t)\boldsymbol{\varphi}(\boldsymbol{g}_t)}_{\mathrm{Term_I}} - \underbrace{\nabla_{\boldsymbol{W}}\boldsymbol{p}_t^{\top}\boldsymbol{\varphi}(\boldsymbol{g}_t)\,\boldsymbol{q}_t^{\top}\boldsymbol{\varphi}(\boldsymbol{g}_t)}_{\mathrm{Term_{II}}}\Big)\,, \tag{31}$$

where we used the property $\operatorname{diag}(\boldsymbol{a}_1)\,\boldsymbol{a}_2 = \operatorname{diag}(\boldsymbol{a}_2)\,\boldsymbol{a}_1$ for vectors $\boldsymbol{a}_1,\boldsymbol{a}_2\in\mathbb{R}^T$.
First, we compute the $\mathrm{Term_I}$ above. Using Eq. (26):

$$\boldsymbol{p}_t^{\top}\operatorname{diag}(\boldsymbol{q}_t)\boldsymbol{\varphi}(\boldsymbol{X}(\boldsymbol{W}+\boldsymbol{\Delta})^{\top}\boldsymbol{x}_t) = \boldsymbol{p}_t^{\top}\operatorname{diag}(\boldsymbol{q}_t)\left(\boldsymbol{\varphi}(\boldsymbol{g}_t)+\boldsymbol{\varphi}'(\boldsymbol{X}\boldsymbol{W}^{\top}\boldsymbol{x}_t)\boldsymbol{X}\boldsymbol{\Delta}^{\top}\boldsymbol{x}_t\right)+o(\|\boldsymbol{\Delta}\|^2)$$

$$= \boldsymbol{p}_t^{\top}\operatorname{diag}(\boldsymbol{q}_t)\boldsymbol{\varphi}(\boldsymbol{g}_t) + \operatorname{tr}(\boldsymbol{x}_t\boldsymbol{p}_t^{\top}\operatorname{diag}(\boldsymbol{q}_t)\boldsymbol{\varphi}'(\boldsymbol{X}\boldsymbol{W}^{\top}\boldsymbol{x}_t)\boldsymbol{X}\boldsymbol{\Delta}^{\top})+o(\|\boldsymbol{\Delta}\|^2)\,. \tag{32}$$

Therefore,

$$\nabla_{\boldsymbol{W}}\boldsymbol{p}_t^{\top}\operatorname{diag}(\boldsymbol{q}_t)\boldsymbol{\varphi}(\boldsymbol{g}_t) = \boldsymbol{x}_t\boldsymbol{p}_t^{\top}\operatorname{diag}(\boldsymbol{q}_t)\boldsymbol{\varphi}'(\boldsymbol{X}\boldsymbol{W}^{\top}\boldsymbol{x}_t)\boldsymbol{X}$$

$$= \boldsymbol{x}_t\,\boldsymbol{c}^{\top}\boldsymbol{x}_t\,\boldsymbol{u}_t^{\top}\boldsymbol{X}^{\top}\operatorname{diag}(\boldsymbol{X}\boldsymbol{b})\boldsymbol{\varphi}'(\boldsymbol{X}\boldsymbol{W}^{\top}\boldsymbol{x}_t)\boldsymbol{X}\,. \tag{33}$$

Similarly for $\mathrm{Term_{II}}$ we have:

$$\boldsymbol{p}_t^{\top}\boldsymbol{\varphi}(\boldsymbol{X}(\boldsymbol{W}+\boldsymbol{\Delta})^{\top}\boldsymbol{x}_t)\,\boldsymbol{q}_t^{\top}\boldsymbol{\varphi}(\boldsymbol{X}(\boldsymbol{W}+\boldsymbol{\Delta})^{\top}\boldsymbol{x}_t) = \left(\boldsymbol{p}_t^{\top}\boldsymbol{\varphi}(\boldsymbol{g}_t)+\boldsymbol{p}_t^{\top}\boldsymbol{\varphi}'(\boldsymbol{X}\boldsymbol{W}^{\top}\boldsymbol{x}_t)\boldsymbol{X}\boldsymbol{\Delta}^{\top}\boldsymbol{x}_t+o(\|\boldsymbol{\Delta}\|^2)\right)$$

$$\left(\boldsymbol{q}_t^{\top}\boldsymbol{\varphi}(\boldsymbol{g}_t)+\boldsymbol{q}_t^{\top}\boldsymbol{\varphi}'(\boldsymbol{X}\boldsymbol{W}^{\top}\boldsymbol{x}_t)\boldsymbol{X}\boldsymbol{\Delta}^{\top}\boldsymbol{x}_t+o(\|\boldsymbol{\Delta}\|^2)\right)$$

$$= \boldsymbol{p}_t^{\top}\boldsymbol{\varphi}(\boldsymbol{g}_t)\,\boldsymbol{q}_t^{\top}\boldsymbol{\varphi}(\boldsymbol{g}_t) + \operatorname{tr}\Big(\boldsymbol{x}_t(\boldsymbol{q}^{\top}\boldsymbol{\varphi}(\boldsymbol{g}_t)\boldsymbol{p}^{\top}$$

$$+\boldsymbol{p}^{\top}\boldsymbol{\varphi}(\boldsymbol{g}_t)\boldsymbol{q}^{\top})\,\boldsymbol{\varphi}'(\boldsymbol{X}\boldsymbol{W}^{\top}\boldsymbol{x}_t)\boldsymbol{X}\boldsymbol{\Delta}^{\top}\Big)+o(\|\boldsymbol{\Delta}\|^2)\,. \tag{34}$$

Thus,

$$\nabla_{\boldsymbol{W}}\boldsymbol{p}_t^{\top}\boldsymbol{\varphi}(\boldsymbol{g}_t)\,\boldsymbol{q}_t^{\top}\boldsymbol{\varphi}(\boldsymbol{g}_t) = \boldsymbol{x}_t(\boldsymbol{q}^{\top}\boldsymbol{\varphi}(\boldsymbol{g}_t)\boldsymbol{p}^{\top}+\boldsymbol{p}^{\top}\boldsymbol{\varphi}(\boldsymbol{g}_t)\boldsymbol{q}^{\top})\boldsymbol{\varphi}'(\boldsymbol{X}\boldsymbol{W}^{\top}\boldsymbol{x}_t)\boldsymbol{X}$$

$$= \boldsymbol{c}^{\top}\boldsymbol{x}_t\,\boldsymbol{x}_t\,\boldsymbol{\varphi}(\boldsymbol{g}_t)^{\top}\left(\boldsymbol{X}\boldsymbol{u}_t\,(\boldsymbol{X}\boldsymbol{b})^{\top}+\boldsymbol{X}\boldsymbol{b}\,(\boldsymbol{X}\boldsymbol{u}_t)^{\top}\right)\boldsymbol{\varphi}'(\boldsymbol{X}\boldsymbol{W}^{\top}\boldsymbol{x}_t)\boldsymbol{X}\,. \tag{35}$$

Combining Eqs. (33) and (35) and plugging in Eq. (31) we conclude that

$$\nabla_{\boldsymbol{W}}\langle\boldsymbol{c},\nabla_{\boldsymbol{W}}\Phi(\boldsymbol{X};\boldsymbol{\theta})\,\boldsymbol{b}\rangle = \sum_{t=1}^{T}\boldsymbol{c}^{\top}\boldsymbol{x}_t\,\boldsymbol{x}_t\Big(\boldsymbol{u}_t^{\top}\boldsymbol{X}^{\top}\operatorname{diag}(\boldsymbol{X}\boldsymbol{b})-\boldsymbol{\varphi}(\boldsymbol{g}_t)^{\top}\boldsymbol{X}\boldsymbol{b}\,(\boldsymbol{X}\boldsymbol{u}_t)^{\top}$$

$$-\boldsymbol{\varphi}(\boldsymbol{g}_t)^{\top}\boldsymbol{X}\boldsymbol{u}_t\,(\boldsymbol{X}\boldsymbol{b})^{\top}\Big)\boldsymbol{\varphi}'(\boldsymbol{X}\boldsymbol{W}^{\top}\boldsymbol{x}_t)\boldsymbol{X}\,. \tag{36}$$

$\square$

We restate Proposition 1 here for convenience.

**Proposition 5** (Restatement of Prop. 1)**.** *Let the softmax attention model $\Phi(\boldsymbol{X};\boldsymbol{\theta})$ in Eq.* (1)*. Then, it holds:*

1. $\|\nabla_{\check{\boldsymbol{U}}}\Phi(\boldsymbol{X};\boldsymbol{\theta})\| \leq \sqrt{T}\,\|\boldsymbol{X}\|_{2,\infty}$ .

2. $\|\nabla_{\check{\boldsymbol{W}}}\Phi(\boldsymbol{X};\boldsymbol{\theta})\| \leq 2\,\|\boldsymbol{X}\|_{2,\infty}^2 \sum_{t=1}^{T}\|\boldsymbol{X}\boldsymbol{u}_t\|_\infty$ .

3. $\|\nabla_{\boldsymbol{\theta}}\Phi(\boldsymbol{X};\boldsymbol{\theta})\| \leq 2\,\|\boldsymbol{X}\|_{2,\infty}^2 \sum_{t=1}^{T}\|\boldsymbol{X}\boldsymbol{u}_t\|_\infty + \sqrt{T}\,\|\boldsymbol{X}\|_{2,\infty}$ .

4. $\|\nabla_{\boldsymbol{\theta}}^2\Phi(\boldsymbol{X};\boldsymbol{\theta})\| \leq 6\,d\,\|\boldsymbol{X}\|_{2,\infty}^2\,\|\boldsymbol{X}\|_{1,\infty}^2 \sum_{t=1}^{T}\|\boldsymbol{X}\boldsymbol{u}_t\|_\infty + 2\,\sqrt{T\,d}\,\|\boldsymbol{X}\|_{2,\infty}^2\,\|\boldsymbol{X}\|_{1,\infty}$ .

*Moreover, if Assumption 1 holds,*

1. $\|\nabla_{\boldsymbol{\theta}}\Phi(\boldsymbol{X};\boldsymbol{\theta})\| \leq \sqrt{T}\,R\,\left(2\,R^2\,\|\boldsymbol{U}\|_F + 1\right)$ .

2. $\|\nabla_{\boldsymbol{\theta}}^2\Phi(\boldsymbol{X};\boldsymbol{\theta})\| \leq 2\,\sqrt{T\,d}\,R^3\,\left(3\,\sqrt{d}\,R^2\,\|\boldsymbol{U}\|_F + 1\right)$ .

*Proof.* Using the first statement of Lemma 4,

$$\|\nabla_{\check{\boldsymbol{U}}}\Phi(\boldsymbol{X};\boldsymbol{\theta})\| = \sqrt{\sum_{t=1}^{T}\|\boldsymbol{X}^\top\boldsymbol{\varphi}(\boldsymbol{g}_t)\|_2^2} \leq \sqrt{\sum_{t=1}^{T}\|\boldsymbol{X}^\top\|_{1,2}^2\|\boldsymbol{\varphi}(\boldsymbol{g}_t)\|_1^2}$$
$$= \sqrt{T}\,\max_{t\in[T]}\|\boldsymbol{x}_t\| = \sqrt{T}\,\|\boldsymbol{X}\|_{2,\infty}\,, \tag{37}$$

where we used that $\|\boldsymbol{\varphi}(\boldsymbol{g}_t)\|_1 = 1$.

Using Lemma 4 for the gradient with respect to $\boldsymbol{W}$,

$$\nabla_{\boldsymbol{W}}\Phi(\boldsymbol{X};\boldsymbol{\theta}) = \sum_{t=1}^{T}\boldsymbol{x}_t\boldsymbol{u}_t^\top\boldsymbol{X}^\top\boldsymbol{\varphi}'(\boldsymbol{X}\boldsymbol{W}^\top\boldsymbol{x}_t)\boldsymbol{X}\,. \tag{38}$$

Therefore, by applying triangle inequality,

$$\|\nabla_{\check{\boldsymbol{W}}}\Phi(\boldsymbol{X};\boldsymbol{\theta})\| \leq \sum_{t=1}^{T}\|\boldsymbol{x}_t\|\,\|\boldsymbol{X}^\top\boldsymbol{\varphi}'(\boldsymbol{g}_t)\boldsymbol{X}\boldsymbol{u}_t\|\,.$$

In the next step we bound $\|\boldsymbol{X}^\top\boldsymbol{\varphi}'(\boldsymbol{g}_t)\boldsymbol{X}\boldsymbol{u}_t\|$,

$$\|\boldsymbol{X}^\top\boldsymbol{\varphi}'(\boldsymbol{X}\boldsymbol{W}^\top\boldsymbol{x}_t)\boldsymbol{X}\boldsymbol{u}_t\| \leq \underbrace{\|\boldsymbol{X}^\top\operatorname{diag}(\boldsymbol{\varphi}(\boldsymbol{g}_t))\boldsymbol{X}\boldsymbol{u}_t\|}_{\text{Term}_\text{I}} + \underbrace{\|\boldsymbol{X}^\top\boldsymbol{\varphi}(\boldsymbol{g}_t)\boldsymbol{\varphi}(\boldsymbol{g}_t)^\top\boldsymbol{X}\boldsymbol{u}_t\|}_{\text{Term}_\text{II}}\,. \tag{39}$$

For $\text{Term}_\text{I}$ we do as follows:

$$\|\boldsymbol{X}^\top\operatorname{diag}(\boldsymbol{\varphi}(\boldsymbol{g}_t))\boldsymbol{X}\boldsymbol{u}_t\| = \max_{\boldsymbol{v}\in\mathbb{R}^d\text{ and }\|\boldsymbol{v}\|=1}(\boldsymbol{X}\boldsymbol{v})^\top\operatorname{diag}(\boldsymbol{\varphi}(\boldsymbol{g}_t))\boldsymbol{X}\boldsymbol{u}_t$$
$$= \max_{\|\boldsymbol{v}\|=1}\sum_{\tau=1}^{T}[\operatorname{diag}(\boldsymbol{\varphi}(\boldsymbol{g}_t))]_{\tau\tau}\,[\boldsymbol{X}\boldsymbol{u}_t]_\tau\,[\boldsymbol{X}\boldsymbol{v}]_\tau$$
$$= \max_{\|\boldsymbol{v}\|=1}\boldsymbol{\varphi}(\boldsymbol{g}_t)^\top\left((\boldsymbol{X}\boldsymbol{u}_t)\odot(\boldsymbol{X}\boldsymbol{v})\right)$$
$$\leq \max_{\|\boldsymbol{v}\|=1}\|\boldsymbol{\varphi}(\boldsymbol{g}_t)\|_1\,\|(\boldsymbol{X}\boldsymbol{u}_t)\odot(\boldsymbol{X}\boldsymbol{v})\|_\infty$$
$$\leq \|\boldsymbol{X}\boldsymbol{u}_t\|_\infty\,\max_{\|\boldsymbol{v}\|=1}\|\boldsymbol{X}\boldsymbol{v}\|_\infty$$
$$= \|\boldsymbol{X}\boldsymbol{u}_t\|_\infty\,\|\boldsymbol{X}\|_{2,\infty}\,, \tag{40}$$

where we used Hölder's inequality in the first inequality.

Then, we compute Term$_{\text{II}}$:

$$
\begin{aligned}
\|\boldsymbol{X}^\top\boldsymbol{\varphi}(\boldsymbol{g}_t)\boldsymbol{\varphi}(\boldsymbol{g}_t)^\top\boldsymbol{X}\boldsymbol{u}_t\| &= \max_{\boldsymbol{v}\in\mathbb{R}^d \text{ and } \|\boldsymbol{v}\|=1}(\boldsymbol{X}\boldsymbol{v})^\top\boldsymbol{\varphi}(\boldsymbol{g}_t)\boldsymbol{\varphi}(\boldsymbol{g}_t)^\top\boldsymbol{X}\boldsymbol{u}_t \\
&\leq \max_{\|\boldsymbol{v}\|=1}\|\boldsymbol{\varphi}(\boldsymbol{g}_t)\|_1^2\|\boldsymbol{X}\boldsymbol{u}_t\|_\infty\|\boldsymbol{X}\boldsymbol{v}\|_\infty \\
&= \|\boldsymbol{X}\boldsymbol{u}_t\|_\infty\max_{\|\boldsymbol{v}\|=1}\|\boldsymbol{X}\boldsymbol{v}\|_\infty \\
&= \|\boldsymbol{X}\boldsymbol{u}_t\|_\infty\|\boldsymbol{X}\|_{2,\infty}\,,
\end{aligned}
\tag{41}
$$

where we used Hölder's inequality in the first inequality. Combining Eqs. (40) and (41) and plugging in Eq. (39) yields

$$
\|(\boldsymbol{X}\boldsymbol{u}_t)^\top\boldsymbol{\varphi}'(\boldsymbol{g}_t)\boldsymbol{X}\| \leq 2\|\boldsymbol{X}\boldsymbol{u}_t\|_\infty\|\boldsymbol{X}\|_{2,\infty}\,.
\tag{42}
$$

Therefore,

$$
\|\nabla_{\vec{\boldsymbol{W}}}\Phi(\boldsymbol{X};\boldsymbol{\theta})\| \leq 2\|\boldsymbol{X}\|_{2,\infty}\sum_{t=1}^{T}\|\boldsymbol{x}_t\|\,\|\boldsymbol{X}\boldsymbol{u}_t\|_\infty\,.
\tag{43}
$$

Using Eqs. (37) and (43) we conclude that

$$
\|\nabla\Phi(\boldsymbol{X};\boldsymbol{\theta})\| \leq 2\|\boldsymbol{X}\|_{2,\infty}^2\sum_{t=1}^{T}\|\boldsymbol{X}\boldsymbol{u}_t\|_\infty + \sqrt{T}\|\boldsymbol{X}\|_{2,\infty}\,.
\tag{44}
$$

By using Assumption 1 to simplify Eq. (44),

$$
\|\nabla_{\boldsymbol{\theta}}\Phi(\boldsymbol{X};\boldsymbol{\theta})\| \leq 2R^2\sum_{t=1}^{T}\|\boldsymbol{X}\boldsymbol{u}_t\|_\infty + \sqrt{T}\,R\,,
\tag{45}
$$

or

$$
\|\nabla_{\boldsymbol{\theta}}\Phi(\boldsymbol{X};\boldsymbol{\theta})\| \leq 2\sqrt{T}\,R^3\|\boldsymbol{U}\|_F + \sqrt{T}\,R\,.
\tag{46}
$$

We need to derive the Hessian to bound the greatest eigenvalue of it.

$$
\nabla_{\boldsymbol{\theta}}^2\Phi(\boldsymbol{X};\boldsymbol{\theta}) = \begin{bmatrix} \nabla_{\vec{\boldsymbol{U}}}^2\Phi(\boldsymbol{X};\boldsymbol{\theta}) & \nabla_{\vec{\boldsymbol{W}}\vec{\boldsymbol{U}}}^2\Phi(\boldsymbol{X};\boldsymbol{\theta}) \\ \nabla_{\vec{\boldsymbol{W}}\vec{\boldsymbol{U}}}^2\Phi(\boldsymbol{X};\boldsymbol{\theta})^\top & \nabla_{\vec{\boldsymbol{W}}}^2\Phi(\boldsymbol{X};\boldsymbol{\theta}) \end{bmatrix}\,.
$$

First, we compute the Hessian with respect to $\boldsymbol{U}$,

$$
\nabla_{\vec{\boldsymbol{U}}}^2\Phi(\boldsymbol{X};\boldsymbol{\theta}) = \nabla_{\boldsymbol{U}}(\boldsymbol{\varphi}(\boldsymbol{G})\boldsymbol{X}) = \boldsymbol{0}_{Td\times Td}\,.
\tag{47}
$$

In the next step, we use the third statement of Lemma 4 and set $\boldsymbol{a} = \mathbf{e}_t^{(T)}$ and $\boldsymbol{b} = \mathbf{e}_j^{(d)}$ to compute the Hessian with respect to $\boldsymbol{W}$ and $\boldsymbol{U}$,

$$
\nabla_{\boldsymbol{W}}([\nabla_{\boldsymbol{U}}\Phi(\boldsymbol{X};\boldsymbol{\theta})]_{tj}) = \boldsymbol{x}_t\boldsymbol{X}_{:,j}^\top\boldsymbol{\varphi}'(\boldsymbol{g}_t)\boldsymbol{X}\,.
\tag{48}
$$

recall, $\boldsymbol{X}_{:,j}$ is the $j$-th column of $\boldsymbol{X}$. Therefore,

$$
\nabla_{\vec{\boldsymbol{W}}\vec{\boldsymbol{U}}}^2\Phi(\boldsymbol{X};\boldsymbol{\theta}) = \begin{bmatrix} \nabla_{\boldsymbol{W}}([\nabla_{\boldsymbol{U}}\Phi(\boldsymbol{X};\boldsymbol{\theta})]_{11}) & \dots & \nabla_{\boldsymbol{W}}([\nabla_{\boldsymbol{U}}\Phi(\boldsymbol{X};\boldsymbol{\theta})]_{1d}) \\ \vdots & \ddots & \vdots \\ \nabla_{\boldsymbol{W}}([\nabla_{\boldsymbol{U}}\Phi(\boldsymbol{X};\boldsymbol{\theta})]_{T1}) & \dots & \nabla_{\boldsymbol{W}}([\nabla_{\boldsymbol{U}}\Phi(\boldsymbol{X};\boldsymbol{\theta})]_{Td}) \end{bmatrix}\,.
\tag{49}
$$

Lastly, in order to compute the Hessian with respect to $\boldsymbol{W}$, we use the last statement of Lemma 4 and set $\boldsymbol{c} = \mathbf{e}_i^{(d)}$ and $\boldsymbol{b} = \mathbf{e}_j^{(d)}$,

$$
\begin{aligned}
\nabla_{\boldsymbol{W}}([\nabla_{\boldsymbol{W}}\Phi(\boldsymbol{X};\boldsymbol{\theta})]_{ij}) = \sum_{t=1}^{T}X_{ti}\boldsymbol{x}_t\big(\boldsymbol{u}_t^\top\boldsymbol{X}^\top\operatorname{diag}(\boldsymbol{X}_{:,j}) - \boldsymbol{\varphi}(\boldsymbol{g}_t)^\top\boldsymbol{X}_{:,j}\boldsymbol{u}_t^\top\boldsymbol{X}^\top \\
- \boldsymbol{\varphi}(\boldsymbol{g}_t)^\top\boldsymbol{X}\boldsymbol{u}_t\boldsymbol{X}_{:,j}^\top\big)\boldsymbol{\varphi}'(\boldsymbol{g}_t)\boldsymbol{X}\,.
\end{aligned}
\tag{50}
$$

Thus,

$$\nabla_{\tilde{\boldsymbol{W}}}^2 \Phi(\boldsymbol{X};\boldsymbol{\theta}) = \begin{bmatrix} \nabla_{\boldsymbol{W}}([\nabla_{\boldsymbol{W}}\Phi(\boldsymbol{X};\boldsymbol{\theta})]_{11}) & \dots & \nabla_{\boldsymbol{W}}([\nabla_{\boldsymbol{W}}\Phi(\boldsymbol{X};\boldsymbol{\theta})]_{1d}) \\ \vdots & \ddots & \vdots \\ \nabla_{\boldsymbol{W}}([\nabla_{\boldsymbol{W}}\Phi(\boldsymbol{X};\boldsymbol{\theta})]_{d1}) & \dots & \nabla_{\boldsymbol{W}}([\nabla_{\boldsymbol{W}}\Phi(\boldsymbol{X};\boldsymbol{\theta})]_{dd}) \end{bmatrix}. \tag{51}$$

To find the maximum eigenvalue of the Hessian we need to upper-bound

$$\max_{\|\boldsymbol{v}\|=1}\langle \boldsymbol{v}, \nabla_{\boldsymbol{\theta}}^2 \Phi(\boldsymbol{X};\boldsymbol{\theta})\,\boldsymbol{v}\rangle,$$

where

$$\boldsymbol{v} = \text{concat}\,(\boldsymbol{p}_1,\dots,\boldsymbol{p}_T,\boldsymbol{q}_1,\dots,\boldsymbol{q}_d) \in \mathbb{R}^{Td+d^2} \qquad \text{and} \qquad \boldsymbol{p}_t \in \mathbb{R}^d, t \in [T] \text{ and } \boldsymbol{q}_j \in \mathbb{R}^d, j \in [d].$$

Then,

$$\|\nabla_{\boldsymbol{\theta}}^2 \Phi(\boldsymbol{X};\boldsymbol{\theta})\|$$

$$\le \underbrace{\max_{\|\boldsymbol{v}\|=1} \sum_{i=1}^{d}\sum_{j=1}^{d}\sum_{t=1}^{T} X_{ti}\,\boldsymbol{q}_i^{\top}\boldsymbol{x}_t \left((\boldsymbol{X}\boldsymbol{u}_t)^{\top}\text{diag}(\boldsymbol{X}_{:,j}) - \boldsymbol{\varphi}(\boldsymbol{g}_t)^{\top}\boldsymbol{X}_{:,j}(\boldsymbol{X}\boldsymbol{u}_t)^{\top} - \boldsymbol{\varphi}(\boldsymbol{g}_t)^{\top}\boldsymbol{X}\boldsymbol{u}_t\boldsymbol{X}_{:,j}^{\top}\right)\boldsymbol{\varphi}'(\boldsymbol{g}_t)\boldsymbol{X}\boldsymbol{q}_j}_{\text{Term}_1}$$

$$+ \underbrace{2\max_{\|\boldsymbol{v}\|=1} \sum_{j=1}^{d}\sum_{t=1}^{T} \boldsymbol{p}_t^{\top}\boldsymbol{x}_t\,\boldsymbol{X}_{:,j}^{\top}\boldsymbol{\varphi}'(\boldsymbol{g}_t)\boldsymbol{X}\boldsymbol{q}_j}_{\text{Term}_2}. \tag{52}$$

First, we bound Term$_1$:

$$\max_{\|\boldsymbol{v}\|=1} \sum_{i=1}^{d}\sum_{j=1}^{d}\sum_{t=1}^{T} X_{ti}\,\boldsymbol{q}_i^{\top}\boldsymbol{x}_t \left((\boldsymbol{X}\boldsymbol{u}_t)^{\top}\text{diag}(\boldsymbol{X}_{:,j}) - \boldsymbol{\varphi}(\boldsymbol{g}_t)^{\top}\boldsymbol{X}_{:,j}(\boldsymbol{X}\boldsymbol{u}_t)^{\top} - \boldsymbol{\varphi}(\boldsymbol{g}_t)^{\top}\boldsymbol{X}\boldsymbol{u}_t\boldsymbol{X}_{:,j}^{\top}\right)\boldsymbol{\varphi}'(\boldsymbol{g}_t)\boldsymbol{X}\boldsymbol{q}_j$$

$$\le \|\boldsymbol{X}\|_{1,\infty}\|\boldsymbol{X}\|_{2,\infty}\left(\underbrace{\max_{\|\boldsymbol{v}\|=1} \sum_{i=1}^{d}\sum_{j=1}^{d}\sum_{t=1}^{T}\|\boldsymbol{q}_i\|\,\|\boldsymbol{q}_j\|\,\|(\boldsymbol{X}\boldsymbol{u}_t)^{\top}\text{diag}(\boldsymbol{X}_{:,j})\boldsymbol{\varphi}'(\boldsymbol{g}_t)\boldsymbol{X}\|}_{\text{Term}_{\text{I}}} + \right.$$

$$\underbrace{\max_{\|\boldsymbol{v}\|=1} \sum_{i=1}^{d}\sum_{j=1}^{d}\sum_{t=1}^{T}\|\boldsymbol{q}_i\|\,\|\boldsymbol{q}_j\|\,\|\boldsymbol{\varphi}(\boldsymbol{g}_t)^{\top}\boldsymbol{X}_{:,j}(\boldsymbol{X}\boldsymbol{u}_t)^{\top}\boldsymbol{\varphi}'(\boldsymbol{g}_t)\boldsymbol{X}\|}_{\text{Term}_{\text{II}}} + $$

$$\left.\underbrace{\max_{\|\boldsymbol{v}\|=1} \sum_{i=1}^{d}\sum_{j=1}^{d}\sum_{t=1}^{T}\|\boldsymbol{q}_i\|\,\|\boldsymbol{q}_j\|\,\|\boldsymbol{\varphi}(\boldsymbol{g}_t)^{\top}\boldsymbol{X}\boldsymbol{u}_t\boldsymbol{X}_{:,j}^{\top}\boldsymbol{\varphi}'(\boldsymbol{g}_t)\boldsymbol{X}\|}_{\text{Term}_{\text{III}}}\right). \tag{53}$$

For Term$_{\text{I}}$, we have:

$$\max_{\|\boldsymbol{v}\|=1} \sum_{i=1}^{d}\sum_{j=1}^{d}\sum_{t=1}^{T}\|\boldsymbol{q}_i\|\,\|\boldsymbol{q}_j\|\,\|(\boldsymbol{X}\boldsymbol{u}_t)^{\top}\text{diag}(\boldsymbol{X}_{:,j})\boldsymbol{\varphi}'(\boldsymbol{g}_t)\boldsymbol{X}\|$$

$$\le 2\max_{\|\boldsymbol{v}\|=1} \sum_{i=1}^{d}\sum_{j=1}^{d}\sum_{t=1}^{T}\|\boldsymbol{q}_i\|\,\|\boldsymbol{q}_j\|\,\|\boldsymbol{X}\|_{2,\infty}\|\text{diag}(\boldsymbol{X}_{:,j})\boldsymbol{X}\boldsymbol{u}_t\|_{\infty}$$

$$\le 2\max_{\|\boldsymbol{v}\|=1} \sum_{i=1}^{d}\sum_{j=1}^{d}\sum_{t=1}^{T}\|\boldsymbol{q}_i\|\,\|\boldsymbol{q}_j\|\,\|\boldsymbol{X}\|_{2,\infty}\|\boldsymbol{X}_{:,j}\|_{\infty}\|\boldsymbol{X}\boldsymbol{u}_t\|_{\infty} \le 2\,d\,\|\boldsymbol{X}\|_{2,\infty}\|\boldsymbol{X}\|_{1,\infty}\sum_{t=1}^{T}\|\boldsymbol{X}\boldsymbol{u}_t\|_{\infty}, \tag{54}$$

where in the last inequality we used Cauchy-Schwarz inequality. Similarly for $\text{Term}_{\text{II}}$ we can write:

$$\max_{\|\boldsymbol{v}\|=1} \sum_{i=1}^{d} \sum_{j=1}^{d} \sum_{t=1}^{T} \|\boldsymbol{q}_i\| \|\boldsymbol{q}_j\| \|\boldsymbol{\varphi}(\boldsymbol{g}_t)^\top \boldsymbol{X}_{:,j} (\boldsymbol{X}\boldsymbol{u}_t)^\top \boldsymbol{\varphi}'(\boldsymbol{g}_t)\boldsymbol{X}\|$$

$$\leq 2 \max_{\|\boldsymbol{v}\|=1} \sum_{i=1}^{d} \sum_{j=1}^{d} \sum_{t=1}^{T} \|\boldsymbol{q}_i\| \|\boldsymbol{q}_j\| \left|\boldsymbol{\varphi}(\boldsymbol{g}_t)^\top \boldsymbol{X}_{:,j}\right| \|\boldsymbol{X}\|_{2,\infty} \|\boldsymbol{X}\boldsymbol{u}_t\|_\infty \leq 2\,d\,\|\boldsymbol{X}\|_{2,\infty} \|\boldsymbol{X}\|_{1,\infty} \sum_{t=1}^{T} \|\boldsymbol{X}\boldsymbol{u}_t\|_\infty\,, \quad (55)$$

where we used Hölder's inequality in the last inequality. And at the end, for $\text{Term}_{\text{III}}$ we have:

$$\max_{\|\boldsymbol{v}\|=1} \sum_{i=1}^{d} \sum_{j=1}^{d} \sum_{t=1}^{T} \|\boldsymbol{q}_i\| \|\boldsymbol{q}_j\| \|\boldsymbol{\varphi}(\boldsymbol{g}_t)^\top \boldsymbol{X}\boldsymbol{u}_t \boldsymbol{X}_{:,j}^\top \boldsymbol{\varphi}'(\boldsymbol{g}_t)\boldsymbol{X}\|$$

$$\leq 2 \max_{\|\boldsymbol{v}\|=1} \sum_{i=1}^{d} \sum_{j=1}^{d} \sum_{t=1}^{T} \|\boldsymbol{q}_i\| \|\boldsymbol{q}_j\| \left|\boldsymbol{\varphi}(\boldsymbol{g}_t)^\top \boldsymbol{X}\boldsymbol{u}_t\right| \|\boldsymbol{X}\|_{2,\infty} \|\boldsymbol{X}_{:,j}\|_\infty \leq 2\,d\,\|\boldsymbol{X}\|_{2,\infty} \|\boldsymbol{X}\|_{1,\infty} \sum_{t=1}^{T} \|\boldsymbol{X}\boldsymbol{u}_t\|_\infty\,. \quad (56)$$

Then, we upper-bound $\text{Term}_2$:

$$2 \max_{\|\boldsymbol{v}\|=1} \sum_{j=1}^{d} \sum_{t=1}^{T} \boldsymbol{p}_t^\top \boldsymbol{x}_t \boldsymbol{X}_{:,j}^\top \boldsymbol{\varphi}'(\boldsymbol{g}_t)\boldsymbol{X}\boldsymbol{q}_j \leq 2 \max_{\|\boldsymbol{v}\|=1} \sum_{j=1}^{d} \sum_{t=1}^{T} \|\boldsymbol{p}_t\| \|\boldsymbol{q}_j\| \|\boldsymbol{X}\|_{2,\infty} \|\boldsymbol{X}_{:,j}^\top \boldsymbol{\varphi}'(\boldsymbol{g}_t)\boldsymbol{X}\|$$

$$\leq 4 \frac{\sqrt{T\,d}}{2} \|\boldsymbol{X}\|_{2,\infty}^2 \|\boldsymbol{X}\|_{1,\infty} = 2\sqrt{T\,d}\,\|\boldsymbol{X}\|_{2,\infty}^2 \|\boldsymbol{X}\|_{1,\infty}\,. \quad (57)$$

The last inequality comes from the calculation of the gradient concluded in Eq. (42). Plugging in Eqs. (54), (55), (56), and (57) in Eq. (52), we conclude that

$$\|\nabla_{\boldsymbol{\theta}}^2 \Phi(\boldsymbol{X};\boldsymbol{\theta})\| \leq 6\,d\,\|\boldsymbol{X}\|_{2,\infty}^2 \|\boldsymbol{X}\|_{1,\infty}^2 \sum_{t=1}^{T} \|\boldsymbol{X}\boldsymbol{u}_t\|_\infty + 2\sqrt{T\,d}\,\|\boldsymbol{X}\|_{2,\infty}^2 \|\boldsymbol{X}\|_{1,\infty}\,. \quad (58)$$

Applying Assumption 1, we have:

$$\|\nabla_{\boldsymbol{\theta}}^2 \Phi(\boldsymbol{X};\boldsymbol{\theta})\| \leq 6\,d\,R^4 \sum_{t=1}^{T} \|\boldsymbol{X}\boldsymbol{u}_t\|_\infty + 2\sqrt{T\,d}\,R^3\,, \quad (59)$$

or

$$\|\nabla_{\boldsymbol{\theta}}^2 \Phi(\boldsymbol{X};\boldsymbol{\theta})\| \leq 6\,d\,R^4 \sum_{t=1}^{T} \max_{\tau \in [T]} \boldsymbol{x}_\tau^\top \boldsymbol{u}_t + 2\sqrt{T\,d}\,R^3$$

$$\leq 6\,d\,R^4 \sum_{t=1}^{T} \max_{\tau \in [T]} \|\boldsymbol{x}_\tau\| \|\boldsymbol{u}_t\| + 2\sqrt{T\,d}\,R^3$$

$$\leq 6\sqrt{T}\,d\,R^5 \|\boldsymbol{U}\|_F + 2\sqrt{T\,d}\,R^3\,, \quad (60)$$

where in the second and the last inequalities we used Cauchy-Schwartz inequality.

$\square$

We now derive bounds for the multi-head attention model.

**Lemma 5.** *Recall the multihead model in Eq.* (2).

$$\widetilde{\Phi}(\boldsymbol{X};\widetilde{\boldsymbol{\theta}}) = \frac{1}{\sqrt{H}} \sum_{h=1}^{H} \langle \boldsymbol{U}_h, \boldsymbol{\varphi}(\boldsymbol{X}\boldsymbol{W}_h\boldsymbol{X}^\top)\boldsymbol{X}\rangle\,. \quad (61)$$

*The following are true under Assumption 1.*

*1.* $\|\nabla_{\widetilde{\boldsymbol{\theta}}}\widetilde{\Phi}(\boldsymbol{X};\widetilde{\boldsymbol{\theta}})\| \leq \sqrt{T}\,R\left(2\,R^2 \max_{h \in [H]} \|\boldsymbol{U}_h\|_F + 1\right).$

2. $\|\nabla_{\widetilde{\boldsymbol{\theta}}}^2 \widetilde{\Phi}(\boldsymbol{X}; \widetilde{\boldsymbol{\theta}})\| \leq \frac{2\sqrt{T d}\, R^3}{\sqrt{H}} \left(3\sqrt{d}\, R^2 \max_{h \in [H]} \|\boldsymbol{U}_h\|_F + 1\right).$

*Proof.* From Equation (46) for single-head attention,

$$\|\nabla_{\widetilde{\boldsymbol{\theta}}} \widetilde{\Phi}(\boldsymbol{X}; \widetilde{\boldsymbol{\theta}})\|^2 \leq \frac{1}{H} \sum_{h=1}^{H} \left(2\sqrt{T}\, R^3 \|\boldsymbol{U}_h\|_F + \sqrt{T}\, R\right)^2$$

$$\leq \left(2\sqrt{T}\, R^3 \max_{h \in [H]} \|\boldsymbol{U}_h\|_F + \sqrt{T}\, R\right)^2. \tag{62}$$

Denote $\boldsymbol{v}^\top = \begin{bmatrix} \boldsymbol{v}_1^\top & \cdots & \boldsymbol{v}_H^\top \end{bmatrix} \in \mathbb{R}^{H(Td + d^2)}$, where $\|\boldsymbol{v}\| = 1$. Using Equation (60),

$$\langle \boldsymbol{v}, \nabla_{\widetilde{\boldsymbol{\theta}}}^2 \widetilde{\Phi}(\boldsymbol{X}; \widetilde{\boldsymbol{\theta}}) \boldsymbol{v} \rangle \leq \frac{1}{\sqrt{H}} \left(6\sqrt{T}\, d\, R^5 \max_{h \in [H]} \|\boldsymbol{U}_h\|_F + 2\sqrt{T d}\, R^3\right) \sum_{h=1}^{H} \|\boldsymbol{v}_h\|^2$$

$$= \frac{1}{\sqrt{H}} \left(6\sqrt{T}\, d\, R^5 \max_{h \in [H]} \|\boldsymbol{U}_h\|_F + 2\sqrt{T d}\, R^3\right). \tag{63}$$

$\square$

## A.2 Proof of Corollary 1: Objective's Gradient/Hessian

Corollary 1 follows immediately by the result of Lemma 6 below and using a more relaxed bound $\max_{h \in [H]} \|\boldsymbol{U}_h\|_F \leq \|\widetilde{\boldsymbol{\theta}}\|_{2,\infty}$.

**Lemma 6** (Tight version of Corollary 1). *Let Assumption 1 hold and we use logistic loss function. Then, the following are true for the loss gradient and Hessian:*

1. $\|\nabla \widehat{L}(\widetilde{\boldsymbol{\theta}})\| \leq \sqrt{T}\, R \left(2\, R^2 \max_{h \in [H]} \|\boldsymbol{U}_h\|_F + 1\right) \widehat{L}(\widetilde{\boldsymbol{\theta}}).$

2. $\|\nabla^2 \widehat{L}(\widetilde{\boldsymbol{\theta}})\| \leq \frac{2\sqrt{T d}\, R^3}{\sqrt{H}} \left(3\sqrt{d}\, R^2 \max_{h \in [H]} \|\boldsymbol{U}_h\|_F + 1\right) + \frac{T R^2}{4} \left(2\, R^2 \max_{h \in [H]} \|\boldsymbol{U}_h\|_F + 1\right)^2.$

3. $\lambda_{min}(\nabla^2 \widehat{L}(\widetilde{\boldsymbol{\theta}})) \geq -\frac{2\sqrt{T d}\, R^3}{\sqrt{H}} \left(3\sqrt{d}\, R^2 \max_{h \in [H]} \|\boldsymbol{U}_h\|_F + 1\right) \widehat{L}(\widetilde{\boldsymbol{\theta}}).$

*Proof.* The loss gradient is derived as follows,

$$\nabla \widehat{L}(\widetilde{\boldsymbol{\theta}}) = \frac{1}{n} \sum_{i=1}^{n} \ell'(y_i \widetilde{\Phi}(\boldsymbol{X}_i; \widetilde{\boldsymbol{\theta}}))\, y_i\, \nabla_{\widetilde{\boldsymbol{\theta}}} \widetilde{\Phi}(\boldsymbol{X}_i; \widetilde{\boldsymbol{\theta}}).$$

Recalling that $y_i \in \{\pm 1\}$, we can write

$$\|\nabla \widehat{L}(\widetilde{\boldsymbol{\theta}})\| = \frac{1}{n} \|\sum_{i=1}^{n} \ell'(y_i \widetilde{\Phi}(\boldsymbol{X}_i; \widetilde{\boldsymbol{\theta}}))\, y_i\, \nabla_{\widetilde{\boldsymbol{\theta}}} \widetilde{\Phi}(\boldsymbol{X}_i; \widetilde{\boldsymbol{\theta}})\| \leq \frac{1}{n} \sum_{i=1}^{n} |\ell'(y_i \widetilde{\Phi}(\boldsymbol{X}_i; \widetilde{\boldsymbol{\theta}}))|\, \|\nabla_{\widetilde{\boldsymbol{\theta}}} \widetilde{\Phi}(\boldsymbol{X}_i; \widetilde{\boldsymbol{\theta}})\|.$$

Thus, using Lemma 5 to bound the norm of the model's gradient:

$$\|\nabla \widehat{L}(\widetilde{\boldsymbol{\theta}})\| \leq \sqrt{T}\, R \left(2\, R^2 \max_{h \in [H]} \|\boldsymbol{U}_h\|_F + 1\right) \widehat{L}(\widetilde{\boldsymbol{\theta}}). \tag{64}$$

For the Hessian of loss, note that

$$\nabla^2 \widehat{L}(\widetilde{\boldsymbol{\theta}}) = \frac{1}{n} \sum_{i=1}^{n} \ell''(y_i \widetilde{\Phi}(\boldsymbol{X}_i; \widetilde{\boldsymbol{\theta}}))\, \nabla_{\widetilde{\boldsymbol{\theta}}} \widetilde{\Phi}(\boldsymbol{X}_i; \widetilde{\boldsymbol{\theta}})\, \nabla_{\widetilde{\boldsymbol{\theta}}} \widetilde{\Phi}(\boldsymbol{X}_i; \widetilde{\boldsymbol{\theta}})^\top + \ell'(y_i \widetilde{\Phi}(\boldsymbol{X}_i; \widetilde{\boldsymbol{\theta}}))\, y_i\, \nabla_{\widetilde{\boldsymbol{\theta}}}^2 \widetilde{\Phi}(\boldsymbol{X}_i; \widetilde{\boldsymbol{\theta}}). \tag{65}$$

It follows that

$$
\|\nabla^2 \widehat{L}(\widetilde{\boldsymbol{\theta}})\| = \|\frac{1}{n} \sum_{i=1}^{n} \ell''(y_i \widetilde{\Phi}(\boldsymbol{X}_i; \widetilde{\boldsymbol{\theta}})) \nabla_{\widetilde{\boldsymbol{\theta}}} \widetilde{\Phi}(\boldsymbol{X}_i; \widetilde{\boldsymbol{\theta}}) \nabla_{\widetilde{\boldsymbol{\theta}}} \widetilde{\Phi}(\boldsymbol{X}_i; \widetilde{\boldsymbol{\theta}})^\top + \ell'(y_i \Phi(\boldsymbol{X}_i; \widetilde{\boldsymbol{\theta}})) y_i \nabla_{\widetilde{\boldsymbol{\theta}}}^2 \widetilde{\Phi}(\boldsymbol{X}_i; \widetilde{\boldsymbol{\theta}})\|
$$

$$
\leq \frac{1}{n} \sum_{i=1}^{n} |\ell'(y_i \widetilde{\Phi}(\boldsymbol{X}_i; \widetilde{\boldsymbol{\theta}}))| \, \|\nabla_{\widetilde{\boldsymbol{\theta}}}^2 \widetilde{\Phi}(\boldsymbol{X}_i; \widetilde{\boldsymbol{\theta}})\| + |\ell''(y_i \widetilde{\Phi}(\boldsymbol{X}_i; \widetilde{\boldsymbol{\theta}}))| \, \|\nabla_{\widetilde{\boldsymbol{\theta}}} \widetilde{\Phi}(\boldsymbol{X}_i; \widetilde{\boldsymbol{\theta}})\|^2
$$

$$
\leq \frac{2\sqrt{T d}\, R^3}{\sqrt{H}} \left( 3\sqrt{d}\, R^2 \max_{h \in [H]} \|\boldsymbol{U}_h\|_F + 1 \right) + \frac{T R^2}{4} \left( 2 R^2 \max_{h \in [H]} \|\boldsymbol{U}_h\|_F + 1 \right)^2 . \tag{66}
$$

To lower-bound the minimum eigenvalue of the Hessian of loss, note that $\ell(\cdot)$ is convex and thus $\ell''(\cdot) \geq 0$. Therefore, the first term in (65) is positive semi-definite and the second term can be lower-bounded as follows,

$$
\lambda_{\min}(\nabla^2 \widehat{L}(\widetilde{\boldsymbol{\theta}})) \geq -\frac{1}{n} \sum_{i=1}^{n} \|\ell'(y_i \widetilde{\Phi}(\boldsymbol{X}_i; \widetilde{\boldsymbol{\theta}})) y_i \nabla_{\widetilde{\boldsymbol{\theta}}}^2 \widetilde{\Phi}(\boldsymbol{X}_i; \widetilde{\boldsymbol{\theta}})\|
$$

$$
\geq -\frac{1}{n} \sum_{i=1}^{n} |\ell'(y_i \widetilde{\Phi}(\boldsymbol{X}_i; \widetilde{\boldsymbol{\theta}}))| \, \|\nabla_{\widetilde{\boldsymbol{\theta}}}^2 \widetilde{\Phi}(\boldsymbol{X}_i; \widetilde{\boldsymbol{\theta}})\|
$$

$$
\geq -\frac{2\sqrt{T d}\, R^3}{\sqrt{H}} \left( 3\sqrt{d}\, R^2 \max_{h \in [H]} \|\boldsymbol{U}_h\|_F + 1 \right) \widehat{L}(\widetilde{\boldsymbol{\theta}}) .
$$

$\square$

## B  Training Analysis

### B.1  Preliminaries

Throughout this section we drop the $\widetilde{\phantom{x}}$ in $\widetilde{\boldsymbol{\theta}}$ and $\widetilde{\Phi}(\boldsymbol{X}_i; \widetilde{\boldsymbol{\theta}})$ as everything refers to the full model. Moreover, $\widetilde{\boldsymbol{\theta}}^{(K)}$ and $\widetilde{\boldsymbol{\theta}}^{(0)}$ are denoted by $\boldsymbol{\theta}_K$ and $\boldsymbol{\theta}_0$.

The proof of both the training and generalization analysis follows the high-level steps outlined in Taheri & Thrampoulidis (2023). However, our analysis focuses on the self-attention model, which differs from the two-layer perceptron studied in the referenced work. Notably, we train both the attention weights and the classifier head, whereas Taheri & Thrampoulidis (2023) assumes fixed outer layer weights. This introduces a new challenge as the smoothness and curvature of the objective function at point $\boldsymbol{\theta}$ become dependent on $\boldsymbol{\theta}$ itself (see Corollary 1). Consequently, we make careful adjustments in the proof to account for this challenge.

Our analysis critically uses the following property of the loss objective from Corollary 1: $\forall \boldsymbol{\theta} : \lambda_{\min}\left(\nabla^2 \widehat{L}(\boldsymbol{\theta})\right) \geq -\kappa(\boldsymbol{\theta}) \cdot \widehat{L}(\boldsymbol{\theta})$, $\kappa(\boldsymbol{\theta}) \coloneqq \frac{\beta_3(\boldsymbol{\theta})}{\sqrt{H}}$. Note from the definition of $\beta_3(\cdot)$ that $\forall \boldsymbol{\theta}_1, \boldsymbol{\theta}_2 : \max_{\boldsymbol{\theta} \in [\boldsymbol{\theta}_1, \boldsymbol{\theta}_2]} \beta_3(\boldsymbol{\theta}) = \beta_3(\boldsymbol{\theta}_1) \vee \beta_3(\boldsymbol{\theta}_2)$. Thus, the above property of the loss implies the following *local self-bounded weak convexity* property on the line $[\boldsymbol{\theta}_1, \boldsymbol{\theta}_2]$:

$$
\forall \boldsymbol{\theta}_1, \boldsymbol{\theta}_2 : \min_{\boldsymbol{\theta} \in [\boldsymbol{\theta}_1, \boldsymbol{\theta}_2]} \lambda_{\min}\left(\nabla^2 \widehat{L}(\boldsymbol{\theta})\right) \geq -\frac{\beta_3(\boldsymbol{\theta}_1) \vee \beta_3(\boldsymbol{\theta}_2)}{\sqrt{H}} \cdot \widehat{L}(\boldsymbol{\theta}). \tag{67}
$$

In turn, Equation (67) can be used to prove that the loss satisfies a generalized local quasi-convexity (GLQC) property as formalized in the proposition below. The proposition is only a slight modification of (Taheri & Thrampoulidis, 2023, Prop. 8). While, a direct application of their result is not possible since the self-bounded weak convexity property in (67) holds only locally on the line $[\boldsymbol{\theta}_1, \boldsymbol{\theta}_2]$, an inspection of their proof shows that this is sufficient.

### B.2  Proof of Proposition 4

First, we restate the proposition below for the reader's convenience.

**Proposition 6** (GLQC property). *Let $\boldsymbol{\theta}_1, \boldsymbol{\theta}_2$ be points sufficiently close to each other, such that*

$$
2 \left( \beta_3(\boldsymbol{\theta}_1) \vee \beta_3(\boldsymbol{\theta}_2) \right) \|\boldsymbol{\theta}_1 - \boldsymbol{\theta}_2\|^2 \leq \sqrt{H} .
$$

*Then, the following generalized local quasi-convexity (GLQC) property holds:*

$$\max_{\boldsymbol{\theta} \in [\boldsymbol{\theta}_1, \boldsymbol{\theta}_2]} \widehat{L}(\boldsymbol{\theta}) \le \frac{4}{3} \left( \widehat{L}(\boldsymbol{\theta}_1) \vee \widehat{L}(\boldsymbol{\theta}_2) \right).$$

*Proof.* Although the loss $\widehat{L}$ is not uniformly self-bounded weakly convex as assumed in (Taheri & Thrampoulidis, 2023, Prop. 8), an inspection of their proof shows that for every $\boldsymbol{\theta}_1, \boldsymbol{\theta}_2$ it suffices that the loss is locally self-bounded weakly convex. With this observation, we can apply their proposition with $\kappa \leftarrow \kappa(\boldsymbol{\theta}_1) \vee \kappa(\boldsymbol{\theta}_2)$, which gives the desired. $\square$

## B.3 Key Lemmas

The proof of Theorem 1 consists of several intermediate lemma, which we state and prove in this section.

**Lemma 7** (Descent Lemma). *Let Assumption 1 hold. Then, for any iteration $k \ge 0$ we have step-wise descent:*

$$\widehat{L}(\boldsymbol{\theta}_{k+1}) \le \widehat{L}(\boldsymbol{\theta}_k) - \frac{\eta}{2} \left\| \nabla \widehat{L}(\boldsymbol{\theta}_k) \right\|^2, \tag{68}$$

*provided $\eta \le \frac{1}{\rho_k}$ where $\rho_k$ is the objective's local smoothness parameter defined as below,*

$$\rho_k := \beta_2(\boldsymbol{\theta}_k) \vee \beta_2(\boldsymbol{\theta}_{k+1}).$$

*Proof.* By Taylor's expansion, there exists a $\boldsymbol{\theta}' \in [\boldsymbol{\theta}_k, \boldsymbol{\theta}_{k+1}]$ such that,

$$\hat{L}(\boldsymbol{\theta}_{k+1}) = \widehat{L}(\boldsymbol{\theta}_k) + \left\langle \nabla \widehat{L}(\boldsymbol{\theta}_k), \boldsymbol{\theta}_{k+1} - \boldsymbol{\theta}_k \right\rangle + \frac{1}{2} \left\langle \boldsymbol{\theta}_{k+1} - \boldsymbol{\theta}_k, \nabla^2 \widehat{L}(\boldsymbol{\theta}') (\boldsymbol{\theta}_{k+1} - \boldsymbol{\theta}_k) \right\rangle$$

$$\le \widehat{L}(\boldsymbol{\theta}_k) + \left\langle \nabla \widehat{L}(\boldsymbol{\theta}_k), \boldsymbol{\theta}_{k+1} - \boldsymbol{\theta}_k \right\rangle + \frac{1}{2} \max_{\boldsymbol{\theta}' \in [\boldsymbol{\theta}_k, \boldsymbol{\theta}_{k+1}]} \left\| \nabla^2 \widehat{L}(\boldsymbol{\theta}') \right\| \cdot \left\| \boldsymbol{\theta}_{k+1} - \boldsymbol{\theta}_k \right\|^2$$

$$\le \widehat{L}(\boldsymbol{\theta}_k) - \eta \left\| \nabla \widehat{L}(\boldsymbol{\theta}_k) \right\|^2 + \frac{\eta^2 \rho_k}{2} \cdot \left\| \nabla \widehat{L}(\boldsymbol{\theta}_k) \right\|^2,$$

where the last step follows from Corollary 1, and using $\max_{\boldsymbol{\theta}' \in [\boldsymbol{\theta}_k, \boldsymbol{\theta}_{k+1}]} \beta_2(\boldsymbol{\theta}') = \beta_2(\boldsymbol{\theta}) \vee \beta_2(\boldsymbol{\theta}_k) = \rho_k$. For $\eta \le \frac{1}{\rho_k}$, we conclude the claim. $\square$

**Lemma 8.** *Assume $\eta > 0$ such that step-wise descent (68) holds for all $k \in [K-1]$. Then, for any $\boldsymbol{\theta}$, the following holds:*

$$\frac{1}{K} \sum_{k=1}^{K} \widehat{L}(\boldsymbol{\theta}_k) \le \widehat{L}(\boldsymbol{\theta}) + \frac{\|\boldsymbol{\theta} - \boldsymbol{\theta}_0\|^2}{2\eta K} + \frac{1}{2K} \sum_{k=0}^{K-1} \tau_k \|\boldsymbol{\theta} - \boldsymbol{\theta}_k\|^2, \tag{69}$$

*where $\tau_k := \frac{1}{\sqrt{H}} \left( \beta_3(\boldsymbol{\theta}) \vee \beta_3(\boldsymbol{\theta}_k) \right) \max_{\alpha \in [0,1]} \widehat{L}(\boldsymbol{\theta}_{k_\alpha})$.*

*Proof.* By Taylor's theorem, for $\boldsymbol{\theta}_{k_\alpha} := \alpha \boldsymbol{\theta}_k + (1 - \alpha) \boldsymbol{\theta}, \ \alpha \in [0, 1]$ we know that:

$$\widehat{L}(\boldsymbol{\theta}) \ge \widehat{L}(\boldsymbol{\theta}_k) + \langle \nabla \widehat{L}(\boldsymbol{\theta}_k), \boldsymbol{\theta} - \boldsymbol{\theta}_k \rangle + \frac{1}{2} \lambda_{\min} \left( \nabla^2 \widehat{L}(\boldsymbol{\theta}_{k_\alpha}) \right) \|\boldsymbol{\theta} - \boldsymbol{\theta}_k\|^2.$$

Using (68), we get:

$$\widehat{L}(\boldsymbol{\theta}_{k+1}) \le \widehat{L}(\boldsymbol{\theta}) + \langle \nabla \widehat{L}(\boldsymbol{\theta}_k), \boldsymbol{\theta}_k - \boldsymbol{\theta} \rangle - \frac{1}{2} \lambda_{\min} \left( \nabla^2 \widehat{L}(\boldsymbol{\theta}_{k_\alpha}) \right) \|\boldsymbol{\theta} - \boldsymbol{\theta}_k\|^2 - \frac{\eta}{2} \| \nabla \widehat{L}(\boldsymbol{\theta}_k) \|^2$$

$$\le \widehat{L}(\boldsymbol{\theta}) + \frac{\|\boldsymbol{\theta} - \boldsymbol{\theta}_k\|^2}{2\eta} - \frac{\|\boldsymbol{\theta} - \boldsymbol{\theta}_{k+1}\|^2}{2\eta} + \frac{1}{2} \tau_k \|\boldsymbol{\theta} - \boldsymbol{\theta}_k\|^2, \tag{70}$$

where the last step follows by completion of squares using $\boldsymbol{\theta}_{k+1} - \boldsymbol{\theta}_k = -\eta \nabla \widehat{L}(\boldsymbol{\theta}_k)$, and also uses Corollary 1 with $\tau_k := \max_{\alpha \in [0,1]} \tau_{k_\alpha}$, where $\tau_{k_\alpha} = \frac{\beta_3(\boldsymbol{\theta}_{k_\alpha})}{\sqrt{H}} \widehat{L}(\boldsymbol{\theta}_{k_\alpha})$. Telescoping in (70) for $k = 0, ..., K - 1$, we get the desired.

$\square$

Next, we use the generalized local quasi-convexity property (Proposition 4) to obtain explicit regret bound from Lemma 8.

**Lemma 9.** *Suppose the assumptions of Lemma 8 hold. Moreover, assume for all $k \in [K-1]$ it holds that $\sqrt{H} \geq 2\left(\beta_3(\boldsymbol{\theta}) \vee \beta_3(\boldsymbol{\theta}_k)\right) \|\boldsymbol{\theta} - \boldsymbol{\theta}_k\|^2$. Then,*

$$\frac{1}{K} \sum_{k=1}^{K} \widehat{L}(\boldsymbol{\theta}_k) \leq 2\widehat{L}(\boldsymbol{\theta}) + \frac{3\|\boldsymbol{\theta} - \boldsymbol{\theta}_0\|^2}{4\eta K} + \frac{\widehat{L}(\boldsymbol{\theta}_0)}{2K}. \tag{71}$$

*Proof.* We have $\min_{\boldsymbol{\theta}' \in [\boldsymbol{\theta}, \boldsymbol{\theta}_k]} \lambda_{\min}(\nabla^2 \widehat{L}(\boldsymbol{\theta}')) \geq -\frac{\beta_3(\boldsymbol{\theta}) \vee \beta_3(\boldsymbol{\theta}_k)}{\sqrt{H}} \max_{\boldsymbol{\theta}' \in [\boldsymbol{\theta}, \boldsymbol{\theta}_k]} \widehat{L}(\boldsymbol{\theta}')$ from Corollary 1. Thus, using Proposition 4, we can control $\max_{\alpha \in [0,1]} \widehat{L}(\boldsymbol{\theta}_{k_\alpha})$. Specifically, by assumption we have for all $k \in [K-1]$ that

$$\sqrt{H} \geq 2\left(\beta_3(\boldsymbol{\theta}) \vee \beta_3(\boldsymbol{\theta}_k)\right) \|\boldsymbol{\theta} - \boldsymbol{\theta}_k\|^2 = 2 \max_{\alpha \in [0,1]} \beta_3(\boldsymbol{\theta}_{k_\alpha}) \|\boldsymbol{\theta} - \boldsymbol{\theta}_k\|^2.$$

Then, by Proposition 4, we have

$$\max_{\alpha \in [0,1]} \widehat{L}(\boldsymbol{\theta}_{k_\alpha}) \leq \frac{4}{3} \max\{\widehat{L}(\boldsymbol{\theta}_k), \widehat{L}(\boldsymbol{\theta})\} \leq \frac{4}{3}\widehat{L}(\boldsymbol{\theta}_k) + \frac{4}{3}\widehat{L}(\boldsymbol{\theta}).$$

Thus, applying this to (69) we have:

$$\begin{aligned}
\frac{1}{K} \sum_{k=1}^{K} \widehat{L}(\boldsymbol{\theta}_k) &\leq \widehat{L}(\boldsymbol{\theta}) + \frac{\|\boldsymbol{\theta} - \boldsymbol{\theta}_0\|^2}{2\eta K} + \frac{2}{3K} \sum_{k=0}^{K-1} \frac{1}{\sqrt{H}} \left(\beta_3(\boldsymbol{\theta}) \vee \beta_3(\boldsymbol{\theta}_k)\right) \left(\widehat{L}(\boldsymbol{\theta}_k) + \widehat{L}(\boldsymbol{\theta})\right) \|\boldsymbol{\theta} - \boldsymbol{\theta}_k\|^2 \\
&\leq \widehat{L}(\boldsymbol{\theta}) + \frac{\|\boldsymbol{\theta} - \boldsymbol{\theta}_0\|^2}{2\eta K} + \frac{1}{3K} \sum_{k=0}^{K-1} \left(\widehat{L}(\boldsymbol{\theta}_k) + \widehat{L}(\boldsymbol{\theta})\right) \\
&\leq \frac{4}{3}\widehat{L}(\boldsymbol{\theta}) + \frac{\|\boldsymbol{\theta} - \boldsymbol{\theta}_0\|^2}{2\eta K} + \frac{1}{3K} \sum_{k=0}^{K} \widehat{L}(\boldsymbol{\theta}_k),
\end{aligned} \tag{72}$$

where we use the condition on $H$ in the second inequality. Rearranging terms above we conclude the claim of the lemma. $\qquad \square$

**Lemma 10** (Iterates-norm bound). *Suppose the descent property (68) holds $\forall k \in [K-1]$, and let Assumption 1 hold. Further, assume that*

$$\|\boldsymbol{\theta} - \boldsymbol{\theta}_0\|^2 \geq \max\{\eta K \widehat{L}(\boldsymbol{\theta}), \eta \widehat{L}(\boldsymbol{\theta}_0)\}. \tag{73}$$

*and*

$$\sqrt{H} \geq 36\sqrt{T d}\, R^3 \left(3\sqrt{d}\, R^2 \left(3\|\boldsymbol{\theta} - \boldsymbol{\theta}_0\| + \|\boldsymbol{\theta}\|_{2,\infty}\right) + 1\right) \|\boldsymbol{\theta} - \boldsymbol{\theta}_0\|^2, \tag{74}$$

*Then, for all $k \in [K]$,*

$$\|\boldsymbol{\theta}_k - \boldsymbol{\theta}\| \leq 3\|\boldsymbol{\theta} - \boldsymbol{\theta}_0\|. \tag{75}$$

*Proof.* Denote $A_k = \|\boldsymbol{\theta}_k - \boldsymbol{\theta}\|$. Start by recalling from Eq. (70) that for all $k$:

$$A_{k+1}^2 \leq A_k^2 + 2\eta \widehat{L}(\boldsymbol{\theta}) - 2\eta \widehat{L}(\boldsymbol{\theta}_{k+1}) + \eta\left(\max_{\alpha \in [0,1]} \tau_{k_\alpha}\right) A_k^2, \tag{76}$$

where recall that $\tau_{k_\alpha} = \frac{\beta_3(\boldsymbol{\theta}_{k_\alpha})\widehat{L}(\boldsymbol{\theta}_{k_\alpha})}{\sqrt{H}}$. We will prove the desired statement (75) using induction. For $k = 0$, $A_0 = \|\boldsymbol{\theta} - \boldsymbol{\theta}_0\|$. Thus, the assumption of induction holds. Now assume (75) is correct for $j \in [k-1]$, i.e. $A_j \leq 3\|\boldsymbol{\theta} - \boldsymbol{\theta}_0\|, \forall j \in [k-1]$. We will then prove it holds for $k$.

By induction hypothesis for all $j \in [k-1]$, and all $\alpha \in [0,1]$:

$$\sqrt{H} \geq 36 \sqrt{T d} R^3 \left(3 \sqrt{d} R^2 \left(3\|\boldsymbol{\theta} - \boldsymbol{\theta}_0\| + \|\boldsymbol{\theta}\|_{2,\infty}\right) + 1\right) \|\boldsymbol{\theta} - \boldsymbol{\theta}_0\|^2$$

$$\geq 4 \sqrt{T d} R^3 \left(3 \sqrt{d} R^2 \left(3\|\boldsymbol{\theta} - \boldsymbol{\theta}_0\| + \|\boldsymbol{\theta}\|_{2,\infty}\right) + 1\right) \|\boldsymbol{\theta} - \boldsymbol{\theta}_j\|^2$$

$$\geq 4 \sqrt{T d} R^3 \left(3 \sqrt{d} R^2 \left(\|\boldsymbol{\theta} - \boldsymbol{\theta}_j\|_{2,\infty} + \|\boldsymbol{\theta}\|_{2,\infty}\right) + 1\right) \|\boldsymbol{\theta} - \boldsymbol{\theta}_j\|^2$$

$$\geq 4 \sqrt{T d} R^3 \left(3 \sqrt{d} R^2 \|\boldsymbol{\theta}_{j_\alpha}\|_{2,\infty} + 1\right) \|\boldsymbol{\theta} - \boldsymbol{\theta}_j\|^2 = 2\beta_3(\boldsymbol{\theta}_{j_\alpha}) \|\boldsymbol{\theta} - \boldsymbol{\theta}_j\|^2 .$$

Thus, by Proposition 4, $\forall j \in [k-1]$

$$\max_{\alpha \in [0,1]} \widehat{L}(\boldsymbol{\theta}_{j_\alpha}) \leq \frac{4}{3} \widehat{L}(\boldsymbol{\theta}_j) + \frac{4}{3} \widehat{L}(\boldsymbol{\theta}) . \tag{77}$$

Using this in (76) we find for all $j \in [k-1]$,

$$A_{j+1}^2 \leq A_j^2 + 2\eta \widehat{L}(\boldsymbol{\theta}) - 2\eta \widehat{L}(\boldsymbol{\theta}_{j+1}) + \eta \frac{\max_{\alpha \in [0,1]} \beta_3(\boldsymbol{\theta}_{j_\alpha}) A_j^2}{\sqrt{H}} \left(\frac{4}{3} \widehat{L}(\boldsymbol{\theta}_j) + \frac{4}{3} \widehat{L}(\boldsymbol{\theta})\right)$$

$$\leq A_j^2 + 2\eta \widehat{L}(\boldsymbol{\theta}) - 2\eta \widehat{L}(\boldsymbol{\theta}_{j+1}) + \eta \left(\frac{2}{3} \widehat{L}(\boldsymbol{\theta}_j) + \frac{2}{3} \widehat{L}(\boldsymbol{\theta})\right) ,$$

where in the second inequality we used (77). We proceed by telescoping the above display over $j = 0, 1, \ldots, k-1$ to get

$$A_k^2 \leq A_0^2 + \frac{8}{3} \eta k \widehat{L}(\boldsymbol{\theta}) + \frac{2}{3} \eta \widehat{L}(\boldsymbol{\theta}_0) - \frac{4}{3} \eta \sum_{j=1}^{k-1} \widehat{L}(\boldsymbol{\theta}_j) - 2\eta \widehat{L}(\boldsymbol{\theta}_k)$$

$$\leq A_0^2 + \frac{8}{3} \eta k \widehat{L}(\boldsymbol{\theta}) + \frac{2}{3} \eta \widehat{L}(\boldsymbol{\theta}_0)$$

$$\leq \|\boldsymbol{\theta} - \boldsymbol{\theta}_0\|^2 + \frac{8}{3} \|\boldsymbol{\theta} - \boldsymbol{\theta}_0\|^2 + \frac{2}{3} \|\boldsymbol{\theta} - \boldsymbol{\theta}_0\|^2$$

$$\leq 9 \|\boldsymbol{\theta} - \boldsymbol{\theta}_0\|^2 , \tag{78}$$

where the second line follows by non-negativity of the loss. Thus, $A_k \leq 3\|\boldsymbol{\theta} - \boldsymbol{\theta}_0\|$. This completes the induction and proves the lemma. $\qquad \square$

## B.4  Proof of Theorem 1

We restate the theorem here for convenience, this time also including exact constants.

**Theorem 3** (Restatement of Thm. 1). *Fix any $\boldsymbol{\theta}$ and $H$ satisfying*

$$\sqrt{H} \geq 36 \sqrt{T d} R^3 \left(3 \sqrt{d} R^2 \left(3\|\boldsymbol{\theta} - \boldsymbol{\theta}_0\| + \|\boldsymbol{\theta}\|_{2,\infty}\right) + 1\right) \|\boldsymbol{\theta} - \boldsymbol{\theta}_0\|^2.$$

*Further, denote for convenience*

$$\alpha(\boldsymbol{\theta}) := 3 \sqrt{d} R^2 \left[3 \sqrt{T} R^3 \left(3\|\boldsymbol{\theta} - \boldsymbol{\theta}_0\| + \|\boldsymbol{\theta}\|_{2,\infty}\right) + 2 \sqrt{T} R\right]$$

*and $\rho(\boldsymbol{\theta}) = \left(\frac{2\sqrt{Td} R^3}{\sqrt{H}} + \frac{T R^2}{4}\right) \alpha(\boldsymbol{\theta})^2$. Then, for any step-size $\eta \leq 1 \wedge 1/\rho(\boldsymbol{\theta}) \wedge \frac{\|\boldsymbol{\theta}-\boldsymbol{\theta}_0\|^2}{K \widehat{L}(\boldsymbol{\theta})} \wedge \frac{\|\boldsymbol{\theta}-\boldsymbol{\theta}_0\|^2}{\widehat{L}(\boldsymbol{\theta}_0)}$, the following bounds hold for the training loss and the weights' norm at iteration $K$ of GD:*

$$\widehat{L}(\boldsymbol{\theta}_K) \leq \frac{1}{K} \sum_{k=1}^{K} \widehat{L}(\boldsymbol{\theta}_k) \leq 2\widehat{L}(\boldsymbol{\theta}) + \frac{5\|\boldsymbol{\theta} - \boldsymbol{\theta}_0\|^2}{4\eta K} \qquad and \qquad \|\boldsymbol{\theta}_K - \boldsymbol{\theta}_0\| \leq 4\|\boldsymbol{\theta} - \boldsymbol{\theta}_0\|. \tag{79}$$

*Proof.* Define

$$\alpha_\eta(\boldsymbol{\theta}) = 3\sqrt{d}\,R^2\left[\left(2\,\eta\,\sqrt{T}\,R^3 + 1\right)\left(3\,\|\boldsymbol{\theta} - \boldsymbol{\theta}_0\| + \|\boldsymbol{\theta}\|_{2,\infty}\right) + \eta\,\sqrt{T}\,R + 1\right],$$

and

$$\rho_\eta(\boldsymbol{\theta}) := \left(\frac{2\sqrt{T\,d}\,R^3}{\sqrt{H}} + \frac{T\,R^2}{4}\right)\alpha_\eta(\boldsymbol{\theta})^2\,.$$

From Lemma 7 recall that:

$$\rho_k := \beta_2(\boldsymbol{\theta}_k) \vee \beta_2(\boldsymbol{\theta}_{k+1})\,.$$

Now, recalling the definition of $\beta_2(\boldsymbol{\theta})$ from Corollary 1, we see that $\rho_k$, the objective's smoothness parameter at step $k$ depends on $\|\boldsymbol{\theta}_k\|_{2,\infty} \vee \|\boldsymbol{\theta}_{k+1}\|_{2,\infty}$, where:

$$
\begin{aligned}
\|\boldsymbol{\theta}_k\|_{2,\infty} \vee \|\boldsymbol{\theta}_{k+1}\|_{2,\infty} &\leq \left(\|\boldsymbol{\theta} - \boldsymbol{\theta}_k\| + \|\boldsymbol{\theta}\|_{2,\infty}\right) \vee \left(\|\boldsymbol{\theta} - \boldsymbol{\theta}_{k+1}\| + \|\boldsymbol{\theta}\|_{2,\infty}\right) \\
&\leq \left(\|\boldsymbol{\theta} - \boldsymbol{\theta}_k\| + \|\boldsymbol{\theta}\|_{2,\infty}\right) \vee \left(\|\boldsymbol{\theta} - \boldsymbol{\theta}_k\| + \|\boldsymbol{\theta}_{k+1} - \boldsymbol{\theta}_k\| + \|\boldsymbol{\theta}\|_{2,\infty}\right) \\
&= \|\boldsymbol{\theta} - \boldsymbol{\theta}_k\| + \|\boldsymbol{\theta}_{k+1} - \boldsymbol{\theta}_k\| + \|\boldsymbol{\theta}\|_{2,\infty} \\
&\leq \eta\sqrt{T}R\left(2R^2\|\boldsymbol{\theta}_k\|_{2,\infty} + 1\right) + \|\boldsymbol{\theta} - \boldsymbol{\theta}_k\| + \|\boldsymbol{\theta}\|_{2,\infty} \\
&\leq \left(2\eta\sqrt{T}R^3 + 1\right)\left(\|\boldsymbol{\theta} - \boldsymbol{\theta}_k\| + \|\boldsymbol{\theta}\|_{2,\infty}\right) + \eta\sqrt{T}R =: \Theta_k\,,
\end{aligned}
$$

where the second-last inequality follows from Corollary 1. For each $\rho_k$, define corresponding $\rho_\eta(\Theta_k) := \frac{2\sqrt{T\,d}\,R^3}{\sqrt{H}}\left(3\sqrt{d}\,R^2\,\Theta_k + 1\right) + \frac{T\,R^2}{4}\left(2\,R^2\,\Theta_k + 1\right)^2$. Consider $\rho_0$, it is easy to see that $\rho_0 \leq \rho_\eta(\Theta_0) \leq \rho_\eta(\boldsymbol{\theta})$. Hence, the descent property of GD holds in first iteration as per Lemma 7. Since, for $\eta \leq 1$, $\rho_\eta(\boldsymbol{\theta}) \leq \rho(\boldsymbol{\theta})$. Thus, $\eta \leq 1 \vee \frac{1}{\rho(\boldsymbol{\theta})} \implies \eta \leq \frac{1}{\rho(\boldsymbol{\theta})} \leq \frac{1}{\rho_\eta(\boldsymbol{\theta})} \leq \frac{1}{\rho_\eta(\Theta_0)} \leq \frac{1}{\rho_0}$. Moreover, note that the assumptions of Lemma 10 are satisfied. Thus, by induction over Lemmas 9-10 and noting that $\rho_k \leq \rho_\eta(\Theta_k) \leq \rho_\eta(\boldsymbol{\theta})$ for all $k \in [K-1]$ by using a similar argument as above, we obtain for any $\eta \leq \frac{1}{\rho(\boldsymbol{\theta})}$,

$$\forall k \in [K] \quad : \quad \|\boldsymbol{\theta}_k - \boldsymbol{\theta}\| \leq 3\|\boldsymbol{\theta} - \boldsymbol{\theta}_0\|,$$

$$\text{and} \quad \frac{1}{K}\sum_{k=1}^{K}\widehat{L}(\boldsymbol{\theta}_k) \leq 2\widehat{L}(\boldsymbol{\theta}) + \frac{3\|\boldsymbol{\theta} - \boldsymbol{\theta}_0\|^2}{4\eta K} + \frac{\widehat{L}(\boldsymbol{\theta}_0)}{2K}\,. \tag{80}$$

Moreover, by assumptions of the theorem we immediately find that $\frac{1}{2K}\widehat{L}(\boldsymbol{\theta}_0) \leq \frac{\|\boldsymbol{\theta} - \boldsymbol{\theta}_0\|^2}{2\eta K}$. We also have $\|\boldsymbol{\theta}_k - \boldsymbol{\theta}_0\| \leq \|\boldsymbol{\theta} - \boldsymbol{\theta}_k\| + \|\boldsymbol{\theta} - \boldsymbol{\theta}_0\| \leq 4\|\boldsymbol{\theta} - \boldsymbol{\theta}_0\|$. This completes the proof. $\qquad\square$

The training proof is summarized in Figure 1.

## B.5 Corollary 3

**Corollary 3** (Training loss under realizability). *Let Assumptions 1 and 2 hold. Fix $K \geq 1$. Assume any $H$ such that*

$$\sqrt{H} \geq 36\sqrt{T\,d}\,R^3\left(3\sqrt{d}\,R^2\left(3g_0(\frac{1}{K}) + g(\frac{1}{K})\right) + 1\right)g_0(\frac{1}{K})^2\,. \tag{81}$$

*Further, denote for convenience*

$$\alpha(K) := 3\sqrt{d}\,R^2\left[3\sqrt{T}\,R^3\left(3g_0(\frac{1}{K}) + g(\frac{1}{K})\right) + 2\sqrt{T}\,R\right]$$

*and $\rho(K) = \left(\frac{2\sqrt{T\,d}\,R^3}{\sqrt{H}} + \frac{T\,R^2}{4}\right)\alpha(K)^2$. Then, for any step-size $\eta \leq 1 \wedge 1/\rho(K) \wedge g_0(1)^2 \wedge \frac{g_0(1)^2}{\widehat{L}(\boldsymbol{\theta}_0)}$, the following bounds hold for the weights' norm and objective at iteration $K$ of GD:*

$$\widehat{L}(\boldsymbol{\theta}_K) \leq \frac{2}{K} + \frac{5g_0(\frac{1}{K})^2}{4\eta K} \qquad \text{and} \qquad \|\boldsymbol{\theta}_K - \boldsymbol{\theta}_0\| \leq 4g_0(\frac{1}{K})\,. \tag{82}$$

Figure 1: Training proof schema.

*Proof.* According to Assumption 2, for any sufficiently small $\varepsilon > 0$, there exists a $\boldsymbol{\theta}^{(\varepsilon)}$ such that $\widehat{L}(\boldsymbol{\theta}^{(\varepsilon)}) \le \varepsilon$ and $\|\boldsymbol{\theta}^{(\varepsilon)} - \boldsymbol{\theta}_0\| = g_0(\varepsilon)$. Pick $\varepsilon = \frac{1}{K}$. Let the step-size $\eta > 0$, satisfy the assumption of Descent Lemma 7. Since $\widehat{L}(\boldsymbol{\theta}^{(1/K)}) \le \frac{1}{K}$, we have:

$$\frac{\|\boldsymbol{\theta}^{(1/K)} - \boldsymbol{\theta}_0\|^2}{K\widehat{L}(\boldsymbol{\theta}^{(1/K)})} \ge \|\boldsymbol{\theta}^{(1/K)} - \boldsymbol{\theta}_0\|^2 = g_0\left(\frac{1}{K}\right)^2 \ge g_0(1)^2,$$

and

$$\frac{\|\boldsymbol{\theta}^{(1/K)} - \boldsymbol{\theta}_0\|^2}{\widehat{L}(\boldsymbol{\theta}_0)} = \frac{g_0\left(\frac{1}{K}\right)^2}{\widehat{L}(\boldsymbol{\theta}_0)} \ge \frac{g_0(1)^2}{\widehat{L}(\boldsymbol{\theta}_0)}.$$

Therefore, following our assumption on step-size $\eta$, we can conclude that

$$\eta \le g_0(1)^2 \wedge \frac{g_0(1)^2}{\widehat{L}(\boldsymbol{\theta}_0)} \le \frac{\|\boldsymbol{\theta}^{(1/K)} - \boldsymbol{\theta}_0\|^2}{K\widehat{L}(\boldsymbol{\theta}^{(1/K)})} \wedge \frac{\|\boldsymbol{\theta}^{(1/K)} - \boldsymbol{\theta}_0\|^2}{\widehat{L}(\boldsymbol{\theta}_0)}. \tag{83}$$

where in the second inequality we used the fact that $g_0(\cdot)$ is a non-increasing function. The desired result is obtained by Theorem 1. $\qquad\square$

## C   Generalization Analysis

Throughout this section we drop the $\sim$ in $\widetilde{\boldsymbol{\theta}}$ and $\widetilde{\Phi}(\boldsymbol{X}_i; \widetilde{\boldsymbol{\theta}})$ as everything refers to the full model. Moreover, $\widetilde{\boldsymbol{\theta}}^{(K)}$ and $\widetilde{\boldsymbol{\theta}}^{(0)}$ are denoted by $\boldsymbol{\theta}_K$ and $\boldsymbol{\theta}_0$.

For the stability analysis below, recall the definition of the leave-one-out (loo) training loss for $i \in [n]$ and note that by denoting $\ell_j(\boldsymbol{\theta}) := \ell(y_j \Phi(\boldsymbol{X}_j; \boldsymbol{\theta}))$ to be the $j$-th sample loss: $\widehat{L}^{\neg i}(\boldsymbol{\theta}) := \frac{1}{n}\sum_{j \neq i} \ell_j(\boldsymbol{\theta})$. With these, define the loo model updates of GD on the loo loss for some $\eta > 0$:

$$\boldsymbol{\theta}_{k+1}^{\neg i} := \boldsymbol{\theta}_k^{\neg i} - \eta\nabla\widehat{L}^{\neg i}(\boldsymbol{\theta}_k^{\neg i}), \; k \ge 0, \qquad \boldsymbol{\theta}_0^{\neg i} = \boldsymbol{\theta}_0.$$

**Lemma 11.** *Assume the conditions of Theorem 1 hold and*

$$\sqrt{H} \geq 256 \sqrt{T d} R^3 \left( 3 \sqrt{d} R^2 \left( 3\|\boldsymbol{\theta} - \boldsymbol{\theta}_0\| + \|\boldsymbol{\theta}\|_{2,\infty} \right) + 1 \right) \|\boldsymbol{\theta} - \boldsymbol{\theta}_0\|^2. \tag{84}$$

*Then, the on-average model stability at iteration K of GD satisfies,*

$$\frac{1}{n} \sum_{i=1}^{n} \|\boldsymbol{\theta}_K - \boldsymbol{\theta}_K^{\neg i}\| \leq \frac{2\eta}{n} \left( \sqrt{T} R \left( 2 R^2 \left( 3\|\boldsymbol{\theta} - \boldsymbol{\theta}_0\| + \|\boldsymbol{\theta}\|_{2,\infty} \right) + 1 \right) \right) \left( 2 K \widehat{L}(\boldsymbol{\theta}) + \frac{9\|\boldsymbol{\theta} - \boldsymbol{\theta}_0\|^2}{4\eta} \right).$$

*Proof.* First recall from Corollary 1 that gradient and hessian's norm satisfy

$$\|\nabla \widehat{L}(\boldsymbol{\theta})\| \leq \beta_1(\boldsymbol{\theta}) \widehat{L}(\boldsymbol{\theta}),$$

$$\|\nabla^2 \widehat{L}(\boldsymbol{\theta})\| \leq \beta_2(\boldsymbol{\theta}),$$

$$\lambda_{\min}(\nabla^2 \widehat{L}(\boldsymbol{\theta})) \geq -\frac{\beta_3(\boldsymbol{\theta})}{\sqrt{H}} \widehat{L}(\boldsymbol{\theta}).$$

Applying (Taheri & Thrampoulidis, 2023, Lemma B.1.), two arbitrary points $\boldsymbol{\theta}, \boldsymbol{\theta}'$ satisfy the following GD-expansiveness inequality:

$$\|(\boldsymbol{\theta} - \eta \nabla \widehat{L}(\boldsymbol{\theta})) - (\boldsymbol{\theta}' - \eta \nabla \widehat{L}(\boldsymbol{\theta}'))\| \leq \max_{\alpha \in [0,1]} \left\{ \left( 1 + \frac{\eta \beta_3(\boldsymbol{\theta}_\alpha)}{\sqrt{H}} \widehat{L}(\boldsymbol{\theta}_\alpha) \right) \vee \eta \beta_2(\boldsymbol{\theta}_\alpha) \right\} \|\boldsymbol{\theta} - \boldsymbol{\theta}'\|, \tag{85}$$

where $\boldsymbol{\theta}_\alpha = \alpha \boldsymbol{\theta} + (1-\alpha)\boldsymbol{\theta}'$ denotes a point parameterized by $\alpha \in [0,1]$ in the line segment between $\boldsymbol{\theta}$ and $\boldsymbol{\theta}'$. We aim to bound the on-average model stability in the r.h.s of the inequality in Lemma 3. Based on Eq. (85),

$$\|\boldsymbol{\theta}_{k+1} - \boldsymbol{\theta}_{k+1}^{\neg i}\| = \left\| \left( \boldsymbol{\theta}_k - \eta \nabla \widehat{L}^{\neg i}(\boldsymbol{\theta}_k) \right) - \left( \boldsymbol{\theta}_k^{\neg i} - \eta \nabla \widehat{L}^{\neg i}(\boldsymbol{\theta}_k^{\neg i}) \right) - \frac{\eta}{n} \nabla \ell_i(\boldsymbol{\theta}_k) \right\|$$

$$\leq \left\| \left( \boldsymbol{\theta}_k - \eta \nabla \widehat{L}^{\neg i}(\boldsymbol{\theta}_k) \right) - \left( \boldsymbol{\theta}_k^{\neg i} - \eta \nabla \widehat{L}^{\neg i}(\boldsymbol{\theta}_k^{\neg i}) \right) \right\| + \frac{\eta}{n} \|\nabla \ell_i(\boldsymbol{\theta}_k)\|$$

$$\leq \left\| \left( \boldsymbol{\theta}_k - \eta \nabla \widehat{L}^{\neg i}(\boldsymbol{\theta}_k) \right) - \left( \boldsymbol{\theta}_k^{\neg i} - \eta \nabla \widehat{L}^{\neg i}(\boldsymbol{\theta}_k^{\neg i}) \right) \right\| + \frac{\eta \beta_1(\boldsymbol{\theta}_k)}{n} \ell_i(\boldsymbol{\theta}_k)$$

$$\leq \left( \max_{\alpha \in [0,1]} \left\{ \left( 1 + \frac{\eta \beta_3(\boldsymbol{\theta}_{k_\alpha}^{\neg i})}{\sqrt{H}} \widehat{L}^{\neg i}(\boldsymbol{\theta}_{k_\alpha}^{\neg i}) \right) \vee \eta \beta_2(\boldsymbol{\theta}_{k_\alpha}^{\neg i}) \right\} \right) \|\boldsymbol{\theta}_k - \boldsymbol{\theta}_k^{\neg i}\| + \frac{\eta \beta_1(\boldsymbol{\theta}_k)}{n} \ell_i(\boldsymbol{\theta}_k). \tag{86}$$

In the above we denoted $\boldsymbol{\theta}_{k_\alpha}^{\neg i} \coloneqq \alpha \boldsymbol{\theta}_k + (1-\alpha)\boldsymbol{\theta}_k^{\neg i}$. We note that based on our guarantees for the weights' norm during training (75) it can be deduced that for all $\alpha \in [0,1]$,

$$\beta_3(\boldsymbol{\theta}_{k_\alpha}^{\neg i}) = 2 \sqrt{T d} R^3 \left( 3 \sqrt{d} R^2 \|\alpha \boldsymbol{\theta}_k + (1-\alpha)\boldsymbol{\theta}_k^{\neg i}\|_{2,\infty} + 1 \right)$$

$$= 2 \sqrt{T d} R^3 \left( 3 \sqrt{d} R^2 \left( \|\boldsymbol{\theta}_k\|_{2,\infty} \vee \|\boldsymbol{\theta}_k^{\neg i}\|_{2,\infty} \right) + 1 \right)$$

$$\leq 2 \sqrt{T d} R^3 \left( 3 \sqrt{d} R^2 \left( (\|\boldsymbol{\theta}_k - \boldsymbol{\theta}\| + \|\boldsymbol{\theta}\|_{2,\infty}) \vee (\|\boldsymbol{\theta}_k^{\neg i} - \boldsymbol{\theta}\| + \|\boldsymbol{\theta}\|_{2,\infty}) + 1 \right) \right)$$

$$\leq 2 \sqrt{T d} R^3 \left( 3 \sqrt{d} R^2 \left( 3\|\boldsymbol{\theta} - \boldsymbol{\theta}_0\| + \|\boldsymbol{\theta}\|_{2,\infty} \right) + 1 \right) =: \tilde{\beta}_3(\boldsymbol{\theta}). \tag{87}$$

Similarly, we obtain

$$\beta_2(\boldsymbol{\theta}_{k_\alpha}^{\neg i}) = \frac{2 \sqrt{T d} R^3}{\sqrt{H}} \left( 3 \sqrt{d} R^2 \|\boldsymbol{\theta}_{k_\alpha}^{\neg i}\|_{2,\infty} + 1 \right) + \frac{T R^2}{4} \left( 2 R^2 \|\boldsymbol{\theta}_{k_\alpha}^{\neg i}\|_{2,\infty} + 1 \right)^2$$

$$\leq \frac{2 \sqrt{T d} R^3}{\sqrt{H}} \left( 3 \sqrt{d} R^2 \left( 3\|\boldsymbol{\theta} - \boldsymbol{\theta}_0\| + \|\boldsymbol{\theta}\|_{2,\infty} \right) + 1 \right)$$

$$+ \frac{T R^2}{4} \left( 2 R^2 \left( 3\|\boldsymbol{\theta} - \boldsymbol{\theta}_0\| + \|\boldsymbol{\theta}\|_{2,\infty} \right) + 1 \right)^2 =: \tilde{\beta}_2(\boldsymbol{\theta}). \tag{88}$$

Hence, by the notation introduced above and noting that by our assumption on the step-size it holds $\eta \le 1/\tilde{\beta}_2(\boldsymbol{\theta})$, we can rewrite Eq. (86) as follows,

$$\left\|\boldsymbol{\theta}_{k+1} - \boldsymbol{\theta}_{k+1}^{\neg i}\right\| \le \left((1 + \frac{\eta\tilde{\beta}_3(\boldsymbol{\theta})}{\sqrt{H}}) \max_{\alpha \in [0,1]} \widehat{L}^{\neg i}(\boldsymbol{\theta}_{k_\alpha}^{\neg i})\right)\left\|\boldsymbol{\theta}_k - \boldsymbol{\theta}_k^{\neg i}\right\| + \frac{\eta\beta_1(\boldsymbol{\theta}_k)}{n}\ell_i(\boldsymbol{\theta}_k) .$$

Assume

$$\sqrt{H} \ge 128\,\tilde{\beta}_3(\boldsymbol{\theta})\left\|\boldsymbol{\theta} - \boldsymbol{\theta}_0\right\|^2 \ge 4\,\tilde{\beta}_3(\boldsymbol{\theta})\left(\left\|\boldsymbol{\theta}_k - \boldsymbol{\theta}_0\right\|^2 + \left\|\boldsymbol{\theta}_k^{\neg i} - \boldsymbol{\theta}_0\right\|^2\right) \ge 2\,\tilde{\beta}_3(\boldsymbol{\theta})\left\|\boldsymbol{\theta}_k - \boldsymbol{\theta}_k^{\neg i}\right\|^2$$
$$\ge 2\left(\beta_3(\boldsymbol{\theta}_k) \vee \beta_3(\boldsymbol{\theta}_k^{\neg i})\right)\left\|\boldsymbol{\theta}_k - \boldsymbol{\theta}_k^{\neg i}\right\|^2,$$

where we used Theorem 1 in the first inequality. We also have

$$\min_{\alpha \in [0,1]} \lambda_{\min}(\nabla^2 \widehat{L}^{\neg i}(\boldsymbol{\theta}_{k_\alpha}^{\neg i})) \ge -\frac{\beta_3(\boldsymbol{\theta}_k) \vee \beta_3(\boldsymbol{\theta}_k^{\neg i})}{\sqrt{H}}\widehat{L}^{\neg i}(\boldsymbol{\theta}_{k_\alpha}^{\neg i}).$$

Thus, by applying Proposition 4 on the leave-one-out loss, it holds that for all $\alpha \in [0,1]$,

$$\max_{\alpha \in [0,1]} \widehat{L}^{\neg i}(\boldsymbol{\theta}_{k_\alpha}^{\neg i}) \le \frac{4}{3}\left(\widehat{L}^{\neg i}(\boldsymbol{\theta}_k) + \widehat{L}^{\neg i}(\boldsymbol{\theta}_k^{\neg i})\right) .$$

Thus,

$$\left\|\boldsymbol{\theta}_{k+1} - \boldsymbol{\theta}_{k+1}^{\neg i}\right\| \le \left(\left(1 + \frac{4\eta\tilde{\beta}_3(\boldsymbol{\theta})}{3\sqrt{H}}\right) \cdot \left(\widehat{L}^{\neg i}(\boldsymbol{\theta}_k) + \widehat{L}^{\neg i}(\boldsymbol{\theta}_k^{\neg i})\right)\right)\left\|\boldsymbol{\theta}_k - \boldsymbol{\theta}_k^{\neg i}\right\| + \frac{\eta\beta_1(\boldsymbol{\theta}_k)}{n}\ell_i(\boldsymbol{\theta}_k) . \tag{89}$$

In order to remove the dependence on $k$, note that

$$\beta_1(\boldsymbol{\theta}_k) \le \sqrt{T}\,R\left(2\,R^2\left\|\boldsymbol{\theta}_k\right\|_{2,\infty} + 1\right)$$
$$\le \sqrt{T}\,R\left(2\,R^2\left(\left\|\boldsymbol{\theta} - \boldsymbol{\theta}_k\right\|_{2,\infty} + \left\|\boldsymbol{\theta}\right\|_{2,\infty}\right) + 1\right)$$
$$\le \sqrt{T}\,R\left(2\,R^2\left(\left\|\boldsymbol{\theta} - \boldsymbol{\theta}_k\right\| + \left\|\boldsymbol{\theta}\right\|_{2,\infty}\right) + 1\right)$$
$$\le \sqrt{T}\,R\left(2\,R^2\left(3\|\boldsymbol{\theta} - \boldsymbol{\theta}_0\| + \left\|\boldsymbol{\theta}\right\|_{2,\infty}\right) + 1\right) =: \tilde{\beta}_1(\boldsymbol{\theta}) . \tag{90}$$

For simplicity of exposition, denote $\alpha_{k,i} := \frac{4\eta\tilde{\beta}_3(\boldsymbol{\theta})}{3\sqrt{H}}\left(\widehat{L}^{\neg i}(\boldsymbol{\theta}_k) + \widehat{L}^{\neg i}(\boldsymbol{\theta}_k^{\neg i})\right)$. Then by unrolling the iterates we have,

$$\left\|\boldsymbol{\theta}_{k+1} - \boldsymbol{\theta}_{k+1}^{\neg i}\right\| \le (1 + \alpha_{k,i})\left\|\boldsymbol{\theta}_k - \boldsymbol{\theta}_k^{\neg i}\right\| + \frac{\eta\tilde{\beta}_1(\boldsymbol{\theta})}{n}\ell_i(\boldsymbol{\theta}_k)$$
$$\le (1 + \alpha_{k,i})(1 + \alpha_{k-1,i})\left\|\boldsymbol{\theta}_{k-1} - \boldsymbol{\theta}_{k-1}^{\neg i}\right\| + \frac{(1 + \alpha_{k,i})\eta\tilde{\beta}_1(\boldsymbol{\theta})}{n}\ell_i(\boldsymbol{\theta}_{k-1}) + \frac{\eta\tilde{\beta}_1(\boldsymbol{\theta})}{n}\ell_i(\boldsymbol{\theta}_k)$$
$$\le \sum_{j=1}^{k} \prod_{l=j}^{k}(1 + \alpha_{l,i})\eta\tilde{\beta}_1(\boldsymbol{\theta})\frac{\ell_i(\boldsymbol{\theta}_{j-1})}{n} + \eta\tilde{\beta}_1(\boldsymbol{\theta})\frac{\ell_i(\boldsymbol{\theta}_k)}{n}$$
$$\le \sum_{j=1}^{k} \exp(\sum_{l=j}^{k}\alpha_{l,i})\,\eta\tilde{\beta}_1(\boldsymbol{\theta})\frac{\ell_i(\boldsymbol{\theta}_{j-1})}{n} + \eta\tilde{\beta}_1(\boldsymbol{\theta})\frac{\ell_i(\boldsymbol{\theta}_k)}{n}$$
$$\le \sum_{j=1}^{k} \exp(\sum_{l=1}^{k}\alpha_{l,i})\,\eta\tilde{\beta}_1(\boldsymbol{\theta})\frac{\ell_i(\boldsymbol{\theta}_{j-1})}{n} + \eta\tilde{\beta}_1(\boldsymbol{\theta})\frac{\ell_i(\boldsymbol{\theta}_k)}{n} , \tag{91}$$

where in the above we used that $\boldsymbol{\theta}_0 = \boldsymbol{\theta}_0^{\neg i}$ in unrolling the iterates as well as the fact that for $x \geq 0 : 1 + x \leq e^x$. Note that by definition $\widehat{L}^{\neg i}(\boldsymbol{\theta}_k) \leq \widehat{L}(\boldsymbol{\theta}_k)$. By training loss guarantees from Eq. (79), we have

$$
\begin{aligned}
\sum_{l=1}^{k} \alpha_{l,i} &\leq \frac{4\eta\tilde{\beta}_3(\boldsymbol{\theta})}{3\sqrt{H}} \sum_{l=1}^{k} \left( \widehat{L}^{\neg i}(\boldsymbol{\theta}_l) + \widehat{L}^{\neg i}(\boldsymbol{\theta}_l^{\neg i}) \right) \\
&\leq \frac{4\eta\tilde{\beta}_3(\boldsymbol{\theta})}{3\sqrt{H}} \sum_{l=1}^{k} \left( \widehat{L}(\boldsymbol{\theta}_l) + \widehat{L}(\boldsymbol{\theta}_l^{\neg i}) \right) \\
&\leq \frac{4\eta\tilde{\beta}_3(\boldsymbol{\theta})}{3\sqrt{H}} \left( 5k\widehat{L}(\boldsymbol{\theta}) + \frac{5\|\boldsymbol{\theta} - \boldsymbol{\theta}_0\|^2}{2\eta} \right) \\
&\leq \frac{10\tilde{\beta}_3(\boldsymbol{\theta})\|\boldsymbol{\theta} - \boldsymbol{\theta}_0\|^2}{\sqrt{H}} \\
&\leq \frac{1}{10},
\end{aligned}
$$

where the last step stems from the condition on $\sqrt{H}$. Proceeding from Eq. (91), we find that for the last iterate

$$
\begin{aligned}
\left\| \boldsymbol{\theta}_K - \boldsymbol{\theta}_K^{\neg i} \right\| &\leq 2\eta\tilde{\beta}_1(\boldsymbol{\theta}) \sum_{k=1}^{K-1} \frac{\ell_i(\boldsymbol{\theta}_{k-1})}{n} + \eta\tilde{\beta}_1(\boldsymbol{\theta}) \frac{\ell_i(\boldsymbol{\theta}_{K-1})}{n} \\
&\leq 2\eta\tilde{\beta}_1(\boldsymbol{\theta}) \sum_{k=0}^{K-1} \frac{\ell_i(\boldsymbol{\theta}_k)}{n} \,.
\end{aligned}
\tag{92}
$$

It follows that the on-average model stability satisfies,

$$
\begin{aligned}
\frac{1}{n} \sum_{i=1}^{n} \left\| \boldsymbol{\theta}_K - \boldsymbol{\theta}_K^{\neg i} \right\| &\leq \frac{2\eta\tilde{\beta}_1(\boldsymbol{\theta})}{n^2} \sum_{i=1}^{n} \sum_{k=0}^{K-1} \ell_i(\boldsymbol{\theta}_k) \\
&= \frac{2\eta\tilde{\beta}_1(\boldsymbol{\theta})}{n} \sum_{k=0}^{K-1} \widehat{L}(\boldsymbol{\theta}_k) \,.
\end{aligned}
$$

Applying our training loss guarantees from Eq. (80) to the r.h.s. above yields,

$$
\frac{1}{n} \sum_{i=1}^{n} \left\| \boldsymbol{\theta}_K - \boldsymbol{\theta}_K^{\neg i} \right\| \leq \frac{2\eta\tilde{\beta}_1(\boldsymbol{\theta})}{n} \left( 2K\widehat{L}(\boldsymbol{\theta}) + \frac{9\|\boldsymbol{\theta} - \boldsymbol{\theta}_0\|^2}{4\eta} \right),
$$

where here we used the assumption that, $\widehat{L}(\boldsymbol{\theta}_0) \leq \|\boldsymbol{\theta} - \boldsymbol{\theta}_0\|^2/\eta$ to simplify the final result. This completes the proof. $\qquad\square$

## C.1 Proof of Theorem 2

We restate the theorem here for convenience, this time also including exact constants.

**Theorem 4** (Restatement of Thm. 2). *Fix any $\boldsymbol{\theta}$ and $H$ satisfying*

$$
\sqrt{H} \geq 256 \sqrt{T d}\, R^3 \left( 3 \sqrt{d}\, R^2 \left( 3\|\boldsymbol{\theta} - \boldsymbol{\theta}_0\| + \|\boldsymbol{\theta}\|_{2,\infty} \right) + 1 \right) \|\boldsymbol{\theta} - \boldsymbol{\theta}_0\|^2.
$$

*Further, denote for convenience*

$$
\alpha(\boldsymbol{\theta}) \coloneqq 3 \sqrt{d}\, R^2 \left[ 3 \sqrt{T}\, R^3 \left( 3 \|\boldsymbol{\theta} - \boldsymbol{\theta}_0\| + \|\boldsymbol{\theta}\|_{2,\infty} \right) + 2 \sqrt{T}\, R \right]
$$

*and $\rho(\boldsymbol{\theta}) = \left( \frac{2\sqrt{T d}\, R^3}{\sqrt{H}} + \frac{T R^2}{4} \right) \alpha(\boldsymbol{\theta})^2$. Then, for any step-size $\eta \leq 1 \wedge 1/\rho(\boldsymbol{\theta}) \wedge \frac{\|\boldsymbol{\theta} - \boldsymbol{\theta}_0\|^2}{K\widehat{L}(\boldsymbol{\theta})} \wedge \frac{\|\boldsymbol{\theta} - \boldsymbol{\theta}_0\|^2}{\widehat{L}(\boldsymbol{\theta}_0)}$, the expected generalization gap at iteration $K$ satisfies,*

$$
\mathbb{E}\left[ L(\widetilde{\boldsymbol{\theta}}^{(K)}) - \widehat{L}(\widetilde{\boldsymbol{\theta}}^{(K)}) \right] \leq \frac{4}{n} \mathbb{E}\left[ 2 K \widehat{L}(\widetilde{\boldsymbol{\theta}}) + \frac{9\|\widetilde{\boldsymbol{\theta}} - \widetilde{\boldsymbol{\theta}}^{(0)}\|^2}{4\eta} \right].
\tag{93}
$$

*Proof.* Note that the assumptions of Lemma 11 are satisfied. Moreover, as per Corollary 1 the objective is Lipschitz at all iterates with parameter $\tilde{\beta}_1(\boldsymbol{\theta})$ since $\forall k \in [K] : \|\boldsymbol{\theta}_k\|_{2,\infty} \le 3\|\boldsymbol{\theta} - \boldsymbol{\theta}_0\| + \|\boldsymbol{\theta}\|_{2,\infty}$. Thus, by Lemma 3 and Lemma 11 we have,

$$\mathbb{E}\left[L(\boldsymbol{\theta}_K) - \widehat{L}(\boldsymbol{\theta}_K)\right] \le \frac{4}{n} \mathbb{E}\left[\eta \left(\tilde{\beta}_1(\boldsymbol{\theta})\right)^2 \left(2K\widehat{L}(\boldsymbol{\theta}) + \frac{9\|\boldsymbol{\theta} - \boldsymbol{\theta}_0\|^2}{4\eta}\right)\right]. \tag{94}$$

Recalling the condition on step-size $\eta \le \frac{1}{\rho(\boldsymbol{\theta})} \le \frac{1}{(\tilde{\beta}_1(\boldsymbol{\theta}))^2}$ concludes the proof. $\qquad\square$

## C.2 Corollary 4

**Corollary 4** (Generalization loss under realizability)**.** *Let boundedness Assumption 1 hold. Also let realizability assumption 2 holds almost surely over the data distribution. Fix $K \ge 1$. Assume any $H$ such that*

$$\sqrt{H} \ge 256\sqrt{T d}\, R^3 \left(3\sqrt{d}\, R^2 \left(3g_0(\frac{1}{K}) + g(\frac{1}{K})\right) + 1\right) g_0(\frac{1}{K})^2. \tag{95}$$

*Let the step-size satisfy $\eta \le 1 \wedge 1/\rho(K) \wedge g_0(1)^2 \wedge \frac{g_0(1)^2}{\widehat{L}(\boldsymbol{\theta}_0)}$ where $\rho(K)$ is as defined in Corollary 3. Then the expected generalization gap at iteration $K$ of GD satisfies,*

$$\mathbb{E}\left[L(\boldsymbol{\theta}_K) - \widehat{L}(\widetilde{\boldsymbol{\theta}}_K)\right] \le \frac{17\, g_0(\frac{1}{K})^2}{\eta\, n}. \tag{96}$$

*Proof.* According to Assumption 2, for any sufficiently small $\varepsilon > 0$, there exists a $\boldsymbol{\theta}^{(\varepsilon)}$ such that $\widehat{L}(\boldsymbol{\theta}^{(\varepsilon)}) \le \varepsilon$ and $\|\boldsymbol{\theta}^{(\varepsilon)} - \widetilde{\boldsymbol{\theta}}_0\| = g_0(\varepsilon)$. Pick $\varepsilon = \frac{1}{K}$. Let the step-size, $\eta > 0$ satisfies the assumption of Descent Lemma 7. Since $\widehat{L}(\boldsymbol{\theta}^{(1/K)}) \le \frac{1}{K}$, we have:

$$\frac{\|\boldsymbol{\theta}^{(1/K)} - \boldsymbol{\theta}_0\|^2}{K\widehat{L}(\boldsymbol{\theta}^{(1/K)})} \ge \|\boldsymbol{\theta}^{(1/K)} - \boldsymbol{\theta}_0\|^2 = g_0(\frac{1}{K})^2 \ge g_0(1)^2$$

and

$$\frac{\|\boldsymbol{\theta}^{(1/K)} - \boldsymbol{\theta}_0\|^2}{\widehat{L}(\boldsymbol{\theta}_0)} = \frac{g_0(\frac{1}{K})^2}{\widehat{L}(\boldsymbol{\theta}_0)} \ge \frac{g_0(1)^2}{\widehat{L}(\boldsymbol{\theta}_0)}.$$

Therefore, we can conclude that

$$\eta \le g_0(1)^2 \wedge \frac{g_0(1)^2}{\widehat{L}(\boldsymbol{\theta}_0)} \le \frac{\|\boldsymbol{\theta}^{(1/K)} - \boldsymbol{\theta}_0\|^2}{K\widehat{L}(\boldsymbol{\theta}^{(1/K)})} \wedge \frac{\|\boldsymbol{\theta}^{(1/K)} - \boldsymbol{\theta}_0\|^2}{\widehat{L}(\boldsymbol{\theta}_0)}. \tag{97}$$

where in the second inequality we used the fact that $g_0(\cdot)$ is a non-increasing function. The desired result is obtained by Theorem 2 and the fact that

$$K\widehat{L}(\boldsymbol{\theta}^{(1/K)}) \le \frac{\|\boldsymbol{\theta}^{(1/K)} - \boldsymbol{\theta}_0\|^2}{\eta} = \frac{g_0(\frac{1}{K})^2}{\eta}.$$

$\qquad\square$

## C.3 From good initialization to realizability

The proposition below shows that starting from a `good` initialization we can always find $\boldsymbol{\theta}^{(\epsilon)}$ satisfying the realizability Assumption 2 provided the number of heads is large enough.

**Proposition 7** (From `good` initialization to realizability)**.** *Suppose `good` initialization $\boldsymbol{\theta}_0$ as per Definition 1. Fix any $1 \ge \varepsilon > 0$ and let*

$$\sqrt{H} \ge \frac{5\sqrt{T d}\, R^3 B_2}{B_\Phi} \cdot \left(3\sqrt{d}\, R^2 + 1\right) \cdot \left(\frac{2B_\Phi + \log(1/\varepsilon)}{\gamma}\right)^2 \cdot \left(1 \vee \frac{2B_\Phi + \log(1/\varepsilon)}{\gamma}\right). \tag{98}$$

*Then, the realizability Assumption 2 holds with $g_0(\varepsilon) = \frac{1}{\gamma}\left(2B_\Phi + \log(1/\varepsilon)\right)$ and $g(\epsilon) = B_2 + g_0(\epsilon)$.*

*Proof.* By Taylor expansion there exists $\boldsymbol{\theta}' \in [\boldsymbol{\theta}, \boldsymbol{\theta}_0]$ such that,

$$y_i \Phi(\boldsymbol{X}_i; \boldsymbol{\theta}) = y_i \Phi(\boldsymbol{X}_i; \boldsymbol{\theta}_0) + y_i \langle \nabla \Phi(\boldsymbol{X}_i; \boldsymbol{\theta}_0), \boldsymbol{\theta} - \boldsymbol{\theta}_0 \rangle + \frac{1}{2} y_i \langle \boldsymbol{\theta} - \boldsymbol{\theta}_0, \nabla^2 \Phi(\boldsymbol{X}_i; \boldsymbol{\theta}')(\boldsymbol{\theta} - \boldsymbol{\theta}_0) \rangle. \tag{99}$$

Pick

$$\boldsymbol{\theta} := \boldsymbol{\theta}^{(\varepsilon)} = \boldsymbol{\theta}_0 + \frac{2B_\Phi + \log(1/\varepsilon)}{\gamma} \boldsymbol{\theta}_\star.$$

Substituting this in (99) and using that $\boldsymbol{\theta}_0$ is a `good` initialization, we obtain for all $i \in [n]$:

$$y_i \Phi(\boldsymbol{X}_i; \boldsymbol{\theta}) \geq -|y_i \Phi(\boldsymbol{X}_i; \boldsymbol{\theta}_0)| + (2B_\Phi + \log(1/\varepsilon)) - \frac{1}{2} \|\nabla^2 \Phi(\boldsymbol{X}_i; \boldsymbol{\theta}')\|_2 \|\boldsymbol{\theta} - \boldsymbol{\theta}_0\|^2$$

$$\geq -B_\Phi + (2B_\Phi + \log(1/\varepsilon)) - \frac{1}{2} \|\nabla^2 \Phi(\boldsymbol{X}_i; \boldsymbol{\theta}')\|_2 \left( \frac{2B_\Phi + \log(1/\varepsilon)}{\gamma} \right)^2. \tag{100}$$

To continue, we show in Lemma 5 in the appendix that $\|\nabla^2 \Phi(\boldsymbol{X}_i; \boldsymbol{\theta}')\|_2 \leq \beta_3(\boldsymbol{\theta}')/\sqrt{H}$ where recall $\beta_3(\boldsymbol{\theta}) := 2\sqrt{T d}\, R^3 \left( 3\sqrt{d}\, R^2 \|\boldsymbol{\theta}\|_{2,\infty} + 1 \right)$ defined in Corollary 1. Now, note that

$$\beta_3(\boldsymbol{\theta}') \leq \beta_3(\boldsymbol{\theta}^{(\varepsilon)}) + \beta_3(\boldsymbol{\theta}_0) \leq \frac{2B_\Phi + \log(1/\varepsilon)}{\gamma} \cdot \beta_3(\boldsymbol{\theta}_\star) + 2\beta_3(\boldsymbol{\theta}_0)$$

$$\leq 2\sqrt{2}\, \sqrt{T d}\, R^3 B_2 \left( 3\sqrt{d}\, R^2 + 1 \right) \cdot \left( 2 + \frac{2B_\Phi + \log(1/\varepsilon)}{\gamma} \right)$$

$$\leq 10\, \sqrt{T d}\, R^3 B_2 \left( 3\sqrt{d}\, R^2 + 1 \right) \cdot \left( 1 \vee \frac{2B_\Phi + \log(1/\varepsilon)}{\gamma} \right),$$

where the first two inequalities follow by triangle inequality and the inequality after those follows because $\|\boldsymbol{\theta}_\star\|_{2,\infty} \leq \|\boldsymbol{\theta}_\star\|_2 \leq \sqrt{2}$, $1 \leq B_2$ and also $\|\boldsymbol{\theta}_0\|_{2,\infty} \leq B_2$ by `good` initialization assumption.

Plugging in this bound in (100) and using the assumption on $H$ yields that $y_i \Phi(\boldsymbol{X}_i; \boldsymbol{\theta}) \geq \log(1/\epsilon)$ for all $i \in [n]$. This in turn implies that $\widehat{L}_i(\boldsymbol{\theta}) := \ell(y_i \Phi(\boldsymbol{X}_i; \boldsymbol{\theta})) \leq \log(1 + \varepsilon) \leq \varepsilon$, and thus $\widehat{L}(\boldsymbol{\theta}) \leq \varepsilon$ as desired. Furthermore, note by definition of $\boldsymbol{\theta}^{(\epsilon)}$ that $g_0(\epsilon) = \frac{2B_\Phi + \log(1/\varepsilon)}{\gamma}$ and $g(\epsilon) = B_2 + g_0(\epsilon)$. For the latter, we used triangle inequality and the rough bound $\|\boldsymbol{\theta}_\star\|_{2,\infty} \leq \|\boldsymbol{\theta}_\star\|_2 \leq \sqrt{2}$. This completes the proof. $\qquad \square$

## C.4 Proof of Corollary 2

We restate the corollary here for convenience, this time also including exact constants.

**Corollary 5** (Restatement of Cor. 2). *Suppose* `good` *initialization* $\boldsymbol{\theta}_0$ *and let*

$$\sqrt{H} \geq 256 \sqrt{T d}\, R^3 B_2 \left( 3\sqrt{d}\, R^2 \left( 4 g_0(\frac{1}{K}) + B_2 \right) + 1 \right) g_0(\frac{1}{K})^2,$$

*where* $g_0(\frac{1}{K}) = \frac{2B_\Phi + \log(K)}{\gamma}$. *Further, denote for convenience*

$$\alpha(K) := 3\sqrt{d}\, R^2 \left[ 3\sqrt{T}\, R^3 \left( 4g_0(\frac{1}{K}) + B_2 \right) + 2\sqrt{T}\, R \right]$$

*and* $\rho(K) = \left( \frac{2\sqrt{T d}\, R^3}{\sqrt{H}} + \frac{T R^2}{4} \right) \alpha(K)^2$. *Then, for any fixed step-size*

$$\eta \leq 1 \wedge 1/\rho(K) \wedge \frac{4B_\Phi^2}{\gamma^2} \cdot \frac{1}{\log(1 + e^{B_\Phi})},$$

*the following bounds hold:*

$$\widehat{L}(\boldsymbol{\theta}_K) \leq \frac{2}{K} + \frac{5\, (2B_\Phi + \log(K))^2}{4\gamma^2 \eta K} \quad and \quad \mathbb{E}\big[ L(\boldsymbol{\theta}_K) - \widehat{L}(\boldsymbol{\theta}_K) \big] \leq \frac{17\, (2B_\Phi + \log(K))^2}{\gamma^2 \eta n}. \tag{101}$$

*Proof.* First, we prove that the given assumption on $H$ satisfies the conditions of Proposition 7 for $\varepsilon = \frac{1}{K}$.

$$\sqrt{H} \geq 256\sqrt{Td}\,R^3\,B_2\left(3\sqrt{d}\,R^2\left(4\,g_0(\frac{1}{K}) + B_2\right) + 1\right)g_0(\frac{1}{K})^2$$

$$= 256\sqrt{Td}\,R^3\,B_2\left(3\sqrt{d}\,R^2\left(3\,g_0(\frac{1}{K}) + g(\frac{1}{K})\right) + 1\right)g_0(\frac{1}{K})^2\,.$$

If $1 > g_0(\frac{1}{K})$,

$$\sqrt{H} \geq 256\sqrt{Td}\,R^3\,B_2\left(3\sqrt{d}\,R^2\left(4\,g_0(\frac{1}{K}) + B_2\right) + 1\right)g_0(\frac{1}{K})^2$$

$$\geq 5\sqrt{Td}\,R^3\,B_2\left(3\sqrt{d}\,R^2 + 1\right)g_0(\frac{1}{K})^2\,.$$

It means that the condition of Proposition 7 on $\sqrt{H}$ is satisfied. Moreover, if $1 \leq g_0(\frac{1}{K})$,

$$\sqrt{H} \geq 256\sqrt{Td}\,R^3\,B_2\left(3\sqrt{d}\,R^2\left(4\,g_0(\frac{1}{K}) + B_2\right) + 1\right)g_0(\frac{1}{K})^2$$

$$\geq 256\sqrt{Td}\,R^3\,B_2\left(12\sqrt{d}\,R^2\,g_0(\frac{1}{K})\right)g_0(\frac{1}{K})^2$$

$$\geq 5\sqrt{Td}\,R^3\,B_2\left(3\sqrt{d}\,R^2 + 1\right)g_0(\frac{1}{K})^3\,,$$

and again the condition of Proposition 7 on $\sqrt{H}$ is satisfied. Then, we can apply the results of Corollaries 3 and 4 for a fixed $K$ which satisfies $K \geq 1$. Note that

$$g_0(1) = \frac{2B_\Phi}{\gamma}\,, \quad \text{and} \quad \widehat{L}(\boldsymbol{\theta}_0) \leq \log(1 + e^{B_\Phi})\,.$$

Thus, the condition on step-size simplifies to $\eta \leq 1 \wedge 1/\rho(K) \wedge \frac{4B_\Phi^2}{\gamma^2} \cdot \frac{1}{\log(1+e^{B_\Phi})}$. This completes the proof. □

# D   Proofs for Section 5

## D.1   Useful facts

**Fact 1.** *Let $\boldsymbol{x} \in \mathbb{R}^d$ be subgaussian vector with $\|\boldsymbol{x}\|_{\psi_2} \leq K$. Then, for any $\delta \in (0,1)$ and absolute constant $C > 0$ it holds with probability at least $1 - \delta$ that $\|\boldsymbol{x}\|_2 \leq CK\left(\sqrt{d} + \sqrt{\log(1/\delta)}\right)$.*

**Fact 2.** *Suppose $X_h, h \in [H]$ are IID realizations of random variable $X$ for which $E[X] = \mu$ and $|X| \leq B$ almost surely. Then, for any $\delta \in (0,1)$, with probability at least $1 - \delta$ it holds that*

$$\left|\frac{1}{H}\sum_{h\in[H]}X_h - \mu\right| \leq 2B\sqrt{\frac{\log(1/\delta)}{2H}}\,.$$

## D.2   Proof of Proposition 2

Recall

$$\widetilde{\Phi}(\boldsymbol{X};\widetilde{\boldsymbol{\theta}}) = \frac{1}{\sqrt{H}}\sum_{h\in[H]}\Phi_h(\boldsymbol{X};\boldsymbol{W}_h,\boldsymbol{U}_h) = \frac{1}{\sqrt{H}}\sum_{h\in[H]}\langle\boldsymbol{U}_h,\boldsymbol{\varphi}(\boldsymbol{X}\boldsymbol{W}_h\boldsymbol{X}^\top)\boldsymbol{X}\rangle\,,$$

where $\widetilde{\boldsymbol{\theta}} = \text{concat}(\boldsymbol{\theta}_1,\boldsymbol{\theta}_2,...,\boldsymbol{\theta}_H)$ denotes the trainable parameters. After completing the first phase of training, the initialization for the second phase is as follows for all $h \in [H]$:

$$\boldsymbol{\theta}_h^{(1)} = \text{concat}(\boldsymbol{U}_h^{(1)},\boldsymbol{W}_h^{(1)}) \quad : \quad \boldsymbol{W}_h^{(1)} = \boldsymbol{0}, \quad \boldsymbol{U}_h^{(1)} = \alpha_h\left(\frac{\zeta}{2}\mathbb{1}_T\boldsymbol{u}_\star^\top + \mathbb{1}_T\boldsymbol{p}^\top\right), \tag{102}$$

where recall that $\alpha_h \sim \text{Unif}(\pm 1)$ and from Lemma 2 it holds with probability $1 - \delta$ that $\|\boldsymbol{p}\| \leq P$ where the parameter $P$ is defined in (6). Onwards, we condition on this good event.

We prove the three properties **P1**, **P2**, and **P3** in the order stated below.

### D.2.1 Proof of P1: Bounded norm per head

This is straightforward by noting that for all $h \in [H]$:

$$\|\boldsymbol{\theta}_h^{(1)}\|_2 = \|\boldsymbol{U}_h^{(1)}\|_F \leq \frac{\zeta}{2}\sqrt{T}\|\boldsymbol{u}_\star\|_2 + \sqrt{T}\|\boldsymbol{p}\|_2 \leq \frac{\zeta}{2}\sqrt{T}\sqrt{2}S + \sqrt{T}P.$$

### D.2.2 Proof of P2: Bounded initialization

**Lemma 12** (Initialization bound). *Let any $\boldsymbol{X}$ sampled from DM1 and satisfying Assumption 3. Given the initialization in (102), for any $\delta \in (0,1)$ it holds with probability at least $1 - \delta$ that*

$$|\widetilde{\Phi}(\boldsymbol{X}; \widetilde{\boldsymbol{\theta}}^{(1)})| \leq T R (S + P)\sqrt{2\log(1/\delta)}.$$

*Proof.*

$$\widetilde{\Phi}(\boldsymbol{X}; \widetilde{\boldsymbol{\theta}}^{(1)}) = \frac{1}{\sqrt{H}}\sum_{h\in[H]}\Phi_h(\boldsymbol{X}; \boldsymbol{\theta}_h^{(1)}) = \frac{1}{\sqrt{H}}\sum_{h\in[H]}\langle \boldsymbol{U}_h^{(1)}, \boldsymbol{\varphi}(\boldsymbol{X}\boldsymbol{W}_h^{(1)}\boldsymbol{X}^\top)\boldsymbol{X}\rangle.$$

Using the initialization in (102) and recalling $\boldsymbol{\varphi}(\boldsymbol{0}) = \mathbb{1}_T\mathbb{1}_T^\top/T$, we have

$$\frac{1}{\sqrt{H}}\sum_{h\in[H]}\langle \boldsymbol{U}_h^{(1)}, \boldsymbol{\varphi}(\boldsymbol{X}\boldsymbol{W}_h^{(1)}\boldsymbol{X}^\top)\boldsymbol{X}\rangle = \frac{1}{\sqrt{H}}\sum_{h\in[H]}\frac{\alpha_h}{T}\left\langle \boldsymbol{U}_h^{(1)}, \mathbb{1}_T\mathbb{1}_T^\top\boldsymbol{X}\right\rangle =: \frac{1}{\sqrt{H}}\sum_{h\in[H]}X_h. \qquad (103)$$

Note for each $h \in [H]$ that $X_h$ in (103) depends only on the random variable $\alpha_h$. Recall that $\alpha_h, h \in [H]$ are IID $\text{Unif}(\pm 1)$. Thus, $\{X_h\}_{h\in[H]}$ are IID with 0 mean as $\mathbb{E}\alpha = 0$. Further, note that

$$\begin{aligned}
|X_h| &= \left|\frac{\alpha}{T}\left\langle \boldsymbol{U}^{(1)}, \mathbb{1}_T\mathbb{1}_T^\top\boldsymbol{X}\right\rangle\right| = \left|\frac{\alpha}{T}\left\langle \frac{\zeta}{2}\mathbb{1}_T\boldsymbol{u}_\star^\top + \mathbb{1}_T\boldsymbol{p}^\top, \mathbb{1}_T\mathbb{1}_T^\top\boldsymbol{X}\right\rangle\right| \\
&\leq \frac{1}{T}\left(\frac{\zeta}{2}\sqrt{T}\|\boldsymbol{u}_\star\| + \sqrt{T}\|\boldsymbol{p}\|\right)\sqrt{T}\|\sum_{t\in[T]}\boldsymbol{x}_t\| \leq \frac{1}{T}\left(\frac{\zeta}{2}\sqrt{T}\|\boldsymbol{u}_\star\| + \sqrt{T}\|\boldsymbol{p}\|\right)\sqrt{T}RT \leq TR(S+P).
\end{aligned}$$

Thus, using Hoeffding's inequality (see Fact 2) we have for some absolute constant $c > 0$, with probability at least $1 - \delta$:

$$|\widetilde{\Phi}(\boldsymbol{X}; \widetilde{\boldsymbol{\theta}}^{(1)})| \leq TR(S+P)\sqrt{2\log(1/\delta)}.$$

$\qquad\qquad\square$

### D.2.3 Proof of P3: NTK separability

We prove property **P3** in two steps each stated in a separate lemma below

**Lemma 13.** *Let*

$$\boldsymbol{W}_\star = \boldsymbol{\mu}_+\boldsymbol{\mu}_+^\top + \boldsymbol{\mu}_-\boldsymbol{\mu}_-^\top + \sum_{\ell\in[M]}\boldsymbol{\nu}_\ell(\boldsymbol{\mu}_+ + \boldsymbol{\mu}_-)^\top, \qquad (104)$$

$$\boldsymbol{U}_\star = \mathbb{1}_T\boldsymbol{u}_\star^\top. \qquad (105)$$

*Given the initialization $\boldsymbol{\theta}^{(1)}$ in (102) and $\boldsymbol{\theta}_\star := (\overline{\boldsymbol{U}}_\star, \boldsymbol{sign}(\alpha)\overline{\boldsymbol{W}}_\star)$ we have for any $(\boldsymbol{X}, y)$ sampled from DM1 and satisfying Assumption 3:*

$$\mathbb{E}_{\boldsymbol{\theta}^{(1)}}\, y\langle \nabla_{\boldsymbol{\theta}}\Phi(\boldsymbol{X}; \boldsymbol{\theta}^{(1)}), \boldsymbol{\theta}_\star\rangle \geq \gamma_\star,$$

*where the expectation is taken over $\alpha \sim \mathrm{Unif}(\pm 1)$ and*

$$\gamma_\star := \frac{T(1-\zeta)\zeta}{4\sqrt{2(M+1)}}\left(\zeta S^4 - 7\bar{Z}S^2 - 12\bar{Z}^2 - 16\frac{\bar{Z}^3}{S^2}\right) - PT^{5/2}(S+Z)^3 + \frac{S\sqrt{T}}{\sqrt{2}}\left(\zeta - 2(1-\zeta)\frac{Z_\mu}{S^2}\right),$$

*where $\bar{Z} = Z_\mu \vee Z_\nu$, and $\overline{U}_\star, \overline{W}_\star$ denote normalized $U_\star, W_\star$, respectively.*

*Proof.*

$$\langle \nabla_{\boldsymbol{\theta}}\Phi(\boldsymbol{X};\boldsymbol{\theta}^{(1)}), \boldsymbol{\theta}_\star\rangle = \mathtt{sign}(\alpha)\langle \nabla_{\boldsymbol{W}}\Phi(\boldsymbol{X};\boldsymbol{\theta}^{(1)}), \overline{\boldsymbol{W}}_\star\rangle + \langle \nabla_{\boldsymbol{U}}\Phi(\boldsymbol{X};\boldsymbol{\theta}^{(1)}), \overline{\boldsymbol{U}}_\star\rangle.$$

Using $\boldsymbol{\theta}^{(1)} = \mathrm{concat}(\boldsymbol{U}^{(1)}, 0)$ and $\overline{\boldsymbol{U}}_\star = \frac{\boldsymbol{U}_\star}{\|\boldsymbol{U}_\star\|_F} = \frac{1}{S\sqrt{2T}}\mathbb{1}_T \boldsymbol{u}_\star^\top$, we have:

$$y\langle \nabla_{\boldsymbol{U}}\Phi(\boldsymbol{X};\boldsymbol{\theta}^{(1)}), \overline{\boldsymbol{U}}_\star\rangle = y\langle \boldsymbol{\varphi}(\boldsymbol{X}\boldsymbol{W}^{(1)}\boldsymbol{X}^\top)\boldsymbol{X}, \overline{\boldsymbol{U}}_\star\rangle = \frac{y}{TS\sqrt{2T}}\langle \mathbb{1}_T\mathbb{1}_T^\top\boldsymbol{X}, \mathbb{1}_T\boldsymbol{u}_\star^\top\rangle$$

$$= \frac{y}{S\sqrt{2T}}\boldsymbol{u}_\star^\top\left(\sum_t \boldsymbol{x}_t\right) \geq \frac{S\sqrt{T}}{\sqrt{2}}\left(\zeta - 2(1-\zeta)\frac{Z_\mu}{S^2}\right). \tag{106}$$

The gradient with respect to $\boldsymbol{W}$ evaluated at $\boldsymbol{\theta}^{(1)} = \mathrm{concat}(\boldsymbol{U}^{(1)}, 0)$ is

$$\nabla_{\boldsymbol{W}}\Phi(\boldsymbol{X};\boldsymbol{\theta}^{(1)}) = \frac{\alpha\zeta}{2}\sum_{t\in[T]}\boldsymbol{x}_t\boldsymbol{u}_\star^\top\boldsymbol{X}^\top\mathbf{R}_t\boldsymbol{X} + \alpha\sum_{t\in[T]}\boldsymbol{x}_t\boldsymbol{p}^\top\boldsymbol{X}^\top\mathbf{R}_t\boldsymbol{X},$$

where

$$\mathbf{R}_t = \mathbf{R}_0 := \frac{1}{T}\cdot\boldsymbol{I}_T - \frac{1}{T^2}\cdot\mathbb{1}_T\mathbb{1}_T^\top, \ \ \forall t\in[T].$$

Thus,

$$y\langle \nabla_{\boldsymbol{W}}\Phi\left(\boldsymbol{X};\boldsymbol{\theta}^{(1)}\right), \mathtt{sign}(\alpha)\overline{\boldsymbol{W}}_\star\rangle$$
$$= \frac{|\alpha|}{\|\boldsymbol{W}_\star\|_F}\left(\frac{\zeta}{2}\underbrace{y\sum_{t\in[T]}\boldsymbol{u}_\star^\top\boldsymbol{X}^\top\mathbf{R}_0\boldsymbol{r}_t(\boldsymbol{X};\boldsymbol{W}_\star)}_{\mathrm{Term_I}} + \underbrace{y\left\langle\sum_{t\in[T]}\boldsymbol{x}_t\boldsymbol{p}^\top\boldsymbol{X}^\top\mathbf{R}_0\boldsymbol{X}, \boldsymbol{W}_\star\right\rangle}_{\mathrm{Term_{II}}}\right), \tag{107}$$

where we set for convenience:

$$\boldsymbol{r}_t(\boldsymbol{X};\boldsymbol{W}_\star) = \boldsymbol{X}\boldsymbol{W}_\star^\top\boldsymbol{x}_t \in \mathbb{R}^T.$$

To compute $\boldsymbol{r}_t(\boldsymbol{X};\boldsymbol{W}_\star)$, we consider two cases where row corresponds to a signal relevant of noisy token. We denote the $t' \in [T]$ entry of $\boldsymbol{r}_t$ as $[\boldsymbol{r}_t]_{t'} \in \mathbb{R}$.

Case 1. Relevance scores of signal tokens: Consider signal token $t \in \mathcal{R}$ so that $\boldsymbol{x}_t = \boldsymbol{\mu}_y$. Using orthogonality in Assumption 3 we can compute for all $t' \in [T]$

$$t \in \mathcal{R} : [\boldsymbol{r}_t]_{t'} = \begin{cases} S^4 & , t' \in \mathcal{R}, \\ S^2(\boldsymbol{\mu}_y^\top\boldsymbol{z}_{t'}) & , t' \in \mathcal{R}^c. \end{cases} \tag{108}$$

Therefore, again using Assumption 3,

$$t \in \mathcal{R} : [\boldsymbol{r}_t]_{t'} \begin{cases} = S^4 & , t' \in \mathcal{R}, \\ \leq S^2 Z_\mu & , t' \in \mathcal{R}^c. \end{cases} \tag{109}$$

Case 2. Relevance scores of noisy tokens: Similar to the calculations above, using Assumption 3 for parameters $\boldsymbol{W}^\star$ as in (126) it holds for noisy tokens $t \in \mathcal{R}^c$ that

$$t \in \mathcal{R}^c : [\boldsymbol{r}_t]_{t'} = \begin{cases} S^4 + S^2(\boldsymbol{\mu}_y^\top\boldsymbol{z}_t) + S^2(\sum_\ell \boldsymbol{\nu}_\ell^\top\boldsymbol{z}_t) & , t' \in \mathcal{R} \\ (\boldsymbol{\mu}_+^\top\boldsymbol{z}_t)(\boldsymbol{\mu}_+^\top\boldsymbol{z}_{t'}) + (\boldsymbol{\mu}_-^\top\boldsymbol{z}_t)(\boldsymbol{\mu}_-^\top\boldsymbol{z}_{t'}) & , t' \in \mathcal{R}^c \\ + \sum_\ell(\boldsymbol{\nu}_\ell^\top\boldsymbol{z}_t)(\boldsymbol{\mu}_+^\top\boldsymbol{z}_{t'}) + \sum_\ell(\boldsymbol{\nu}_\ell^\top\boldsymbol{z}_t)(\boldsymbol{\mu}_-^\top\boldsymbol{z}_{t'}) + S^2(\boldsymbol{\mu}_+ + \boldsymbol{\mu}_-)^\top\boldsymbol{z}_{t'} \end{cases} \tag{110}$$

Now, we can start to bound each of the two terms in (107) separately below.

Bounding Term$_\mathrm{I}$: Recall the data matrix complies with Assumption 3, hence.

$$[\boldsymbol{X}\boldsymbol{u}_\star]_{t'} = \begin{cases} yS^2 & ,t' \in \mathcal{R} \\ \boldsymbol{z}_{t'}^\top(\boldsymbol{\mu}_+ - \boldsymbol{\mu}_-) & ,t' \in \mathcal{R}^c \end{cases}. \tag{111}$$

Using this, we can compute:

$$\frac{y}{T}\sum_{t\in[T]} \boldsymbol{u}_\star^\top \boldsymbol{X}^\top \boldsymbol{r}_t(\boldsymbol{X};\boldsymbol{W}^*) = \frac{y}{T}\left\{\sum_{t\in\mathcal{R}} \boldsymbol{u}_\star^\top \boldsymbol{X}^\top \boldsymbol{r}_t(\boldsymbol{X};\boldsymbol{W}_\star) + \sum_{t\in\mathcal{R}^c} \boldsymbol{u}_\star^\top \boldsymbol{X}^\top \boldsymbol{r}_t(\boldsymbol{X};\boldsymbol{W}_\star)\right\}$$

$$= \frac{y}{T}\left\{\sum_{t\in\mathcal{R}}\left(\sum_{t'\in\mathcal{R}}[\boldsymbol{X}\boldsymbol{u}_\star]_{t'}\,\boldsymbol{r}_t(\boldsymbol{X};\boldsymbol{W}_\star)_{t'} + \sum_{t'\in\mathcal{R}^c}[\boldsymbol{X}\boldsymbol{u}_\star]_{t'}\,\boldsymbol{r}_t(\boldsymbol{X};\boldsymbol{W}_\star)_{t'}\right)\right.$$

$$\left. + \sum_{t\in\mathcal{R}^c}\left(\sum_{t'\in\mathcal{R}}[\boldsymbol{X}\boldsymbol{u}_\star]_{t'}\,\boldsymbol{r}_t(\boldsymbol{X};\boldsymbol{W}_\star)_{t'} + \sum_{t'\in\mathcal{R}^c}[\boldsymbol{X}\boldsymbol{u}_\star]_{t'}\,\boldsymbol{r}_t(\boldsymbol{X};\boldsymbol{W}_\star)_{t'}\right)\right\}$$

$$= \frac{y}{T}\left\{\zeta T\left(\zeta T yS^2 S^4 + \sum_{t'\in\mathcal{R}^c}\boldsymbol{z}_{t'}^\top(\boldsymbol{\mu}_+ - \boldsymbol{\mu}_-)S^2(\boldsymbol{\mu}_y^\top \boldsymbol{z}_{t'})\right)\right.$$

$$+ \sum_{t\in\mathcal{R}^c}\left[\zeta T yS^2 \cdot \left(S^4 + S^2(\boldsymbol{\mu}_y^\top \boldsymbol{z}_t) + S^2(\sum_{j_t}\boldsymbol{\nu}_{j_t}^\top \boldsymbol{z}_t)\right)\right.$$

$$+ \sum_{t'\in\mathcal{R}^c}\boldsymbol{z}_{t'}^\top(\boldsymbol{\mu}_+ - \boldsymbol{\mu}_-)\cdot\left((\boldsymbol{\mu}_+^\top \boldsymbol{z}_t)(\boldsymbol{\mu}_+^\top \boldsymbol{z}_{t'}) + (\boldsymbol{\mu}_-^\top \boldsymbol{z}_t)(\boldsymbol{\mu}_-^\top \boldsymbol{z}_{t'})\right.$$

$$\left.\left.\left. + \sum_{j_t}(\boldsymbol{\nu}_{j_t}^\top \boldsymbol{z}_t)(\boldsymbol{\mu}_+^\top \boldsymbol{z}_{t'}) + \sum_{j_t}(\boldsymbol{\nu}_{j_t}^\top \boldsymbol{z}_t)(\boldsymbol{\mu}_-^\top \boldsymbol{z}_{t'}) + S^2(\boldsymbol{\mu}_+ + \boldsymbol{\mu}_-)^\top \boldsymbol{z}_{t'}\right)\right]\right\}.$$

Further using the noise bounds in Assumption 3, and $\boldsymbol{r}_t(\boldsymbol{X};\boldsymbol{W}_\star)$ from (127), (110) we have:

$$\frac{y}{T}\sum_{t\in[T]} \boldsymbol{u}_\star^\top \boldsymbol{X}^\top \boldsymbol{r}_t(\boldsymbol{X};\boldsymbol{W}_\star)$$

$$\geq \zeta T\left[\zeta S^6 - 2(1-\zeta)Z_\mu^2 S^2\right] + (1-\zeta)T\left[\zeta S^6 - \zeta(Z_\mu + Z_\nu)S^4 - 2Z_\mu(1-\zeta)(2Z_\mu^2 + 2Z_\mu Z_\nu + 2Z_\mu S^2)\right]$$

$$\geq T\left[\zeta S^6 - \zeta(1-\zeta)(Z_\mu + Z_\nu)S^4 - 2(1-\zeta)(\zeta + 2(1-\zeta))Z_\mu^2 S^2 - 4(1-\zeta)^2 Z_\mu^2(Z_\mu + Z_\nu)\right]. \tag{112}$$

For the second part of Term$_\mathrm{I}$:

$$-\frac{y}{T^2}\sum_{t\in[T]} \boldsymbol{u}_\star^\top \boldsymbol{X}^\top \mathbb{1}_T \mathbb{1}_T^\top \boldsymbol{r}_t(\boldsymbol{X};\boldsymbol{W}_\star) = -\frac{y}{T^2}\sum_{t\in[T]} \mathbb{1}_T^\top \boldsymbol{X}\boldsymbol{u}_\star\,\mathbb{1}_T^\top \boldsymbol{r}_t(\boldsymbol{X};\boldsymbol{W}_\star)$$

$$= -\frac{1}{T^2}\left\{\sum_{t'\in\mathcal{R}}y^2 S^2 + \sum_{t'\in\mathcal{R}^c}y\boldsymbol{z}_{t'}^\top(\boldsymbol{\mu}_+ - \boldsymbol{\mu}_-)\right\}\cdot\left\{\sum_{t\in\mathcal{R}}\left[\sum_{t'\in\mathcal{R}}S^4 + \sum_{t'\in\mathcal{R}^c}S^2 \boldsymbol{z}_{t'}^\top \boldsymbol{\mu}_y\right]\right.$$

$$+ \sum_{t\in\mathcal{R}^c}\left[\sum_{t'\in\mathcal{R}}\left(S^4 + S^2 \boldsymbol{z}_t^\top \boldsymbol{\mu}_y + S^2\sum_{j_t}\boldsymbol{\nu}_{j_t}^\top \boldsymbol{z}_t\right) + \sum_{t'\in\mathcal{R}^c}\left((\boldsymbol{z}_t^\top \boldsymbol{\mu}_+)(\boldsymbol{z}_{t'}^\top \boldsymbol{\mu}_+) + (\boldsymbol{z}_t^\top \boldsymbol{\mu}_-)\boldsymbol{\mu}_-\right)\right.$$

$$\left.\left. + (\sum_{j_t}\boldsymbol{\nu}_{j_t}^\top \boldsymbol{z}_t)(\boldsymbol{z}_{t'}^\top \boldsymbol{\mu}_+) + (\sum_{j_t}\boldsymbol{\nu}_{j_t}^\top \boldsymbol{z}_t)(\boldsymbol{z}_{t'}^\top \boldsymbol{\mu}_-) + S^2 \boldsymbol{z}_{t'}^\top(\boldsymbol{\mu}_+ + \boldsymbol{\mu}_-)\right)\right]\right\}.$$

Using Assumption 3 to simplify the second term:

$$-\frac{y}{T^2}\sum_{t\in[T]}\boldsymbol{u}_\star^\top\boldsymbol{X}^\top\mathbb{1}_T\mathbb{1}_T^\top\boldsymbol{r}_t(\boldsymbol{X};\boldsymbol{W}_\star)$$

$$\geq T\bigg[-\zeta^3 S^6-\zeta^2(1-\zeta)Z_\mu S^4-\zeta^2(1-\zeta)S^6-\zeta^2(1-\zeta)(Z_\mu+Z_\nu)S^4-2\zeta(1-\zeta)^2 Z_\mu S^4$$

$$-2\zeta(1-\zeta)^2 Z_\mu(Z_\mu+Z_\nu)S^2-2\zeta^2(1-\zeta)Z_\mu S^4-2\zeta(1-\zeta)^2 Z_\mu^2 S^2-2\zeta(1-\zeta)^2 Z_\mu S^4$$

$$-2\zeta(1-\zeta)^2 Z_\mu(Z_\mu+Z_\nu)S^2-4(1-\zeta)^3 Z_\mu^2 S^2-4(1-\zeta)^3 Z_\mu^2(Z_\mu+Z_\nu)\bigg].$$

Therefore,

$$-\frac{y}{T^2}\sum_{t\in[T]}\boldsymbol{u}_\star^\top\boldsymbol{X}^\top\mathbb{1}_T\mathbb{1}_T^\top\boldsymbol{r}_t(\boldsymbol{X};\boldsymbol{W}_\star)\geq T\bigg[-\zeta^2 S^6-\zeta(1-\zeta)(4Z_\mu+\zeta Z_\nu)S^4-2(1-\zeta)^2((2+\zeta)Z_\mu$$

$$+2\zeta Z_\nu)Z_\mu S^2-4(1-\zeta)^3 Z_\mu^2(Z_\mu+Z_\nu)\bigg].\tag{113}$$

Bounding Term$_{\mathrm{II}}$:

$$y\langle\sum_{t\in[T]}\boldsymbol{x}_t\boldsymbol{p}^\top\boldsymbol{X}^\top\mathbf{R}_0\boldsymbol{X},\boldsymbol{W}_\star\rangle\geq-\sum_{t\in[T]}\|\boldsymbol{x}_t\|\|\boldsymbol{p}\|\|\boldsymbol{X}^\top\mathbf{R}_0\boldsymbol{X}\|_F\|\boldsymbol{W}_\star\|_F$$

$$\geq-P(\zeta TS+(1-\zeta)T(S+Z))\|\mathbf{R}_0\|_F\|\boldsymbol{X}\|_F^2 S^2\sqrt{2(M+1)}$$

$$\geq-\sqrt{2(M+1)}S^2 PT^{5/2}(\zeta S+(1-\zeta)(S+Z))(\zeta S^2+(1-\zeta)(S+Z)^2).\tag{114}$$

Combining (112), (113), (114) in (107) we get:

$$y\langle\nabla_{\boldsymbol{W}}\Phi(\boldsymbol{X};\boldsymbol{\theta}^{(1)}),\boldsymbol{W}_\star\rangle$$

$$\geq\alpha\bigg\{\frac{T\zeta}{2}\bigg[\zeta S^6-\zeta(1-\zeta)(Z_\mu+Z_\nu)S^4-2(1-\zeta)(\zeta+2(1-\zeta))Z_\mu^2 S^2-4(1-\zeta)^2 Z_\mu^2(Z_\mu+Z_\nu)\bigg]$$

$$-\frac{T\zeta}{2}\bigg[\zeta^2 S^6+\zeta(1-\zeta)(4Z_\mu+\zeta Z_\nu)S^4+2(1-\zeta)^2((2+\zeta)Z_\mu+2\zeta Z_\nu)Z_\mu S^2+4(1-\zeta)^3 Z_\mu^2(Z_\mu+Z_\nu)\bigg]$$

$$-\sqrt{2(M+1)}S^2 PT^{5/2}(\zeta S+(1-\zeta)(S+Z))(\zeta S^2+(1-\zeta)(S+Z)^2)\bigg\}$$

$$=\alpha\bigg\{\frac{T(1-\zeta)\zeta}{2}\bigg[\zeta S^6-\zeta(5Z_\mu+(1+\zeta)Z_\nu)S^4-2((4-\zeta^2)Z_\mu+2\zeta(1-\zeta)Z_\nu)Z_\mu S^2$$

$$-4(1-\zeta)(2-\zeta)(Z_\mu+Z_\nu)Z_\mu^2\bigg]-\sqrt{2(M+1)}S^2 PT^{5/2}(S+(1-\zeta)Z)(\zeta S^2+(1-\zeta)(S+Z)^2)\bigg\}$$

$$\geq\alpha\bigg\{\frac{T(1-\zeta)\zeta}{2}\bigg[\zeta S^6-(5Z_\mu+2Z_\nu)S^4-4(2Z_\mu+Z_\nu)Z_\mu S^2-8(Z_\mu+Z_\nu)Z_\mu^2\bigg]$$

$$-\sqrt{2(M+1)}S^2 PT^{5/2}(S+Z)(S^2+(S+Z)^2)\bigg\}$$

$$\geq\alpha\bigg\{\frac{T(1-\zeta)\zeta}{2}\Big(\zeta S^6-7\bar{Z}S^4-12\bar{Z}^2 S^2-16\bar{Z}^3\Big)-2\sqrt{2(M+1)}PT^{5/2}S^2(S+Z)^3\bigg\},\tag{115}$$

where $\bar{Z} = Z_\mu \vee Z_\nu$. Using this, we get

$$
\begin{aligned}
\mu := \mathbb{E}_{\alpha \sim \text{Unif}(\pm 1)} y \left\langle \nabla_{\boldsymbol{W}} \Phi(\boldsymbol{X}; \boldsymbol{\theta}^{(1)}), \frac{\text{sign}(\alpha) \boldsymbol{W}_\star}{\|\boldsymbol{W}_\star\|_F} \right\rangle & \\
\geq \mathbb{E}_{\alpha \sim \text{Unif}(\pm 1)} \frac{|\alpha|}{\|\boldsymbol{W}_\star\|} & \left\{ \frac{T(1-\zeta)\zeta}{2} \left( \zeta S^6 - 7\bar{Z} S^4 - 12\bar{Z}^2 S^2 - 16\bar{Z}^3 \right) - 2\sqrt{2(M+1)} P T^{5/2} S^2 (S+Z)^3 \right\} \\
\geq \frac{T(1-\zeta)\zeta}{4\sqrt{2(M+1)}} & \left( \zeta S^4 - 7\bar{Z} S^2 - 12\bar{Z}^2 - 16\frac{\bar{Z}^3}{S^2} \right) - P T^{5/2} (S+Z)^3 .
\end{aligned}
\tag{116}
$$

Combining (106) and (116) concludes the proof. $\qquad \square$

**Lemma 14** (NTK Separability). *Assume initialization* $\widetilde{\boldsymbol{\theta}}^{(1)} = \text{concat}\left(\boldsymbol{\theta}_1^{(1)}, \ldots, \boldsymbol{\theta}_1^{(H)}\right)$ *as in* (102) *and IID* $\alpha_h \sim \text{Unif}(\pm 1)$ *for all* $h \in [H]$. *Recall* $\gamma_\star$ *and* $\boldsymbol{\theta}_\star(\cdot)$ *from Lemma 13 above. Set*

$$
\widetilde{\boldsymbol{\theta}}_\star = \frac{1}{\sqrt{H}} \text{concat}\left( \boldsymbol{\theta}_\star\left(\boldsymbol{\theta}_1^{(1)}\right), \ldots, \boldsymbol{\theta}_\star\left(\boldsymbol{\theta}_H^{(1)}\right) \right).
$$

*Let any* $(\boldsymbol{X}, y)$ *from DM1. Then, with probability at least* $1 - \delta$ *over the randomness of* $\alpha_h, h \in [H]$ *it holds*

$$
y \left\langle \nabla \widetilde{\Phi}\left(\boldsymbol{X}; \widetilde{\boldsymbol{\theta}}^{(1)}\right), \widetilde{\boldsymbol{\theta}}_\star \right\rangle \geq \gamma_\star - 2\left( 2R^3 T (S+P) + \sqrt{T} R \right) \sqrt{\frac{2\log(1/\delta)}{H}} .
$$

*Proof.* We start by expanding the empirical margin:

$$
y \left\langle \nabla \widetilde{\Phi}\left(\boldsymbol{X}; \widetilde{\boldsymbol{\theta}}^{(1)}\right), \widetilde{\boldsymbol{\theta}}_\star \right\rangle = \frac{1}{H} \sum_{h \in [H]} y \langle \nabla_{\boldsymbol{\theta}} \Phi\left(\boldsymbol{X}; \boldsymbol{\theta}_h^{(1)}\right), \boldsymbol{\theta}_\star(\boldsymbol{\theta}_h^{(1)}) \rangle =: \frac{1}{H} \sum_{h \in [H]} X_h .
\tag{117}
$$

Note that each summand $X_h$ defined above depends only on $\alpha_h, h \in [H]$. Thus, $\{X_h\}_{h \in [H]}$ are IID random variables because $\{\alpha_h\}_{h \in [H]}$ are IID random variables. Moreover, $X_h$ is almost-surely bounded satisfying

$$
|X_h| \leq \|\nabla_{\boldsymbol{\theta}} \Phi\left(\boldsymbol{X}; \boldsymbol{\theta}_h^{(1)}\right)\| \|\boldsymbol{\theta}_\star(\boldsymbol{\theta}_h^{(1)})\| \leq \sqrt{2T} R \left( 2R^2 \|\boldsymbol{\theta}_h^{(1)}\|_F + 1 \right) \leq 2\left( 2R^3 T (S+P) + \sqrt{T} R \right) .
\tag{118}
$$

where the second inequality follows by Proposition 1 and by the assumption $\|\boldsymbol{\theta}_\star(\boldsymbol{\theta}_h^{(1)})\|_2 = \sqrt{2}$ for all $h \in [H]$, and the last inequality follows because $\|\boldsymbol{\theta}_h^{(1)}\| \leq \frac{\zeta}{2}\sqrt{T}\|\boldsymbol{u}_\star\|_2 + \sqrt{T}\|\boldsymbol{p}\|_2 \leq \sqrt{T}(S+P)$.

Finally, note that $\mathbb{E}[X_h] \geq \gamma_\star$ from Lemma 13. Given these the desired claim follows by applying Hoeffding's inequality (see Fact 2) to (117).

$\qquad \square$

**Lemma 15** (Margin). *Define* $\bar{Z} := Z_\mu \vee Z_\nu$ *and*

$$
\gamma_\star := \frac{T(1-\zeta)\zeta}{4\sqrt{2(M+1)}} \left( \zeta S^4 - 7\bar{Z} S^2 - 12\bar{Z}^2 - 16\frac{\bar{Z}^3}{S^2} \right) - P T^{5/2} (S+Z)^3 + \frac{S\sqrt{T}}{\sqrt{2}} \left( \zeta - 2(1-\zeta)\frac{Z_\mu}{S^2} \right).
\tag{119}
$$

*Suppose*

$$
\sqrt{H} \geq 4 \cdot \frac{2R^3 T (S+P) + \sqrt{T} R}{\gamma_\star} \cdot \sqrt{2\log(n/\delta)} .
$$

*Then, with probability* $1 - \delta \in (0, 1)$, **P3** *holds with* $\gamma = \gamma_\star/2$.

*Proof.* The proof is straightforward by using union bound and plugging the condition of $H$ in the result of Lemma 14. $\qquad \square$

### D.3 Proof of Lemma 2

The proof of the lemma follows directly by combining the two lemmas below and using $\zeta \le 1$.

**Lemma 16.** *Fix any $h \in [H]$. Suppose zero initialization $\boldsymbol{\theta}_h^{(0)} = 0$ and consider first gradient step as in* (102). *It then holds that $\boldsymbol{W}_h^{(1)} = 0$ and $\boldsymbol{U}_h^{(1)} \in \mathbb{R}^{T \times d}$ has identical rows all equal to*

$$\underbrace{\frac{\zeta \alpha_h}{2} \boldsymbol{u}_\star + \frac{\zeta \alpha_h}{2} \left( \frac{1}{n} \sum_{i \in [n_1]} y_i \boldsymbol{\mu}_{y_i} - \boldsymbol{u}_\star \right)}_{\boldsymbol{p}_1} + \underbrace{\frac{\alpha_h(1-\zeta)}{2} \frac{1}{n} \sum_{i \in [n_1]} y_i \frac{1}{(1-\zeta)T} \sum_{t \in \mathcal{R}_i^c} (\boldsymbol{\nu}_{j_t} + \boldsymbol{z}_{i,t})}_{\boldsymbol{p}_2} . \tag{120}$$

*where recall that $\boldsymbol{u}_\star = \boldsymbol{\mu}_+ - \boldsymbol{\mu}_-$.*

*Proof.* We start by computing

$$n \nabla_{\boldsymbol{\theta}_h} \widehat{L}(\boldsymbol{\theta}_h^{(0)}) = \sum_{i \in [n_1]} \nabla_{\boldsymbol{\theta}_h} \ell(y_i \widetilde{\Phi}(\boldsymbol{X}; \boldsymbol{\theta}_h^{(0)})) = \sum_{i \in [n_1]} y_i \ell'(y_i \widetilde{\Phi}(\boldsymbol{X}; \boldsymbol{\theta}_h^{(0)})) \nabla_{\boldsymbol{\theta}_h} \widetilde{\Phi}(\boldsymbol{X}; \boldsymbol{\theta}_h^{(0)})$$

$$= \ell'(0) \sum_{i \in [n_1]} y_i \nabla_{\boldsymbol{\theta}_h} \widetilde{\Phi}(\boldsymbol{X}; \boldsymbol{\theta}_h^{(0)}) = \frac{\ell'(0)}{\sqrt{H}} \sum_{i \in [n_1]} y_i \nabla_{\boldsymbol{\theta}_h} \Phi_h(\boldsymbol{X}; \boldsymbol{\theta}_h^{(0)})$$

$$= \frac{1}{2\sqrt{H}} \sum_{i \in [n_1]} y_i \nabla_{\boldsymbol{\theta}_h} \Phi_h(\boldsymbol{X}; \boldsymbol{\theta}_h^{(0)})$$

where we used that $\Phi_h(\boldsymbol{X}; \boldsymbol{\theta}_h^{(0)}) = \widetilde{\Phi}_h(\boldsymbol{X}; \boldsymbol{\theta}_h^{(0)}) = 0$ because $\boldsymbol{U}_h = \boldsymbol{0}$ and also $\ell'(0) = 1/2$ for the logistic loss. Now, recall that $\Phi_h(\boldsymbol{X}; \theta_h) = \langle \boldsymbol{U}_h, \mathrm{ATTN}_h(\boldsymbol{X}; \boldsymbol{W}_h) \rangle$. Hence, $\nabla_{\boldsymbol{W}_h} \Phi_h(\boldsymbol{X}; \boldsymbol{\theta}_h^{(0)}) = \boldsymbol{0}$, which gives us $\boldsymbol{W}_h^{(1)} = \boldsymbol{W}_h^{(0)}$. Also,

$$\nabla_{\boldsymbol{U}_h} \Phi_h(\boldsymbol{X}; \boldsymbol{\theta}_h^{(0)}) = \mathrm{ATTN}(\boldsymbol{X}; \boldsymbol{W}_h^{(0)}) = \boldsymbol{\varphi}(0)\boldsymbol{X} = \frac{1}{T} \mathbb{1}_T \mathbb{1}_T^\top \boldsymbol{X} .$$

Hence, the $\tau$-th column of $\boldsymbol{U}_h^{(1)}$ becomes for all $\tau \in [T]$:

$$[\boldsymbol{U}_h^{(1)}]_{:,\tau} = \frac{1}{2nT} \alpha_h \sum_{i \in [n_1]} y_i \sum_{t \in [T]} \boldsymbol{x}_{i,t}$$

$$= \frac{\zeta}{2n} \alpha_h \sum_{i \in [n_1]} y_i \boldsymbol{\mu}_{y_i} + \frac{1}{2nT} \alpha_h \sum_{i \in [n_1]} y_i \sum_{t \in \mathcal{R}_i^c} (\boldsymbol{\nu}_{j_t} + \boldsymbol{z}_{i,t}) .$$

The claim of the lemma follows by rearranging the above. $\qquad\square$

**Lemma 17.** *Suppose labels are IID and equal probable, i.e. $y_i \sim \mathrm{Rad}(\pm 1)$. Then for the two terms $\boldsymbol{p}_1, \boldsymbol{p}_2$ in* (120) *it holds with probability at least $1 - 2\delta$ and absolute constant $C > 0$ over the randomness of labels that*

$$\|\boldsymbol{p}_1\| \le CS\left( \sqrt{\frac{d}{n_1}} + \sqrt{\frac{\log(1/\delta)}{n_1}} \right), \tag{121}$$

*and*

$$\|\boldsymbol{p}_2\| \le C(S + Z)\left( \sqrt{\frac{d}{n_1}} + \sqrt{\frac{\log(1/\delta)}{n_1}} \right). \tag{122}$$

*Proof.* For arbitrary $\|\boldsymbol{v}\| = 1$, let $X_v = \langle \boldsymbol{v}, \frac{1}{n} \sum_{i \in [n_1]} y_i \boldsymbol{\mu}_{y_i} \rangle$. Then,

$$\|X_v\|_{\psi_2} = \frac{1}{n} \| \sum_{i \in [n_1]} y_i \langle \boldsymbol{v}, \boldsymbol{\mu}_{y_i} \rangle \|_{\psi_2} \le \frac{C}{n} \sqrt{\sum_{i \in [n_1]} \|y_i \langle \boldsymbol{v}, \boldsymbol{\mu}_{y_i} \rangle\|_{\psi_2}^2} \le \frac{CS}{\sqrt{n_1}} .$$

where in the second inequality we used approximate rotation invariance of subgaussians and in the last step we used that for all $i \in [n_1]$, $|y_i \langle \boldsymbol{v}, \boldsymbol{\mu}_{y_i} \rangle| \leq S$, thus they are subgaussians with parameter $CS$. Further note that Note that $\mathbb{E}[X_v] = \langle \boldsymbol{v}, \boldsymbol{u}_\star \rangle$. Thus, by centering property of subgaussians $\langle \boldsymbol{v}, \boldsymbol{p}_1 \rangle$ is also subgaussian with same constant $CS/\sqrt{n_1}$. Since this holds for all $\boldsymbol{v}$ on the sphere, we conclude that $\boldsymbol{p}_1$ is $CS/\sqrt{n_1}$-subgaussian. From this, we can directly apply Fact 1 for concentration of Euclidean norm of random vectors to arrive at (121).

We can follow exactly same steps to prove (122) for $\boldsymbol{p}_2$. The only difference is noting that for all $i \in [n_1]$ and unit norm $\boldsymbol{v}$:

$$\left| y_i \frac{1}{(1-\zeta)T} \sum_{t \in \mathbb{R}_i^c} \langle \boldsymbol{\nu}_{j_t} + \boldsymbol{z}_{i,t}, \boldsymbol{v} \rangle \right| \leq S + Z \,.$$

$\square$

## E  Optimal Model

We first restate the optimal parameters $\boldsymbol{\theta}_{\text{opt}} = (\boldsymbol{U}_{\text{opt}}, \boldsymbol{W}_{\text{opt}})$:

$$\boldsymbol{U}_{\text{opt}} := \frac{1}{S\sqrt{T}} \mathbb{1}_d (\boldsymbol{\mu}_+ - \boldsymbol{\mu}_-)^\top \,, \tag{123}$$

$$\boldsymbol{W}_{\text{opt}} := \frac{1}{S^2 \sqrt{2(M+1)}} \Big( \boldsymbol{\mu}_+ \boldsymbol{\mu}_+^\top + \boldsymbol{\mu}_- \boldsymbol{\mu}_-^\top + \sum_{\ell \in [M]} \boldsymbol{\nu}_\ell (\boldsymbol{\mu}_+ + \boldsymbol{\mu}_-)^\top \Big) \,, \tag{124}$$

normalized so that $\|\boldsymbol{\theta}_{\text{opt}}\|_F = \sqrt{2}$.

The following lemma about saturation in softmax scores is used to prove Proposition 3.

**Lemma 18** (Softmax saturation). *Let relevance-scores vector $\boldsymbol{r} = [r_1, \dots, r_T] \in \mathbb{R}^T$ be such that for some $L \in [T]$ and $A, B \in \mathbb{R}$:*

$$r_1 \geq r_2 \geq \dots \geq r_L \geq A \qquad \text{and} \qquad B \geq \max\{r_i \,|\, i = L+1, \dots, T\}.$$

*Further assume $A > 2B$. Fix any $\epsilon > 0$ and*

$$\Gamma \geq \frac{2}{A} \log \Big( \frac{T/L - 1}{\epsilon} \Big).$$

*Then, for the attention weights $\boldsymbol{a} = [a_1, \dots, a_T] := \boldsymbol{\varphi}(\Gamma \boldsymbol{r}) \in \mathbb{R}^T$ it holds that*

$$0 \leq 1 - \sum_{i \in [L]} a_i = \sum_{i=L+1}^T a_i \leq \epsilon \,. \tag{125}$$

*Proof.* For convenience denote $D := \sum_{j \in [T]} e^{\Gamma r_j}$. Note that $D \geq L e^{\Gamma A}$. Consider any $i > L$. Then,

$$a_i = \frac{e^{\Gamma r_i}}{D} \leq \frac{e^{\Gamma B}}{D} \leq \frac{e^{\Gamma B}}{L e^{\Gamma A}} = \frac{1}{L e^{\Gamma(A-B)}} \leq \frac{1}{L e^{\Gamma A/2}} \,.$$

Suppose $\Gamma \geq \frac{2}{A} \log \big( \frac{C}{\epsilon} \big)$, which ensures that

$$e^{\Gamma A/2} \geq C/\epsilon$$

Setting $C = \frac{T-L}{L} = \frac{T}{L} - 1$, and combining the above two displays yields the desired:

$$a_i \leq \frac{\epsilon}{T-L}, \qquad i > L \,.$$

Thus, $\sum_{i>L} a_i \leq \epsilon$. The proof is complete by recalling that $\sum_{i \in [T]} a_i = 1$, hence $\sum_{i \in [L]} a_i \geq 1 - \epsilon$. $\square$

### E.1 Proof of Proposition 3

First, we compute the attention matrix when the parameter $\boldsymbol{W}$ is set to the value below:

$$\boldsymbol{W} = \boldsymbol{\mu}_+\boldsymbol{\mu}_+^\top + \boldsymbol{\mu}_-\boldsymbol{\mu}_-^\top + \sum_{\ell \in [M]} \boldsymbol{\nu}_\ell(\boldsymbol{\mu}_+ + \boldsymbol{\mu}_-)^\top. \tag{126}$$

For convenience, use the notation $\boldsymbol{r}_t^\top := \boldsymbol{x}_t^\top \boldsymbol{W} \boldsymbol{X}^\top \in \mathbb{R}^T$ for the $t$-th row of matrix used to find attention scores. Similar to the proof of Lemma 13 we consider two cases where row corresponds to a signal relevant of noisy token. We denote the $t' \in [T]$ entry of $\boldsymbol{r}_t$ as $[\boldsymbol{r}_t]_{t'} \in \mathbb{R}$.

Case 1. Relevance scores of signal tokens: Consider signal token $t \in \mathcal{R}$ so that $\boldsymbol{x}_t = \boldsymbol{\mu}_y$. Then for weights $\boldsymbol{W}$ in (126), and using orthogonality in Assumption 3 we can compute for all $t' \in [T]$

$$t \in \mathcal{R} : [\boldsymbol{r}_t]_{t'} = \begin{cases} S^4 & , t' \in \mathcal{R}, \\ S^2(\boldsymbol{\mu}_y^\top \boldsymbol{z}_{t'}) & , t' \in \mathcal{R}^c. \end{cases}$$

Therefore, again using Assumption 3,

$$t \in \mathcal{R} : [\boldsymbol{r}_t]_{t'} \begin{cases} = S^4 & , t' \in \mathcal{R}, \\ \leq S^2 Z_\mu & , t' \in \mathcal{R}^c. \end{cases} \tag{127}$$

Case 2. Relevance scores of noisy tokens: Similar to the calculations above, using Assumption 3 for parameters $\boldsymbol{W}$ as in (126) it holds for noisy tokens $t \in \mathcal{R}^c$ that

$$t \in \mathcal{R}^c : [\boldsymbol{r}_t]_{t'} = \begin{cases} S^4 + S^2(\boldsymbol{\mu}_y^\top \boldsymbol{z}_t) + S^2(\sum_\ell \boldsymbol{\nu}_\ell^\top \boldsymbol{z}_t) & , t' \in \mathcal{R} \\ (\boldsymbol{\mu}_+^\top \boldsymbol{z}_t)(\boldsymbol{\mu}_+^\top \boldsymbol{z}_{t'}) + (\boldsymbol{\mu}_-^\top \boldsymbol{z}_t)(\boldsymbol{\mu}_-^\top \boldsymbol{z}_{t'}) & , t' \in \mathcal{R}^c \\ + \sum_\ell (\boldsymbol{\nu}_\ell^\top \boldsymbol{z}_t)(\boldsymbol{\mu}_+^\top \boldsymbol{z}_{t'}) + \sum_\ell (\boldsymbol{\nu}_\ell^\top \boldsymbol{z}_t)(\boldsymbol{\mu}_-^\top \boldsymbol{z}_{t'}) + S^2(\boldsymbol{\mu}_+ + \boldsymbol{\mu}_-)^\top \boldsymbol{z}_{t'} \end{cases}$$

Therefore, using the noise bound assumptions, we have

$$t \in \mathcal{R}^c : [\boldsymbol{r}_t]_{t'} \begin{cases} \geq S^4 - S^2(Z_\mu + Z_\nu) & , t' \in \mathcal{R} \\ \leq 2Z_\mu^2 + 2Z_\mu Z_\nu + 2S^2 Z_\mu & , t' \in \mathcal{R}^c. \end{cases} \tag{128}$$

Combining the above two cases, specifically Equations (127) and (128), we find that for all $t \in [T]$ the relevance-score vectors $\boldsymbol{r}_t$ are such that

$$t \in [T] : [\boldsymbol{r}_t]_{t'} \begin{cases} \geq S^4 - S^2(Z_\mu + Z_\nu) := A & , t' \in \mathcal{R} \\ \leq 2(Z_\mu^2 + Z_\mu Z_\nu + S^2 Z_\mu) := B & , t' \in \mathcal{R}^c, \end{cases} \tag{129}$$

where we defined the parameters $A$ and $B$ for convenience. Note from assumption that

$$Z_\mu = Z_\nu \leq \frac{S^2}{8} \implies A \geq \frac{3}{4}S^4 > \frac{1.25}{4}S^4 \geq 2B.$$

Thus, the conditions of Lemma 18 hold for $L = |\mathcal{R}| = \zeta T$. Applying the lemma we can immediately conclude that for

$$\Gamma_* \geq \frac{8}{3S^4} \log\left(\frac{\zeta^{-1} - 1}{\epsilon}\right) \geq \frac{2}{A} \log\left(\frac{\zeta^{-1} - 1}{\epsilon}\right).$$

it holds:

$$\forall t \in [T] : 0 \leq 1 - \sum_{t' \in \mathcal{R}} [\boldsymbol{\varphi}(\Gamma_* \boldsymbol{r}_t)]_{t'} = \sum_{t' \in \mathcal{R}^c} [\boldsymbol{\varphi}(\Gamma_* \boldsymbol{r}_t)]_{t'} \leq \epsilon. \tag{130}$$

Now, recall that

$$\boldsymbol{W}_\star = \widetilde{\Gamma}\boldsymbol{W} \qquad \text{for} \quad \widetilde{\Gamma} = \Gamma/(S^2\sqrt{2(M+1)})\,.$$

Thus, it holds for all $t \in [T]$ that $\boldsymbol{\varphi}(\boldsymbol{x}_t^\top \boldsymbol{W}_\star \boldsymbol{X}^T) = \boldsymbol{\varphi}(\widetilde{\Gamma}\boldsymbol{r}_t)$. Combining this with (130) and the proposition's assumption on $\Gamma$ (satisfying $\widetilde{\Gamma} \geq \Gamma_\star$), we have found that

$$\forall t \in [T] : 0 \leq 1 - \sum_{t' \in \mathcal{R}} \left[\boldsymbol{\varphi}(\boldsymbol{x}_t^\top \boldsymbol{W}_\star \boldsymbol{X}^T)\right]_{t'} = \sum_{t' \in \mathcal{R}^c} \left[\boldsymbol{\varphi}(\boldsymbol{x}_t^\top \boldsymbol{W}_\star \boldsymbol{X}^T)\right]_{t'} \leq \epsilon\,. \tag{131}$$

In the rest of the proof, we use (131) to lower-bound the margin:

$$y\Phi(\boldsymbol{X};\boldsymbol{\theta}_\star) = y\langle \boldsymbol{U}_\star, \mathrm{ATTN}(\boldsymbol{X};\boldsymbol{W}_\star)\rangle = \frac{y}{S\sqrt{2T}}\sum_{t\in[T]}\sum_{t'\in[T]}\left[\phi(\boldsymbol{x}_t^\top\boldsymbol{W}_\star\boldsymbol{X}^T)\right]_{t'}(\boldsymbol{\mu}_+ - \boldsymbol{\mu}_-)^\top\boldsymbol{x}_{t'} =: \frac{y}{S\sqrt{2T}}\sum_{t\in[T]}\psi_t\,,$$

where we defined $\psi_t, t \in [T]$ for convenience. For any $t \in [T]$, we have

$$\begin{aligned}
\psi_t &= \sum_{t'\in\mathcal{R}}\left[\phi(\boldsymbol{x}_t^\top\boldsymbol{W}_\star\boldsymbol{X}^T)\right]_{t'}yS^2 + \sum_{t'\in\mathcal{R}^c}\left[\phi(\boldsymbol{x}_t^\top\boldsymbol{W}_\star\boldsymbol{X}^T)\right]_{t'}(\boldsymbol{\mu}_+ - \boldsymbol{\mu}_-)^\top\boldsymbol{z}_{t'} \\
&\geq yS^2\sum_{t'\in\mathcal{R}}\left[\phi(\boldsymbol{x}_t^\top\boldsymbol{W}_\star\boldsymbol{X}^T)\right]_{t'} - 2Z_\mu\sum_{t'\in\mathcal{R}^c}\left[\phi(\boldsymbol{x}_t^\top\boldsymbol{W}_\star\boldsymbol{X}^T)\right]_{t'} \\
&\geq yS^2(1-\epsilon) - 2\epsilon Z_\mu\,.
\end{aligned}$$

Putting the last two displays together and using $y^2 = 1$ completes the proof of the proposition.

## F  Linear Model

To gain additional insights into the classification of the data model DM1 and also contrast our results to a simplified model, we examine here a linear classifier:

$$\Phi_{\mathrm{lin}}(\boldsymbol{X};\boldsymbol{U}) = \langle \boldsymbol{U}, \boldsymbol{X}\rangle\,.$$

For this linear model, consider the oracle classifier

$$\boldsymbol{U}_\star = \frac{1}{S\sqrt{2T}}\mathbb{1}_T\boldsymbol{u}_\star^\top, \quad \text{with} \quad \boldsymbol{u}_\star = \boldsymbol{\mu}_+ - \boldsymbol{\mu}_-\,,$$

and normalization such that $\|\boldsymbol{U}_\star\|_F = 1$. By using Assumption 3, almost surely for all examples $(\boldsymbol{X}, y)$ the margin of the oracle classifier is lower bounded by:

$$\begin{aligned}
y\Phi_{\mathrm{lin}}(\boldsymbol{X};\boldsymbol{U}_\star) &= \frac{1}{S\sqrt{2T}}\Big(|\mathcal{R}|\cdot S^2 + y\sum_{t\in\mathcal{R}^c}\langle\boldsymbol{\mu}_+ - \boldsymbol{\mu}_-, \boldsymbol{z}_t\rangle\Big) \\
&\geq \frac{1}{S\sqrt{2T}}\Big(|\mathcal{R}|\cdot S^2 - 2|\mathcal{R}^c|\cdot Z_\mu\Big) = \frac{S\sqrt{T}}{\sqrt{2}}\Big(\zeta - 2(1-\zeta)\frac{Z_\mu}{S^2}\Big) =: \gamma_{\mathrm{lin}}\,. \tag{132}
\end{aligned}$$

## G  Experiments

In this section we provide some experiments discussing the role of number of heads $H$ in the training dynamics on synthetic data models.

**Data Model DM1**  We set the number of tokens $T = 10$ and sparsity level $\zeta = 0.1$. We set $\{\boldsymbol{\mu}_+, \boldsymbol{\mu}_-, \boldsymbol{\nu}_1, \boldsymbol{\nu}_2, ..., \boldsymbol{\nu}_M\}$ as the canonical basis vectors in $\mathbb{R}^d$, with $d = 4, M = 2$ and signal strength $S = 2$. Noisy tokens $\boldsymbol{z}$ are sampled from a Gaussian $\mathcal{N}(0, \sigma^2\boldsymbol{I}_d)$, with $\sigma = 0.1$. We use $n = 100$ training samples in each experiment and evaluate on a test set of size 300 (total 5 trials). All models are initialized as $\widetilde{\boldsymbol{\theta}}^{(0)} = \boldsymbol{0}$.

Figure 2 shows the effect of increasing the number of heads when running GD with constant step-size $\eta = 1.0$ and data generated from data model DM1. Notice that rate of train loss decay reduces as we increase $H$, highlighting a potential downside of overparameterization. A similar observation has been recently noted by

Xu & Du (2023) when optimizing with GD to learn a single neuron with ReLU activation under Gaussian input. We also observe that at least for smaller $H$, GD indeed achieves margin $\gamma_{\text{attn}}$. It is worth noting that these observations do not contradict our theoretical findings in Theorems 1 and 2 which guarantee training convergence and generalization decay given sufficiently large number of heads $H$, without making an explicit connection between the rates of convergence as we increase $H$. We also test how these rates change if we scale step-size as $\eta = \mathcal{O}\left(\sqrt{H}\right)$ for GD (Figure 3, left) or optimize with Adam (Figure 3, right) using constant step-size $\eta = 0.06$. It is interesting to observe that in both these cases, the convergence speeds up for larger $H$, especially when optimizing with Adam, but somewhat strangely the margin attained by GD for larger $H$ continues to fall away from $\gamma_{\text{attn}}$. Further, note that our theory only covers step-size $\eta = \mathcal{O}\left(1\right)$ and the trends observed in Figure 3 with $\eta = \mathcal{O}\left(\sqrt{H}\right)$ for GD fall outside this regime. In essence, we believe that it would be interesting future work to see how well these observations generalize to different datasets and develop theory that explains the relation of overparameterization to rates of convergence.

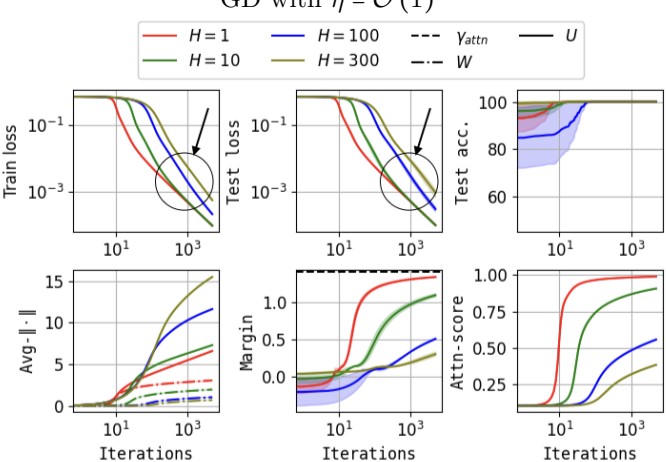

Figure 2: **For data model DM1**. Effect of number of heads $H$ on convergence rates when trained with GD for constant step-size $\eta = \mathcal{O}\left(1\right)$. The average $\|\cdot\|$ illustrates $1/H$ and $1/\sqrt{H}$ average for $\boldsymbol{W}$ and $\boldsymbol{U}$ across heads, respectively. Attn-score denotes the softmax scores for the relevant tokens averaged across all train samples and heads. The average $\|\boldsymbol{W}\|$ indicates the saturation of softmax scores and consequently the token-selection (attn-score), and the average $\|\boldsymbol{U}\|$ controls the loss behaviour. Results demonstrate that overparameterization slows down GD with constant step-size. The circled area shows a $\mathcal{O}(1/t)$ trend similar to what our training and generalization bounds predict.

**Planted data model** Fix some $\boldsymbol{W}^* \in \mathbb{R}^{d \times d}$, $\boldsymbol{U}^* \in \mathbb{R}^{T \times d}$. Entries within $\boldsymbol{X}$ are sampled IID $X_{ij} \sim \mathcal{N}(0,1)$, $\forall i \in [T]$, $j \in [d]$. Given such an $\boldsymbol{X} \in \mathbb{R}^{T \times d}$, generate the label $y$ using $\boldsymbol{W}^*$ as the attention matrix and $\boldsymbol{U}^*$ as the projection classifier:

$$y = \texttt{sign}(\Phi(\boldsymbol{X}; \boldsymbol{W}^*, \boldsymbol{U}^*)) = \texttt{sign}(\langle \boldsymbol{U}^*, \boldsymbol{\varphi}(\boldsymbol{X}\boldsymbol{W}^*\boldsymbol{X}^\top)\boldsymbol{X}\rangle). \tag{DM2}$$

Data generated using model DM2 is used to train a multi-head self-attention model as given in equation (2). Such a *teacher-student* setting (train the student network to learn the ground truth parent) has been well explored in the past in the context of neural networks (Zhou et al., 2021; Safran et al., 2021). We set $d = 5$, $T = 10$. The train set contains $n = 1000$ samples in each experiment and we evaluate on a test set of size 3000. Each result is averaged over 5 trials. For numerical ease, while generating (example, label) pairs we drop the samples for which |output logit| $\leq \gamma_{\text{attn}}$, where we call $\gamma_{\text{attn}} > 0$ to be margin for the data model. We set $\gamma_{\text{attn}} = 0.2$ in all the experiments. All models are initialized as $\widetilde{\boldsymbol{\theta}}^{(0)} = \boldsymbol{0}$. From Figure 4 we observe that overparameterization somewhat improves convergence speeds for GD with step-size $\mathcal{O}\left(\sqrt{H}\right)$, similar to tokenized mixture model (Figure 3, left). Addition of momentum significantly helps speeding up convergence (see Figure 5, left) and so does optimizing with Adam (Figure 5, right). Interesting to note that all the models reach the expected margin $\gamma_{\text{attn}}$ which was not the case for large $H$ for the tokenized mixture model. Further, we can observe that initializing at $\widetilde{\boldsymbol{\theta}}^{(0)} = \boldsymbol{0}$, all the optimizers find the planted model.

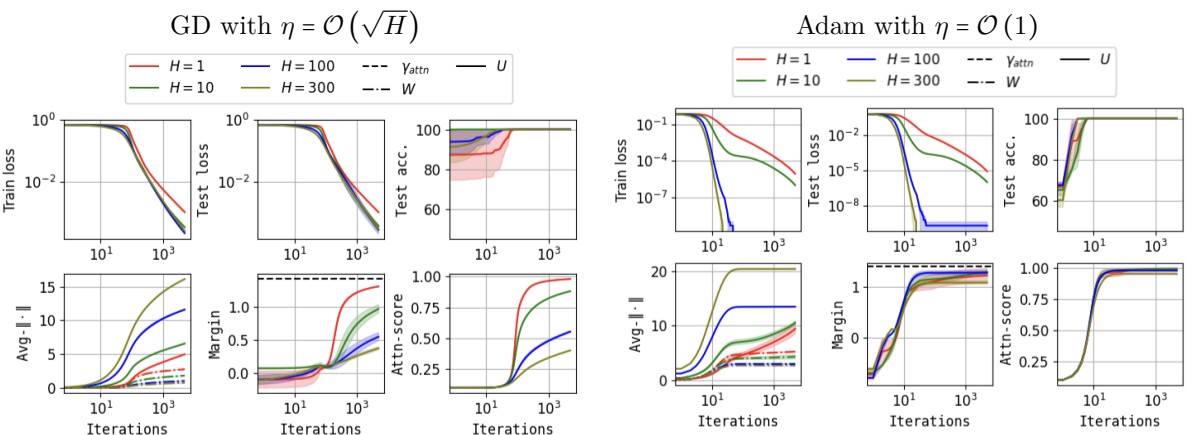

Figure 3: **For data model DM1**. Effect of number of heads $H$ on convergence rates when (left) trained with GD when scaling step-size as $\eta = \mathcal{O}\left(\sqrt{H}\right)$; (right) trained with Adam with constant step-size $\eta = \mathcal{O}\left(1\right)$. Quantities plotted are same as in Figure 2. Results demonstrate that overparameterization speeds-up with train and test loss convergence in both the scenarios.

**SST2 dataset** We conduct an additional experiment on a simple real-world dataset. The SST2 dataset (Socher et al., 2013) consists of sentences, with each sentence having a associated binary label to classify the sentiment. We fine-tune RoBERTa based models with varying number of heads using AdamW (Loshchilov & Hutter, 2019) optimizer with a learning rate of $5e-6$. We train all the models for 5 epochs, with the batch-size set to 32. We use the Hugging Face `pytorch-transformers` implementation of the `roberta-base` model, with pretrained weights (Liu et al., 2019). In Figure 6 we see that increasing the number of heads speeds up the optimization and generalization. This behaviour is similarly observed for GD with momentum and Adam in Figure 5.

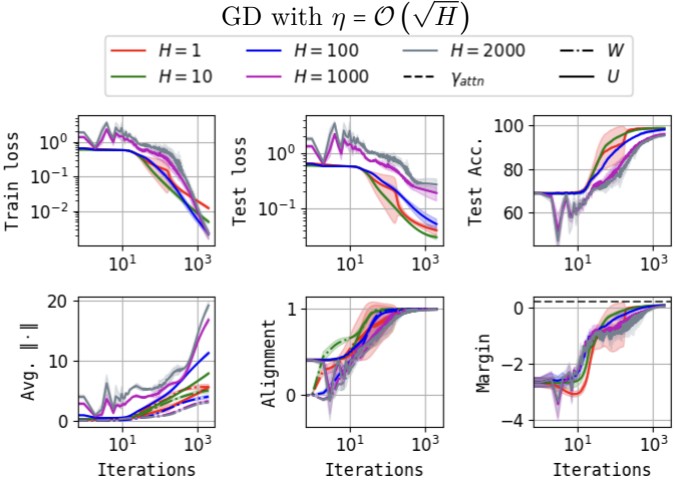

Figure 4: **For data model DM2**. Effect of number of heads $H$ on convergence rates when trained with GD when scaling step-size as $\eta = \mathcal{O}\left(\sqrt{H}\right)$. See Figure 2 caption for to get more context on average $\|\cdot\|$. Alignment of $\boldsymbol{W}$ with the planted-head $\widetilde{\boldsymbol{W}}^{\star}$ at any iteration $k$ is given by $\frac{\langle \widetilde{\boldsymbol{W}}_k, \widetilde{\boldsymbol{W}}^{\star}\rangle}{\|\widetilde{\boldsymbol{W}}_k\|\|\widetilde{\boldsymbol{W}}^{\star}\|}$, where $\widetilde{\boldsymbol{W}}^{\star} := \text{concat}\left(\{\boldsymbol{W}^{\star}\}_{h\in[H]}\right)$ contains $\boldsymbol{W}^{\star}$ repeated $H$ times. Alignment between $\widetilde{\boldsymbol{U}}$ and $\widetilde{\boldsymbol{U}}^{\star}$ is computed similarly.

# H    Related work

This section elaborates on the paragraph on related work in Section 1.

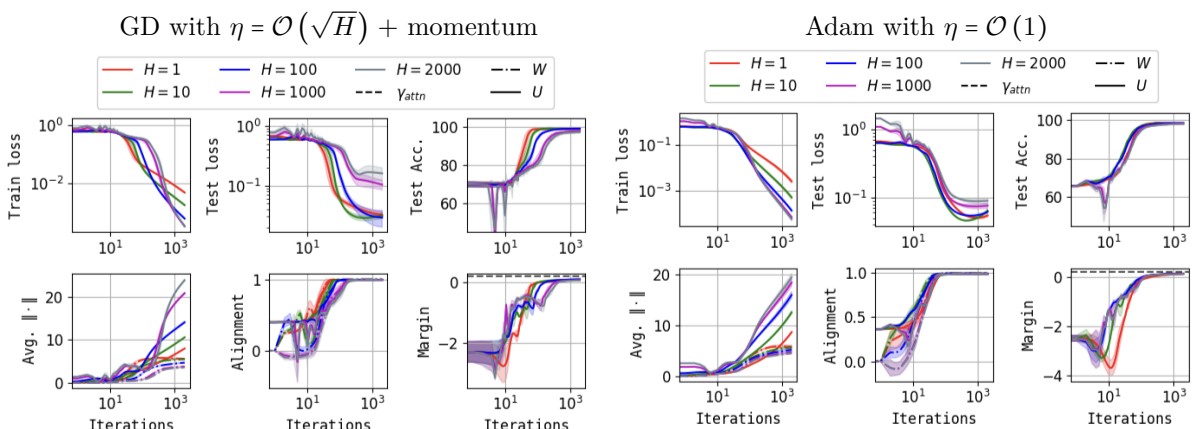

Figure 5: **For data model DM2**. Effect of number of heads $H$ on convergence rates when trained with (left) GD + momentum where step-size scales as $\eta = \mathcal{O}\left(\sqrt{H}\right)$; (right) Adam with constant step-size $\eta = \mathcal{O}\left(1\right)$. Quantities plotted are same as in Figure 4. Results demonstrate that overparameterization speeds up convergence in both scenarios.

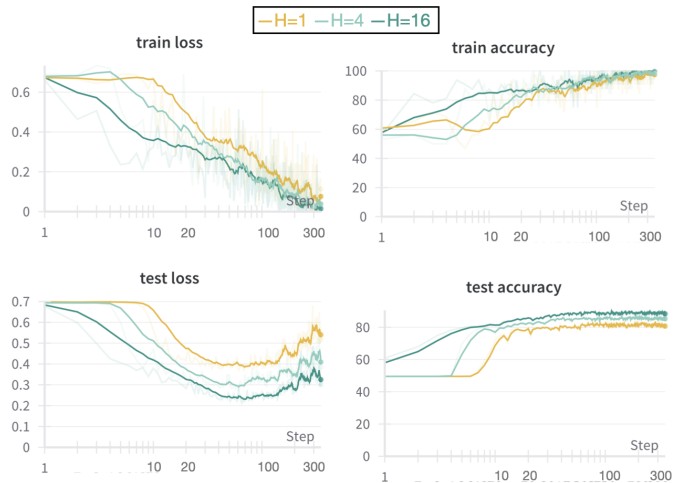

Figure 6: **For SST-2 dataset (Socher et al., 2013)**. Effect of number of heads $H$ on convergence rates when trained with AdamW. Results demonstrate that increasing the number of heads speeds up the training and generalization dynamics.

**Transformers and Self-attention.** The landscape of NLP and machine translation was profoundly reshaped by the advent of Transformers, as pioneered by Vaswani et al. (2017) building upon earlier investigations into self-attention as explored in the works of Cheng et al. (2016); Lin et al. (2017); Parikh et al. (2016). More recent developments include the transformative success of large language models like, LLaMA (Touvron et al., 2023), ChatGPT (OpenAI, 2022), and GPT4 (OpenAI, 2023). Despite this, the learning dynamics of the self-attention mechanism remain largely unknown. Some recent works have focused on understanding the expressive power (Baldi & Vershynin, 2022; Dong et al., 2021; Yun et al., 2020a;b; Sanford et al., 2023; Bietti et al., 2023) and memory capacity of the attention mechanism (Baldi & Vershynin, 2022; Dong et al., 2021; Yun et al., 2020a;b; Mahdavi et al., 2023). Other aspects which are explored include obtaining convex reformulations for the training problem (Sahiner et al., 2022; Ergen et al., 2022), studying sparse function representations in self-attention mechanism (Edelman et al., 2021; Likhosherstov et al., 2021) and investigating the inductive bias of masked self-attention models. Additionally, a sub-area gaining increasing popularity is theoretical investigation of in-context learning, e.g. (von Oswald et al., 2022; Zhang et al., 2023; Akyürek et al., 2023; Li et al., 2023b).

Here, we discuss works that aim to understand the optimization and generalization dynamics of self-Attention or its variants. Oymak et al. (2023) diverges from traditional self-Attention by focusing on a variant called prompt-Attention, aiming to gain understanding of prompt-tuning. Lu et al. (2021) show that for a bag of words model, an attention model optimized with gradient flow for a topic classification task discovers the "topic" word as training proceeds. However, they don't provide finite-time optimization-generalization rates. Jelassi et al. (2022) shed light on how ViTs learn spatially localized patterns, even though this spatial structure is no longer explicitly represented after the image is split into patches. Specifically, they show that for binary classification using gradient-based methods from random initialization, transformers implicitly prefer the solution that learns the spatial structure of the dataset. Li et al. (2023a) provided theoretical results on training three-layer ViTs for classification tasks for a similar data model as ours (tokenized mixture data). They provide sample complexity for achieving zero generalization error, and also examined the degree of sparsity in attention maps when trained using SGD. Contemporaneous works include (Tian et al., 2023; Tarzanagh et al., 2023a): The former presents SGD-dynamics of single-layer transformer for the task of next-token prediction by re-parameterizing the original problem in terms of the softmax and classification logit matrices, and analyzing their training dynamics instead. The latter studies the implicit bias of training the softmax weights $W$ with a fixed decoder $U$ via a regularization path analysis. All these works focus on a single attention head. Instead, we leverage the use of multiple heads to establish connections to the literature on GD training of overparameterized neural networks. Conceptually similar connections have also been studied by Hron et al. (2020) who connect multi-head attention to a limiting Gaussian process when the number of heads increase to infinity. In contrast, we study performance in the more practical regime of finite number of heads and obtain and obtain *finite-time* optimization and generalization bounds.

**Overparameterized MLPs.** There has been an abundance of literature that discusses NN training convergence and generalization dynamics via an NTK type approach, e.g. (Allen-Zhu et al., 2019; Oymak & Soltanolkotabi, 2020; Arora et al., 2019; Nguyen & Mondelli, 2020; Banerjee et al., 2022; Nguyen et al., 2021; Zhu et al., 2023). However, most of these works focus on GD dynamics on regression problems using square loss and relate the training convergence and generalization to the minimum eigenvalue of the Hessian of the NTK. On the other hand, relatively fewer works focus on classification with logistic loss under an NTK separable data assumption, and (Nitanda et al., 2019; Ji & Telgarsky, 2020; Cao & Gu, 2019; Chen et al., 2020; Telgarsky, 2013; Taheri & Thrampoulidis, 2023) are most relevant works that share overlapping ideas with our work. We refer the reader to these for a more thorough overview of the NTK-regime analysis for NNs.

Other than these, Richards & Kuzborskij (2021); Richards & Rabbat (2021); Taheri & Thrampoulidis (2023); Lei et al. (2022) use algorithmic-stability based tools to understand the training dynamics and generalization of GD in shallow NNs. Lei et al. (2022); Richards & Kuzborskij (2021) establish generalization bounds with polynomial width $\tilde{\Omega}(\text{poly}(n))$ requirement while minimizing square loss. Here, we make use of the tools developed by Taheri & Thrampoulidis (2023) who work with self-bounded Lipschitz loss functions, like logistic loss, similar to our analysis. The algorithmic stability arguments in the analysis of the above referenced papers are rooted on a technique to bound generalization gap by directly relating it to train loss based on the notion of average model stability introduced by Lei & Ying (2020). This technique has also been leveraged by Schliserman & Koren (2022) to study linear logistic regression and Nikolakakis et al. (2022) who establish generalization-risk bounds for Lipschitz function optimization with bounded optimal set.

