# OpenReview forum: "On the Optimization and Generalization of Multi-head Attention"
_TMLR — Accepted by TMLR_

### Review · Reviewer_HGup · 2023-12-05

**Summary Of Contributions:**

This paper studies the learning trajectory and generalization properties of a single-layer multi-head attention mechanism trained by gradient descent. Unlike the related work based on NTK, this work does not assume infinite width and infinite head of attention. Additionally, they apply their theoretical results to a bag-of-word synthetic dataset and show that the training involves picking key-query weights of sufficiently large norms and suppresses label irrelevant tokens to saturate the softmax nonlinearity.

**Audience:**

Yes

**Broader Impact Concerns:**

There aren't any major ethical implications of the work.

**Claims And Evidence:**

Yes

**Requested Changes:**

I am not actively working in this area; thus, it would be challenging to provide useful comments on how to improve the work on the theoretical study. Overall, I am not aware of any critical issues. Authors may refer to the Weaknesses for suggestions to strengthen their work. Additionally,  Lu et al. [1] consider a related problem and characterize the training dynamics of attention models. Authors may consider discussing its relationship to this work.

[1] On the Dynamics of Training Attention Models. Haoye Lu, Yongyi Mao, Amiya Nayak, ICLR 2021.

**Strengths And Weaknesses:**

## Strengths ##
1. Although the model under study is simple, I think it is interesting to provide theoretical insights into this popular architecture.
2. The work covers the multi-head cases, which I believe is non-trivial.

## Weakness ##
1. It appears the numerical study is not well-aligned with the theoretical insights, although the authors discuss why this does not contradict their theoretical findings.
2. It would be interesting to see some empirical studies on some simple real-world datasets, e.g. SST2. [1]

[1] Richard Socher, Alex Perelygin, Jean Wu, Jason Chuang, Christopher Manning, Andrew Ng, and Christopher Potts. Parsing With Compositional Vector Grammars. In Empirical Methods in Natural Language Processing (EMNLP), 2013.

---

> ### Author Response · Authors · 2024-02-15
> **Response to Reviewer HGup**
>
> Thank you for your constructive and valuable feedback. Thanks for suggesting this paper. We have added a discussion in the section on related work in the appendix. Additionally, we have added one experiment highlighting the benefits of overparameterization for speeding up the training. Please refer to the global response regarding this.

---

### Review · Reviewer_rp84 · 2023-12-27

**Summary Of Contributions:**

The paper presents a theoretical analysis of multi-head attention in transformer models, contributing new insights into its optimization, that is convergence, and generalization dynamics, with a theoretical application to a tokenized mixture model.

---

## Summary with GPT-4

### Page 1-2
- **Abstract and Introduction**: This paper delves into the Transformer's multi-head attention mechanism and the lack of analyses concerning its optimization and generalization dynamics. The authors establish conditions for a multi-head self-attention model to be stable under certain realizability conditions on the data. They focus on logistic loss optimization and derive bounds on the Hessian of the attention weights, concluding that their results can be generalized to other data models and architectures.

- **Related Work**: The paper situates itself within the field of deep learning and specifically attention mechanisms, citing recent work in optimization and generalization dynamics of gradient descent for Transformers. The literature review spans various studies, including those on single-head self-attention and multi-head attention models, pointing out the research gap in finite-time optimization and generalization dynamics for the latter.

### Page 3-4
- **Preliminaries**: The authors provide notations for the softmax map and attention functions and explain the architecture of a multi-head self-attention model. They also mention the logistic regression loss used for training.

- **Training and Gradient and Hessian bounds of soft-max attention**: Here, the authors discuss the empirical risk minimization framework for training, the gradient and Hessian bounds for softmax attention, and the model gradient/Hessian bounds. They introduce lemmas and propositions to establish the bounds, with detailed derivations provided for the bounds of the softmax attention model's gradients and Hessians.

### Page 5-6
- **Main results**: The authors present theorems detailing training and generalization bounds for multi-head attention. They provide theorems and corollaries that formalize the bounds on training loss and expected generalization gap, including assumptions and implications. These results connect the number of heads in the attention mechanism with the stability of optimization and generalization.

- **Primitive conditions for checking realizability**: This section introduces conditions necessary for the realizability assumption. The conditions involve the data and initialization of the model, and bounds on the model parameters. Corollaries provide generalized bounds under good initialization.

### Page 7-8
- **Application to tokenized mixture model**: This section applies the theoretical findings to a specific data model, the tokenized mixture model. It outlines the model, assumptions, and how to find a good initialization. They also prove that under certain conditions, their approach can achieve NTK separability, which aids in suppressing noise and improving classification accuracy.

### Page 9-10
- **Concentration on maximum margin**: The authors discuss the optimality of the NTK margin in relation to the softmax attention scores, providing propositions that link attention scores to margin maximization. They show that as the number of heads grows, the model approaches a hard-max function, which is desirable for classification tasks.

- **Gradient-based optimization and margin maximization**: Here, they discuss the application of their theoretical bounds to gradient-based optimization methods, specifically how these methods contribute to maximizing the NTK margin. This involves detailed mathematical analysis and lemmas to support their claims.

### Page 11-12
- **Algorithmic stability and expected generalization**: This section provides a framework for algorithmic stability and how it relates to generalization. The authors present lemmas and proofs to support their claims on stability and then tie this into the expected generalization gap.

- **Concluding remarks**: The paper wraps up with a summary of the results and their implications for the training and generalization of multi-head attention layers. The authors note that random initialization of attention weights satisfies NTK separability, and they point to interesting open questions regarding different data models for future research.

This paper is quite technical, focusing on the mathematical aspects of multi-head attention mechanisms within deep learning models, specifically in the context of Transformers used for classification tasks. The authors contribute new theoretical insights into the optimization and generalization dynamics of these models, which are important for understanding the behavior of deep learning algorithms.

**Audience:**

Yes

**Claims And Evidence:**

Yes

**Requested Changes:**

Maybe:

1. the evaluation above so that it is possible to see the order of magnitude of the constants and the tightness of the bounds;
2. adding a schema with the proof and result structure of the paper. (the outline is already quite comprehensive, but it might be nice to have that visualized?).

And fixing the following typos:

1. The derivative of the softmap map in §2 Preliminaries is wrong. It should be $$\varphi'(v) := diag(\varphi(v)) - \varphi(v) \varphi(v)^\top;$$
2. Just before (19) "fives" -> "gives".

**Strengths And Weaknesses:**

The paper explores convergence and generalization. It is well-written and outlines the contributions and steps in detail, making it easy to follow the overall structure.

I am not an expert in this area by any means (not even an intermediate) and I could hardly follow the maths and derivations. As such, I can only acknowledge that it seems correct and likely interesting for experts in the area, given the related work and placement in the literature.

Given that is hard to verify for me: would it be possible to evaluate the bounds on an actual example of a simple MHA architecture? I have no idea what size of constants are involved. Then, it would be interesting to see how this compares to training such a network using finite precision and see how tight the bounds are in practice.

---

> ### Author Response · Authors · 2024-02-15
> **Response to Reviewer rp84**
>
> We have added a schema summarizing the proof technique in the appendix (please see Figure 1). We hope that this adds more clarity to the reader. Thank you for this suggestion.
>
> Also, we have updated Figure 2 highlighting that training and generalization trends observed for the tokenized mixture model show a $O(1/t)$ behavior, as our bounds predict. Since the general bounds obtained by our theory are applicable to a number of data models, it is rather challenging to obtain constants that are tight. Instead, please note that our results capture behaviors with respect to critical problem parameters, such as iterations $t$ and sample size $n$. This feature is not unique to our analysis; rather it is shared by similar techniques in the literature for even simpler models than self-attention. Obtaining bounds that are tight, might be possible under strong assumptions on the data distribution. In general, this requires specialized techniques; thus, we leave it for future work.

---

### Review · Reviewer_LzY5 · 2024-01-18

**Summary Of Contributions:**

The authors derive training and generalization bounds for the multi-head attention layer of transformers under a binary classification setting with logistic loss under gradient descent optimization.
After demonstrating some properties of the loss related to weak-convexity with respect to the model parameters, the authors first show general training bounds bounded by the distance of the initial parameters to a target vector satisfying some realizability condition. The authors then show generalization bounds after K iterations in terms of the empirical loss and the same distance of the target vector as in the training bounds.
The authors show in Sec. 4 that the realizability condition is realized under some initialization regimes and for a large enough number of attention heads H. In Sec. 5, they apply these results to a tokenized mixture model with orthogonal token vectors. They show that the required initialization condition can be found after one randomized initial gradient step from 0, and derive the generalization bounds in this setting given a polylogarithmic number of heads.
The authors leave questions to future work: studying convergence and margin achieved under random initialization; studying the convergence to the optimal margin for the tokenized model, which they show is not possible under this theoretical framework due to the constraints on the weights; and studying different data models.
In the appendix, the authors include all the proofs, as well as an experimental study of the convergence and generalization bounds under the data model.

**Audience:**

Yes

**Broader Impact Concerns:**

No broader impact concerns.

**Claims And Evidence:**

Yes

**Requested Changes:**

I have only minor changes to suggest:
* The cross footnote in page 3 is difficult to parse right alongside a mathematical equation. I would keep the remark in that footnote in the main text.
* p.5: "the minimum eigenvalue becomes less positive" -> I believe this should be: the minimum eigenvalue becomes less negative?
* p.7: patters->patterns in "all patters are orthogonal".

**Strengths And Weaknesses:**

The paper is well written and presents the results in a very instructive fashion, showing how the presented results relate to previous works. The strengths and limitations of the derived bounds is discussed in length, in particular the authors do not shy away from the fact that the bounds cannot accomodate convergence towards the optimal margin of their data model. The proofs appear correct and the derived bounds seem useful and general enough.
Altough not in the main paper and results, the experiments in appendix are somewhat underwhelming as they do not point towards one of the conclusions that I took from the paper, which is that realizability is achieved for a large enough number of attention heads; indeed, the model with one head already achieves the optimal margin and converges faster. I would be interested if there where settings in which the scaling induced by the derived bounds were more appearant.
All in all, I believe this is an excellent contribution to TMLR, and that the framework provided should advance the state of the theoretical study the convergence and generalization properties of attention transformers.

---

> ### Author Response · Authors · 2024-02-15
> **Response to Reviewer LzY5**
>
> Thank you for appreciating the theoretical contributions of our work. We have added an experimental study on a simple dataset where overparameterization speeding up the training is more apparent. Please refer to the global response. Thank you for your careful read of the paper and for the rest of the suggestions; we have incorporated them in the paper.

---

### Author Response · Authors · 2024-02-15
**Global Response to Reviewers**

We thank the reviewers for their positive comments and feedback. We are encouraged that the reviewers find our theoretical study interesting (HGup, rp84), well-written (rp84, LzY5), presented in an instructive fashion (LzY5) and an excellent contribution to advance the theory of transformers (LzY5). We would like to address one common point regarding the effect of heads on a simple real-world dataset:

Figure 6 in the appendix shows the effect of varying the number of heads on the training and generalization dynamics while fine-tuning a RoBERTa model on SST2 dataset. The experiment confirms that overparameterization speeds up the training, especially early during the training process. We thank the reviewers for their suggestion; we believe this experiment indeed serves to complement our theoretical studies, which constitute the main focus of our paper.

---

### Decision · Action_Editor_2sJ3 · 2024-03-06

**Recommendation:** Accept as is

**Comment:**

All reviewers find the contributions of the paper strong and meaningful. The paper considers the challenging problem of analyzing training dynamics of multi head self attention in the non-asymptotic setting. The paper makes good progress in this direction and the results can be build upon in followup works. Overall I am happy to recommend acceptance of this draft.

**Audience:**

Paper studies Training dynamics of self attention, which is of great interest to the community.

**Claims And Evidence:**

Paper provides rigorous analysis of attention dynamics.